# Complexity of avian evolution revealed by family-level genomes

Josefin Stiller[1]✉, Shaohong Feng[2,3,4], Al-Aabid Chowdhury[5], Iker Rivas-González[6], David A. Duchêne[7], Qi Fang[8], Yuan Deng[8,9], Alexey Kozlov[10], Alexandros Stamatakis[10,11,12], Santiago Claramunt[13,14], Jacqueline M. T. Nguyen[15,16], Simon Y. W. Ho[5], Brant C. Faircloth[17], Julia Haag[10], Peter Houde[18], Joel Cracraft[19], Metin Balaban[20], Uyen Mai[21], Guangji Chen[9,22], Rongsheng Gao[9,22], Chengran Zhou[9], Yulong Xie[2], Zijian Huang[2], Zhen Cao[23], Zhi Yan[23], Huw A. Ogilvie[23], Luay Nakhleh[23], Bent Lindow[24], Benoit Morel[10,11], Jon Fjeldså[24], Peter A. Hosner[24,25], Rute R. da Fonseca[25], Bent Petersen[7,26], Joseph A. Tobias[27], Tamás Székely[28,29], Jonathan David Kennedy[30], Andrew Hart Reeve[24], Andras Liker[31,32], Martin Stervander[33], Agostinho Antunes[34,35], Dieter Thomas Tietze[36], Mads F. Bertelsen[37], Fumin Lei[38,39], Carsten Rahbek[25,30,40,41], Gary R. Graves[30,42], Mikkel H. Schierup[6], Tandy Warnow[43], Edward L. Braun[44], M. Thomas P. Gilbert[7,45], Erich D. Jarvis[46,47], Siavash Mirarab[48]✉ & Guojie Zhang[2,4,9,49]✉

Despite tremendous efforts in the past decades, relationships among main avian lineages remain heavily debated without a clear resolution. Discrepancies have been attributed to diversity of species sampled, phylogenetic method and the choice of genomic regions[1–3]. Here we address these issues by analysing the genomes of 363 bird species[4] (218 taxonomic families, 92% of total). Using intergenic regions and coalescent methods, we present a well-supported tree but also a marked degree of discordance. The tree confirms that Neoaves experienced rapid radiation at or near the Cretaceous–Palaeogene boundary. Sufficient loci rather than extensive taxon sampling were more effective in resolving difficult nodes. Remaining recalcitrant nodes involve species that are a challenge to model due to either extreme DNA composition, variable substitution rates, incomplete lineage sorting or complex evolutionary events such as ancient hybridization. Assessment of the effects of different genomic partitions showed high heterogeneity across the genome. We discovered sharp increases in effective population size, substitution rates and relative brain size following the Cretaceous–Palaeogene extinction event, supporting the hypothesis that emerging ecological opportunities catalysed the diversification of modern birds. The resulting phylogenetic estimate offers fresh insights into the rapid radiation of modern birds and provides a taxon-rich backbone tree for future comparative studies.

Understanding the evolutionary relationships among species is fundamental to biology, not only as an account of speciation events but also as the basis for comparative analyses of trait evolution. However, for deep phylogenetic relationships, different studies often show incongruence across analyses[5,6]. Large amounts of data may be required to resolve certain relationships yet others can remain recalcitrant even with genome-scale efforts, particularly for rapid radiations[7,8]. Phylogenomic incongruence can point to statistical and systematic errors but is also increasingly linked to complex biological processes that accompany rapid diversification[9,10]. Prime examples of this problem are the phylogenetic relationships among modern birds (Neornithes), which are inconsistently resolved even with large-scale datasets[1–3,11]. The widespread incongruences in evolutionary histories across avian genomes[1,12,13] has left the phylogenetic relationships of major extant groups unclear and possibly irresolvable[14].

Modern birds comprise three major groups: ratites and tinamous (Palaeognathae), landfowl and waterfowl (Galloanseres) and all other living birds (Neoaves). The early Neoaves experienced rapid diversification into at least ten major clades[15], the so-called 'magnificent seven' and three 'orphans'[12], encompassing 95% of extant species and a significant portion of their phylogenetic diversity. Due to the short internal branches between these clades, their relationships remain contentious[1–3,16]. Furthermore, the timing of the radiation of these major groups is debated[17,18]. The 'mass survival' scenario places the radiation before the Cretaceous–Palaeogene (K–Pg) mass extinction (66.043 ± 0.011 million years ago (Ma)[19]), requiring survival of multiple neoavian lineages through the global changes caused by the Chicxulub impact[11,17,20]. The alternative 'big bang' scenario implies a rapid diversification of neoavian groups following the mass extinction, driven by adaptive radiation into new habitats and in the absence of

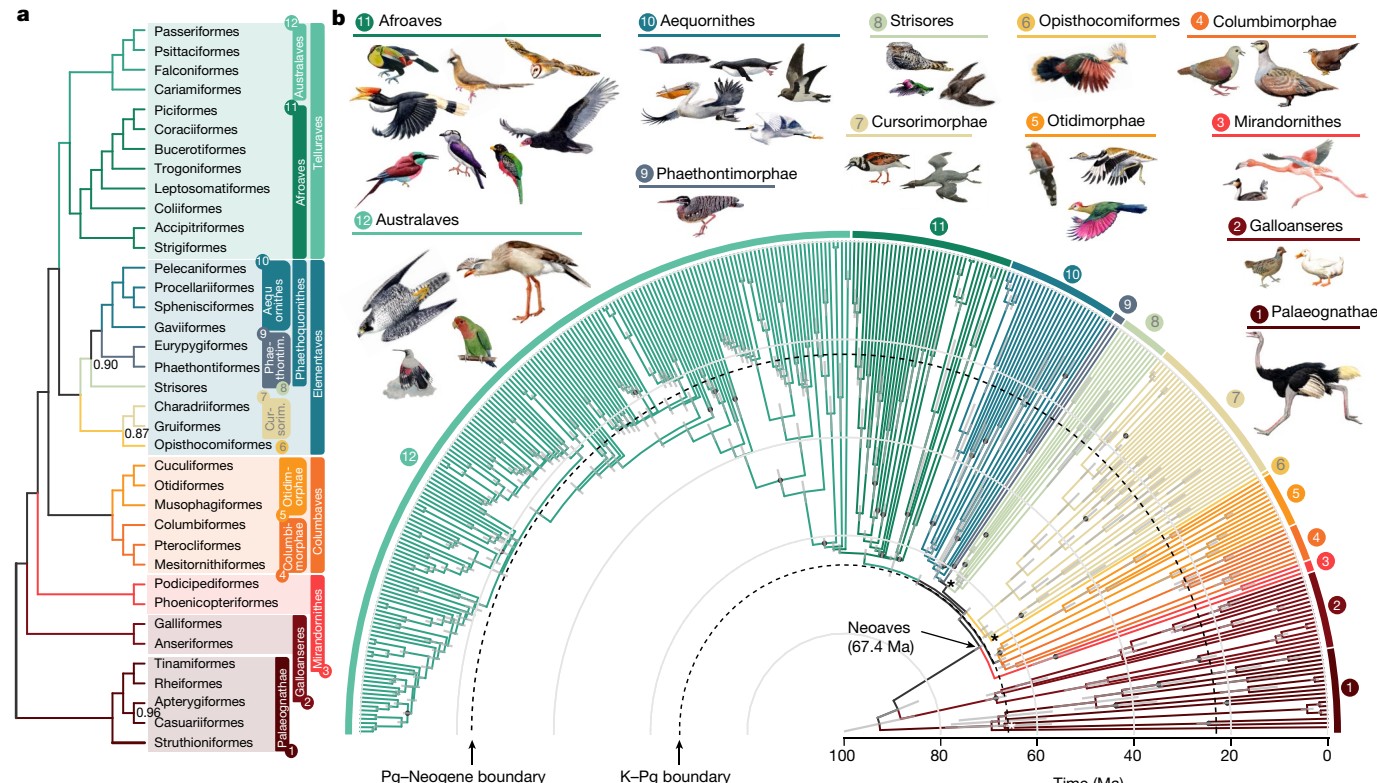

**Fig. 1 | Relationships and divergence times for 363 bird species based on 63,430 intergenic loci. a**, Topology simplified to orders with higher clade names following ref. 50. Numbers on branches represent local posterior probability if below 1. **b**, Time tree of all species. Grey bars represent 95% credible intervals for age estimation; dots indicate nodes with fossil calibrations; asterisks mark the three branches lacking full support. A tree with tip labels is shown in Extended Data Figs. 2 and 3.

competitors[21]. Fossil evidence supports the scenario of morphological diversification following the K–Pg event[22]. Several molecular studies also supported rapid divergences[1–3], yet wide credible intervals allowed for the possibility that some of the earliest neoavian divergences predated the K–Pg boundary[23]. Uncertain placement of key taxa and a wide range of time estimates also persist within Passeriformes, the largest avian order with over 6,000 living species[3,24].

Efforts to resolve high-level avian phylogeny face two major challenges. First, it is difficult to obtain large numbers of orthologous loci with suitable properties for phylogenetic analyses. Many studies have been limited to conserved genomic regions such as protein-coding sequence (exons) and ultraconserved elements (UCEs)[2,25]. Conserved regions exhibit complex patterns of sequence evolution: for example, selection to maintain protein structure and function places constraints on exon evolution[12]. Standard models of sequence evolution practical for large datasets exhibit poor fit to these regions, and model misspecifications probably result in topological discrepancies across data types[1,12,13]. Analysis of large numbers of loci does not remove, but can instead reinforce, biases introduced by model violations[1,7]. In principle, data types under lower selective pressure such as introns and intergenic regions are preferable; intergenic regions are arguably ideal because they are less probably under strong selection[13]. The second challenge is collecting genomic data from sufficient numbers of species, given that dense taxon sampling can improve phylogenetic estimation[26,27]. Thus, the debate in avian phylogenetics has revolved around the trade-off between using diverse loci extracted from entire genomes but for few species (one genome per taxonomic order)[1] or using a smaller number of potentially biased loci sampled from more species[2,3]. Both approaches have shortcomings. The most compelling solution is also the most challenging: to create comprehensive datasets with whole genomes sampled across many taxa that inform on deeper timescales.

Here, as one of the main missions of the 'family phase' of the Bird 10K Genomes Project (B10K)[28], we generated a phylogeny for modern birds by sampling across genome assemblies of 363 species representing 218 families (92% of the total)[4] (Supplementary Data). We analysed nearly 100 billion nucleotides (around 275 Mb for each species; Extended Data Fig. 1a), an alignment 50 times the size of the largest available dataset of 48 species[1] (Extended Data Fig. 1b). As our main data source, we used evenly spaced sampling of intergenic regions across 10 kb windows of a whole-genome alignment[4] (Extended Data Fig. 1c). We found that selection of a 1 kb locus within the first 2 kb of each window balanced phylogenetic informativeness against the inclusion of recombination within loci (Extended Data Fig. 1d and Methods). This resulted in 94,402 loci of 1 kb from which we removed those that overlapped with exon and intron regions, resulting in a set of 63,430 purely intergenic loci (in total, 63.43 megabase pairs). In addition to analysis of this main set we tested the effect of various factors, including additional introns and exons, describe the major sources of phylogenetic incongruence and identify the remaining cases of uncertainty.

## Intergenic regions resolve deep branches

Our main phylogenetic tree (called 'main tree') was obtained by analysis of the 63,430 intergenic loci within a coalescent-based framework (Fig. 1 and Extended Data Figs. 2 and 3). We focus on this tree because the findings reported below show that intergenic regions reduce systematic error due to model misspecifications—results that match a priori expectations and previous analyses[12,29]. The use of a coalescent-based method[30,31] accounts for well-documented incomplete lineage sorting (ILS) in early Neoaves[1,32]. A concatenated analysis of the same 63,430 loci (Extended Data Fig. 4) resulted in a similar tree that differed in only ten of the 360 branches (2.8%). In these topologies, 98.1% of nodes

had full statistical support (main tree, three nodes below 1.00 posterior probability; concatenation, seven nodes below 100% bootstrap support). Although our main topology differed from that of all previous studies, it was more similar to the genome-wide 'TENT' tree from ref. 1 of 48 species than to the main topology from ref. 2, which was based mostly on protein-coding genes of 198 species (Extended Data Fig. 5).

Within Neoaves we resolve four major clades (Fig. 1a), three of which are Mirandornithes (grebes and flamingos), Columbaves (Columbimorphae (doves, sandgrouse and mesites) and Otidimorphae (cuckoos, bustards and turacos)), in addition to Telluraves (higher landbirds including Afroaves and Australaves). The fourth major clade is new and phenotypically diverse, containing Aequornithes (pelicans, tubenoses, penguins and loons), Phaethontimorphae (kagu, sunbittern and tropicbirds), Strisores (nightbirds, swifts and hummingbirds), Opisthocomiformes (hoatzin) and Cursorimorphae (shorebirds and cranes). This clade was supported in coalescent-based analyses of intergenic regions and UCEs, but not by exons, introns or in concatenated analysis of intergenic regions (Fig. 3d and Extended Data Fig. 4). We name this clade Elementaves because its lineages have diversified into terrestrial, aquatic and aerial niches, corresponding to the classical elements of earth, water and air, and several Phaethontimorphae have names derived from the sun, representing fire.

## Most Neoaves diversified post K–Pg

To time calibrate our main tree we empirically generated calibration densities for 34 nodes using 187 fossil occurrences (Supplementary Information) and applied these in a Bayesian sequential-subtree framework (Methods). We estimated branch lengths from intergenic regions and excluded loci that had evolved at the lowest and highest rates, and also those with the greatest rate variation across lineages. Our analysis produced age estimates with 95% credible intervals that were considerably narrower than previously achieved (Extended Data Fig. 6a). The widest credible intervals were observed for nodes positioned furthest from the calibration points, including the secondary calibrations involved in subtree dating. The prospects for narrowing these intervals are promising, through future refinement and the addition of fossil-based age constraints. In contrast to a recent study proposing a diversification of Neoaves during the Upper Cretaceous[11], we found that the early divergences in Neoaves were tightly associated with the K–Pg boundary (Fig. 1b). Only two divergences occurred before the boundary: Mirandornithes diverged from the remaining Neoaves 67.4 Ma (95% credible interval 66.2–68.9) and Columbaves diverged 66.5 Ma (95% credible interval 65.2–67.9). All subsequent divergences postdate the boundary, although the 95% credible interval of the divergence time between Telluraves and Elementaves and the crown age of Elementaves spans the K–Pg boundary. This evolutionary timeline, wherein only a few neoavian lineages diverged before the K–Pg event, is reflected in all alternative dating analyses (Methods and Extended Data Fig. 6b–e), highlighting the robustness of our estimated chronology. This lends more support to a post-K–Pg diversification of Neoaves than previous studies, where the 95% credible interval of between ten and 18 of the nodes allowed for pre-K–Pg divergences[1,2,18,23].

## Abundant discordance among gene trees

Assessing the level of incongruence between gene trees (GTs) across the tree, order-level relationships ranged from showing little or no discordance to high levels of discordance (measured by the quartet score; Fig. 2a). The percentage of GT quartets matching a species-tree branch at the ordinal level ranged from 99.9 to 33.7% (close to one in three, which corresponds to a polytomy). In particular, 14 nodes had quartet support below 37%. These are the same nodes that have proved difficult to resolve in past studies[15]. For 29 out of 33 nodes, the quartet support of the main topology was significantly higher than both

alternatives (one-sided $\chi^2$ test with Benjamini–Hochberg multiple test correction), consistent with expectations under ILS models. We discuss the remaining nodes (26, 39, 43 and 49 in Fig. 2a) below.

## Mirandornithes is sister to other Neoaves

The placement of Mirandornithes (also called Phoenicopterimorphae[33]) as the sister lineage to the remaining Neoaves was supported by both the main tree and concatenation. Although this topology was reported previously[3] it differs from the TENT tree from ref. 1, which grouped Mirandornithes and Columbimorphae into a clade called Columbea. In our main tree, Columbimorphae combined with Otidimorphae to form Columbaves. This clade has also been reported previously, albeit with low bootstrap support[2]. Mirarab et al.[34] showed that a 21 Mb outlier region of chromosome 4 with abnormally strong signal for Columbea (potentially due to the effects of ancient interchromosomal rearrangements) is responsible for the previous recovery of Columbea. However, with additional taxon sampling of Otidimorphae and Columbimorphae, the effect of this outlier region gradually lessened in favour of an increasingly dominant signal from the rest of the genome that placed Mirandornithes as the sister to other Neoaves (Extended Data Fig. 7a). In the concatenated analysis, Mirandornithes and Columbimorphae are successive sister groups to the remaining neoavian clades but with limited support (bootstrap = 64; Extended Data Fig. 4). Finally, when analysing exons, Mirandornithes were placed deeper in Neoaves as sister to Aequornithes + Phaethontiformes (Extended Data Fig. 4), which may relate to previous association with mostly aquatic birds in studies analysing large portions of coding regions (sister to Charadriiformes[2], Opisthocomiformes + Aequornithes + Phaethontimorphae[11]).

There is a rapid succession of nodes in this part of the tree, with only 0.92 Ma between the divergence of Mirandornithes and of Columbaves from other groups. Within Columbaves, Otidimorphae has been found in some studies[1,2] but not in others[3,12]. Within Otidimorphae we resolved Otidiformes as the sister group to Cuculiformes, like some studies[12] but unlike several others[1–3]. The difficulty in this case could be explained by the very short branch (0.57 Ma) separating Otidiformes and other Otidimorphae. Similarly, Columbiformes diverged from the remaining Columbimorphae within 0.26 Ma. These fast divergences partially explain why previous analyses with fewer data led to conflicting resolutions of these earliest neoavian branches.

## Waterbirds are deep in a diverse clade

Unlike previous hypotheses that placed Phaethoquornithes (Aequornithes + Phaethontimorphae) as sister to landbirds[1,3], the main tree placed Phaethoquornithes deep inside the diverse Elementaves (Fig. 1a). The 'orphans' Charadriiformes and Gruiformes were consistently grouped together (forming Cursorimorphae), as found in some other studies[1,3]. The placement of the third orphan, Opisthocomiformes, as the sister to this group (with a short branch of 0.58 Ma) was the sole instance across the entire phylogeny with statistically indistinguishable levels of GT support for all three possible configurations around this branch[35] (node 43 in Fig. 2b), a noteworthy finding given the extensive amount of available data.

Whereas the main tree placed Phaethontimorphae as the sister to Aequornithes, further investigations showed a competing placement as the sister lineage to Telluraves. Both topologies have previously been reported[1–3,12], with their difference attributed to the effects of using protein-coding (Phaethontimorphae + Aequornithes) versus non-coding regions (Phaethontimorphae + Telluraves)[15]. We found instead that both topologies have support in the intergenic data. Whereas Phaethontimorphae + Aequornithes had a slightly better quartet score, it was recovered in only 60% of trees resulting from random subsampling of half of the 63,430 loci (Extended Data Fig. 7b). The two alternative positions of Phaethontimorphae, which are three

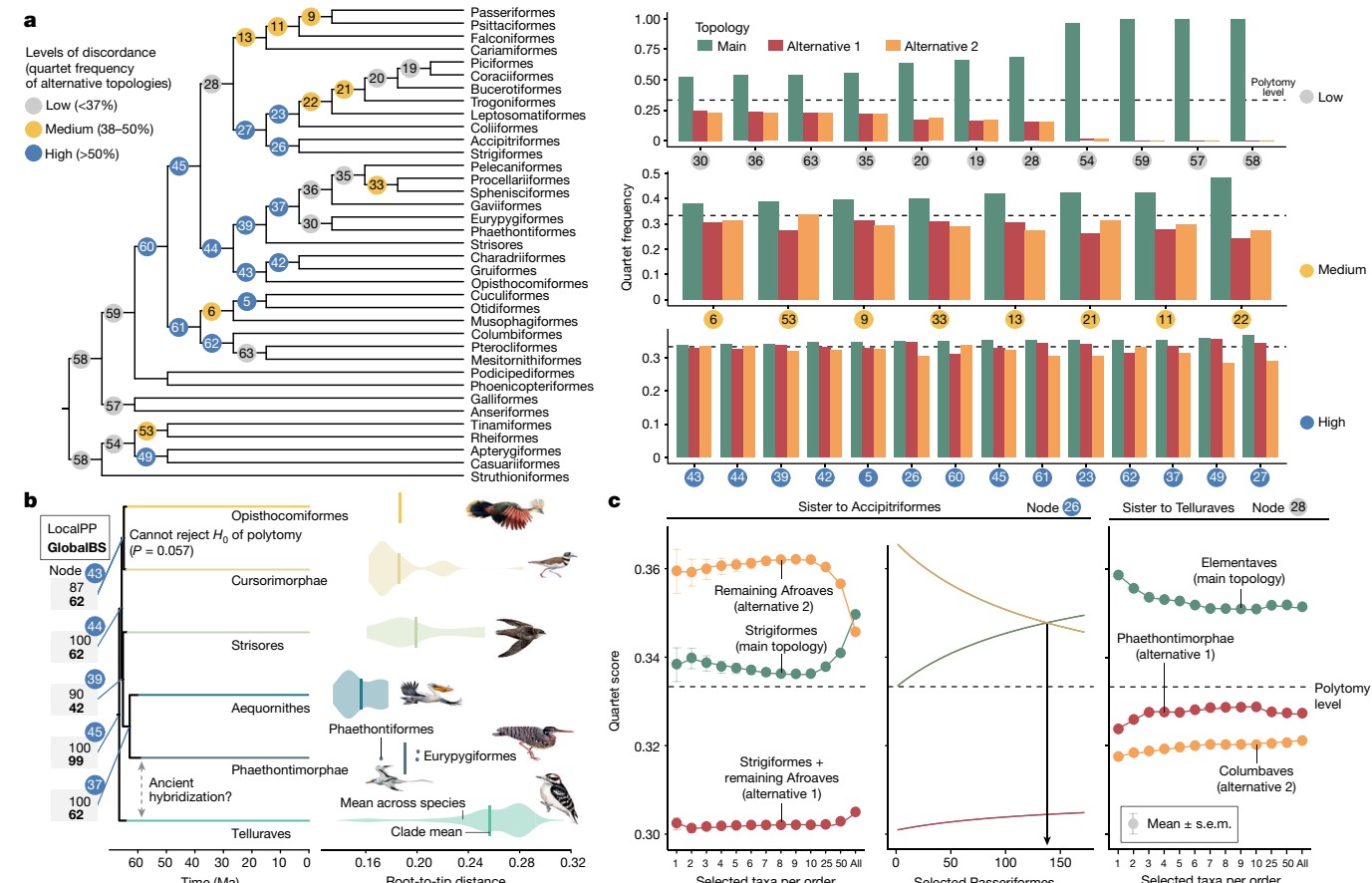

**Fig. 2 | Explaining difficult placements. a**, Gene tree discordance across the backbone of the main tree. Node colours and numbers represent the bar plots of quartet frequencies for three possible resolutions around each branch. **b**, Uncertainty at the base of Elementaves. Phaethontimorphae + Aequornithes had high local posterior probability (LocalPP), but global bootstrap resampling (GlobalBS) showed support for an alternative placement. Violin plots (points for the species-poor Phaethontiformes) show higher root–tip distances of Phaethontiformes, and particularly for Eurypygiformes, than Aequornithes, which may cause attraction to the long-branched Telluraves. Further, the

placement of Opisthocomiformes is the only branch where a null hypothesis ($H_0$) of a polytomy cannot be refuted. **c**, Addition of taxa occasionally affects topology and support. Across 41,918 GTs with at least one species from each group, the alternative placement of Afroaves + Accipitriformes had higher quartet support when only a few species were sampled but declined with increasing taxon sampling (left), particularly of Passeriformes. The main topology dominated when 138 or more passerines were sampled (middle, arrow). Support for Telluraves + Elementaves decreased with increasing taxon sampling (right).

branches (9.1 Ma) away, each had full local support (posterior probability = 1.0). Nevertheless, global bootstrap support estimated from resampling of GTs showed uncertainty in the three nodes connecting the two placements (global bootstrap = 42–62; Fig. 2b). Two hypotheses could explain this non-local uncertainty, the first being ancient hybridization between ancestral Phaethontimorphae and Telluraves 3.96 Ma after their divergence. Alternatively, the high support for the alternative placement could be due to problems arising from long branches. Phaethontimorphae have around 25% longer terminal branches than Aequornithes (paired $t$-test across loci, $P < 2.2 \times 10^{-16}$), showing greater similarity to Telluraves in this regard (Fig. 2b). Consistent with this explanation, topological changes resulted from data filtering that targeted long branches (clocklikeness, stemminess, total coverage and tree length; Extended Data Fig. 7c).

Our main tree placed Strisores (also called Caprimulgiformes[33]) with Phaethoquornithes with moderate support (posterior probability = 0.90; Fig. 1a), but the concatenated tree grouped them as sister to Telluraves with low support (bootstrap = 32; Extended Data Fig. 4). Quartet frequencies did not follow an ILS-alone scenario, because moving Strisores to the base of Elementaves had quartet frequencies similar to the main tree ($\chi^2$ test, $P_{\text{Benjamini–Hochberg adjusted}} = 0.317$, node 39), but the third alternative had lower frequency ($P = 0.488 \times 10^{-11}$). Possible explanations include hybridization or long branch attraction, because

Strisores have 4–28% longer branches than the other Elementaves, which may attract them to the long-branched Telluraves (Fig. 2b). Previous studies also failed to find unequivocal support for the relationship of Strisores, placing it as sister to Otidimorphae[1], Cursorimorphae[11], Opisthocomiformes[3] or all other Neoaves[2]. Within Strisores our tree positioned Caprimulgidae (nightjars), rather than Sedentaves (oilbird + potoos)[12], as sister to all others (Extended Data Fig. 2), as found previously[2,11].

## Difficult placement of owls and hawks

Within Telluraves our main tree supported the proposed split into Australaves and Afroaves[1,3] in contrast to other studies[2,11]. Our tree grouped Accipitriformes and Strigiformes as the sister to the remaining Afroaves, similar to previous coalescent-based analyses[1]. Concatenated analyses[1,3], including ours, supported Accipitriformes alone as sister to the remaining Afroaves (Extended Data Fig. 4). This node also showed quartet frequencies that were statistically indistinguishable for two topologies (35 versus 34.6%, $\chi^2$ test, $P_{\text{Benjamini–Hochberg adjusted}} = 0.130$), but the third was significantly lower (30.5%, $P < 10^{-16}$; node 26 in Fig. 2a), contradicting expectations of ILS. Because we found no evidence of long branch attraction (Extended Data Fig. 7d), the non-ILS patterns could be indicative of ancestral hybridization[36]. In contrast to GTs,

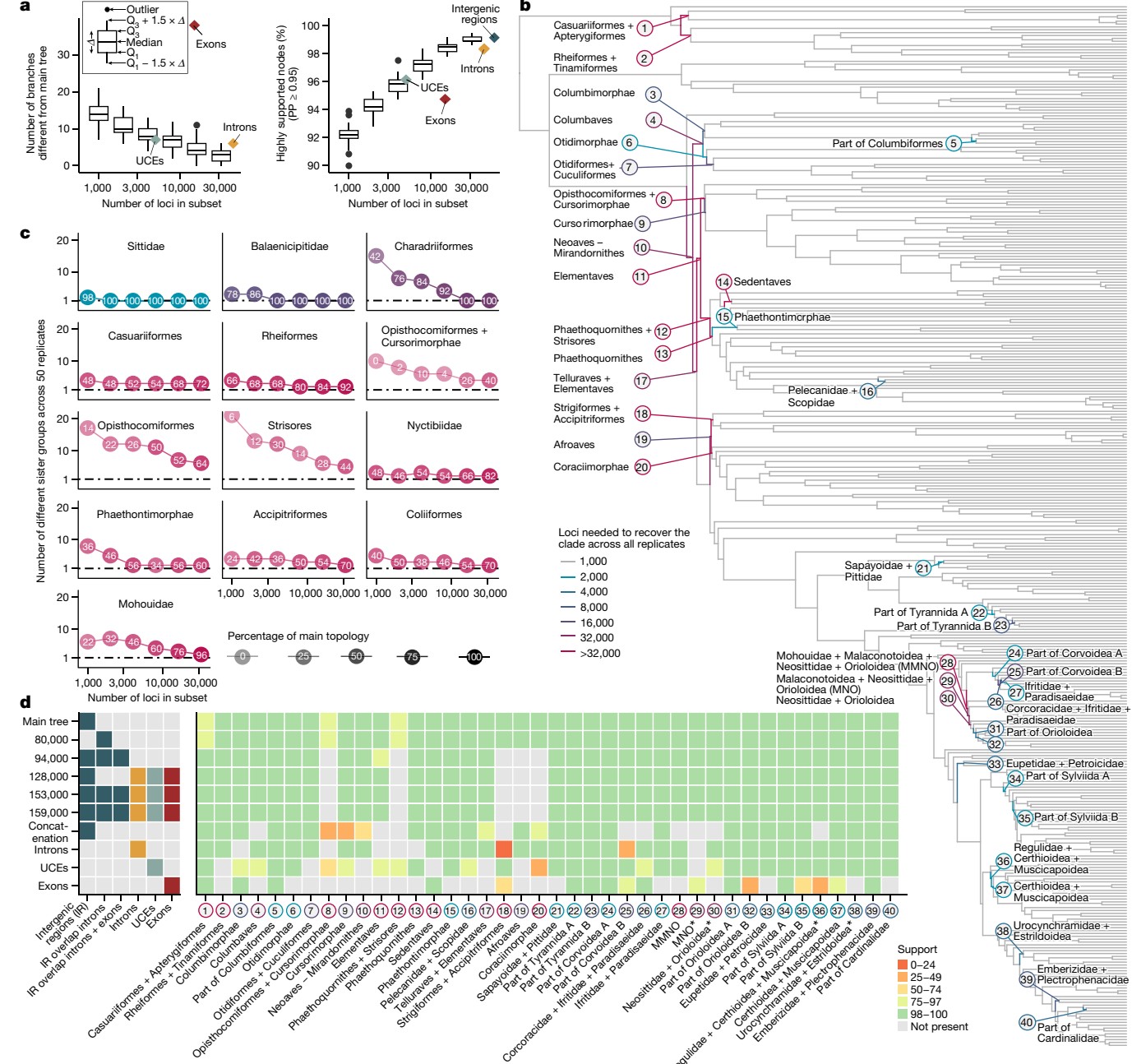

**Fig. 3 | Effect of increasing data quantity. a–c**, Species trees were reconstructed from subsets of GTs (1,000, 2,000, …, 32,000) of the 63,430 intergenic regions in 50 replicates. **a**, The addition of loci increases similarity to the main tree (left) and increases the proportion of highly supported nodes (right). **b**, The main tree, with branches coloured according to the difficulty involved in consistently recovering the clade across subsets. Most branches were consistently obtained with only 1,000 GTs (grey); the remaining 40 branches required more loci. **c**, Increasing the number of loci decreases the number of possible sister groups. We recorded the number of unique sister groups for each node across subsets. Colours correspond to the difficulty (from **b**), and shading and number show the frequency, with which the main topology was obtained. The top row illustrates examples of easy nodes. in which the same sister group was consistently recovered with 2,000, 4,000 and 16,000 loci, respectively. The remaining plots show the most difficult nodes, in which multiple sister groups were supported even when 32,000 loci were subsampled. **d**, Ten selected species trees, data types used in each and the support for all challenging branches (labelled in **b**). Asterisks indicate relationships in Passeriformes that differ from previous studies. MNO, Malaconotoidea + Neosittidae + Orioloidea; MMNO, Mohouidae + MNO, PP, posterior probability; Q, quartiles.

direct analysis of alignment sites using CoalHMM (Methods) supported an ILS-like pattern in which the two alternative topologies had similar scores (31.2 versus 29.6%). However, CoalHMM assumes ILS a priori and only a strong signal of hybridization can lead to inferring unbalanced quartet frequencies. Thus an ancestral hybridization event, albeit too weak to be detected by CoalHMM, remains plausible. In addition, we observed that the relationship between Accipitriformes and

Strigiformes depended on the number of passeriform taxa sampled. The main topology was obtained only when at least 138 Passeriformes were included, whereas sampling fewer taxa of each order favoured Accipitriformes as the sister to the remaining Afroaves (Fig. 2c). This case demonstrates that the effect of taxon sampling of one group can extend to others and that these sampling effects are not easily predictable.

## Insights into the passerine radiation

Our analyses of phylogenetic relationships among Passeriformes (perching birds) included 173 species in 121 families and seven fossil calibrations. The most recent common ancestor of Passeriformes was dated to 50.7 Ma (95% credible interval 48.3–53.0; Fig. 1). This estimate is broadly similar to those from other studies with broad taxon sampling (47–53 Ma (refs. 2,3,23,24)), whereas a previous genomic study that included only five passeriforms found a considerably younger age (39 Ma (ref. 1)). The split between Tyranni (Suboscines) and Passeri (Oscines) was estimated at 47.3 Ma (95% credible interval 45.1–49.8; Extended Data Fig. 3), in line with a previous study[2], but 3–4 Ma older than other estimates[3,24]. Tyranni and Passeri were estimated to have started diversifying around the same time whereas other studies supported a 3 Ma difference between the onset of their diversification[2,3]. The three main clades of Tyranni (Eurylaimides, Tyrannides and Furnariides) were inferred to be 4–12 Ma younger than previously found[37]. In Passeri, the age of Corvides was estimated to 25.7 Ma (95% credible interval 23.8–27.7), agreeing with some previous estimates[24] but over 5 Ma younger than others[3]. The divergence of a major subclade of Passerides (Sylviida + Muscicapida + Passerida) was inferred to have occurred shortly after the Palaeogene–Neogene boundary (22.4 Ma, 95% credible interval 20.6–24.2; Extended Data Fig. 3) whereas previous studies placed its divergence before the boundary[3,23,24]. This branch and some subsequent divergences occurred in close succession, indicating rapid diversification.

In Passeri, our tree differed from studies based on UCEs or 5′-untranslated region sequences[3,24,38], including in the positions for Orioloidea, Malaconotoidea, Corvoidea, Mohouidae, Neosittidae, Regulidae and Urocynchramidae (Fig. 3d (asterisks) and Supplementary Information). Some of these difficulties also appear to be related to rapid diversification, seen for example in the extremely short internode of Mohouidae (0.18 Ma).

## Rheas have conflicting placements

Outside of Neoaves we found support for different relationships of Rheiformes within Palaeognathae, a conflict previously attributed primarily to ILS[39]. Whereas our main topology found Rheiformes as the sister to Tinamiformes, analysis with CoalHMM put it as sister to Apterygiformes + Casuariiformes (Extended Data Fig. 7g), in agreement with that previous study[39]. We found that Rheiformes and Tinamiformes had a higher proportion of loci with high guanine–cytosine (GC) content than other taxa (Extended Data Fig. 7e). We observed that omission of loci with similar GC content for Tinamiformes and Rheiformes, but not for others, tended to reduce (but not eliminate) support for this clade (Extended Data Fig. 7g). These results suggest that the strong support for this grouping in our main tree was enhanced by biased GC content, leaving other placements of Rheiformes (for example, as sister to Apterygiformes + Casuariiformes, as recovered by CoalHMM) as plausible.

## Effect of taxon sampling varies

The question of whether to sample more species or more genetic loci is pivotal in phylogenetic study design[40]. Whereas expansion of taxon sampling helps to mitigate the confounding impact of long branches within GTs[26,41], its effects on species-tree inference are less clear. To investigate this question we randomly selected between one and ten species for each order and constrained the 63,430 intergenic GTs to the selected taxa before rescoring the species tree. These changes in taxon sampling affected ordinal relationships in only three cases (Extended Data Fig. 7f), with the aforementioned Accipitriformes + Strigiformes being the strongest example (Fig. 2c). More frequently we observed that increasing taxon sampling affected only the amount of GT discordance

but not the topology (for example, Telluraves + Elementaves in Fig. 2c). Thus our results are relatively robust to taxon sampling, although with some exceptions.

## Number of loci needed varies across nodes

As access to large numbers of loci becomes common, the choice of how many and which loci to select is a fundamental decision[42]. Using repeated subsets of the 63,430 dataset, we found that greater locus sampling resulted in trees more similar to the main tree and with higher support (Fig. 3a). The same trend was observed across all partitions of the genome (intergenic regions, introns, UCEs and exons; Extended Data Fig. 8a,b) and with other species trees as reference, except the purely exonic one (Extended Data Fig. 8c).

We assessed how many loci were required to consistently recover each clade of the main tree (Fig. 3b). We found that most clades (321 of 361, 89%) could be identified with just 1,000 loci. A minority of clades (30 of 361, 8%) needed substantially more, from 2,000 to 32,000 loci, before analyses could consistently support them (Fig. 3c). In the remaining ten clades (2.8%) increasing the number of loci reduced incongruence but did not consistently recover the main topology across replicates, even with 32,000 loci (Fig. 3c and Extended Data Fig. 9). Most of these difficult nodes were associated with short branches after the K–Pg boundary and within Corvides (Fig. 3b). For example, mousebirds (Coliiformes), placed in agreement with some studies[1–3] in our main tree, had an alternative placement in 30% of subsets of 32,000 loci, consistent with previously reported difficulties[1,14].

## Strong effects of different locus types

Species trees built from GTs of different data types were substantially different, especially between protein- and non-coding data, akin to previous findings[1,12,13]. The species tree built from 14,355 exon loci (excluding the hypervariable third codon position) differed in 38 of 360 branches from the main tree (compared with six or seven differences for the other data types; Extended Data Fig. 4). Beyond dissimilarity to the main tree (Fig. 3d), trees inferred from exons were less internally consistent—they were more sensitive to subsampling than trees built from other data types (Extended Data Fig. 8a–c). Even when controlling for the number of GTs used in species-tree construction, exons produced more variable trees than other data types (Fig. 4a).

We found that data types differed in regard to the risk of violating assumptions of phylogenetic models. A much higher proportion of exonic loci was found to be at risk of sequence saturation (30.83%) compared with the other data types (intergenic regions, 0.07%; UCEs, 0.34%; introns, 0.83%). The evidence for violation of stationarity was generally low, yet highest among exons (exons, 2.45% of loci failing the test; UCEs, 0.02%; intergenic regions, 0.07%; introns, 0.08%). Moreover, because individual exons of the same gene were combined into one locus, the assumption that phylogenetic loci are recombination free is expected to be more frequently violated by exonic loci. An exonic locus can span wide stretches of the genome because its individual exons are not contiguous (mean sequence length 16,964 base pairs, range 149–566,199) as opposed to loci of other data types (mean sequence length: introns, 2,543 base pairs; UCEs, 2,095 base pairs; intergenic regions, 897 base pairs). Because the increased length of exons increases the risk of within-locus recombination, analysis of only intergenic regions minimizes the risk of recombination and model violations.

We found that exonic loci had less phylogenetic information and were more variable in their signal than the other data types (Extended Data Fig. 8d,e). Exons also scored highest in a measure of phylogenetic estimation difficulty (Extended Data Fig. 8f), indicating that their GTs are less reliable than those of other data types. To examine whether

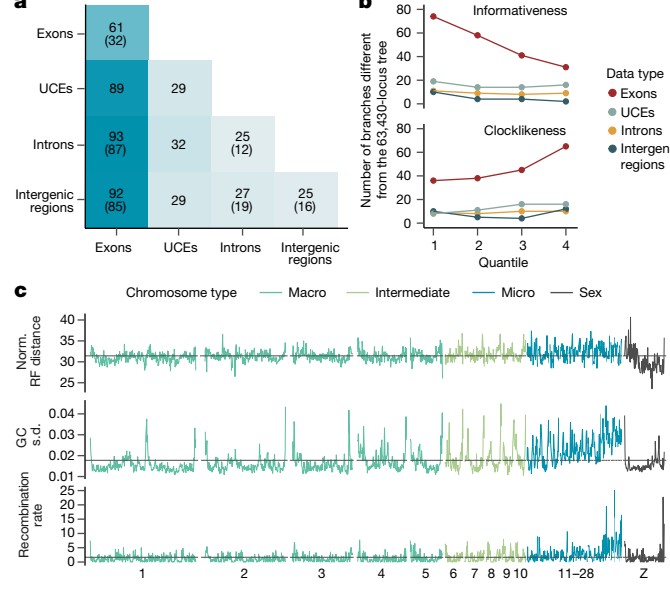

**Fig. 4 | Phylogenetic signal across the genome. a**, Protein-coding regions yield more varied species trees when they are subsampled. Each heatmap cell shows the average Robinson–Foulds distance between 1,250 (diagonal, 1,225) pairs of species trees, each built from 2,000 GTs of different data types. Values in parentheses give the same metrics for 8,000 GTs, omitting UCEs with fewer loci. **b**, Effect of subsetting loci by data type and different metrics. The *y* axis represents the number of differences to the main tree; the *x* axis shows two metrics split into four quartiles, from low to high. Phylogenetic informativeness is the proportion of parsimony-informative sites. Clocklikeness is the coefficient of variation in root–tip distances, a measure of branch length heterogeneity. Extended Data Fig. 8g shows other metrics. **c**, Patterns of phylogenomic incongruence along the genome. Using the 94,402 loci binned approximately every 500 kb, lines show Robinson–Foulds (RF) distances to the main tree (top), variance in GC content (middle) and recombination rate (bottom). Horizontal lines indicate genome-wide averages.

exons had a misleading signal, we restricted species-tree inference to GTs with more signal, less gappy alignments, greater clocklikeness and greater total length. Unlike intergenic regions, in which subsampling did not systematically change the species trees, the use of more informative, less gappy and more clocklike exons reduced incongruence between the resulting species trees and the main tree (Fig. 4b and Extended Data Fig. 8g). Thus exons yield phylogenetic trees that are less reliable. This conclusion is consistent with earlier analyses based on fewer genomes[1,12,13,29]. Our results indicate that the damaging effects of model violation and limited signal of exons are not offset by increased taxon sampling, as one might hope[2,43].

To investigate whether the confounding effects of exons could be swept out by other data, we gradually augmented purely intergenic loci (Extended Data Fig. 1b). The addition of 1 kb windows overlapping with introns (resulting in a total of 80,047 loci) led to the same topology (Fig. 3d). However, when windows overlapping with exons were added (94,402 loci), the resulting tree agreed with the main tree on the first four neoavian clades (Mirandornithes, Columbaves, Telluraves and Elementaves) but differed in five difficult branches (Fig. 3d and Extended Data Fig. 4). This 94,402-locus topology was also obtained when adding UCEs, purely intronic loci and purely exonic loci (not those overlapping with 1 kb windows) to either the 63,430 set (128,233 loci) or the 94,402 set (159,205 loci). Removal of loci that failed saturation and stationarity tests from the full set (153,789 loci remaining) returned the same tree, albeit with low support on branches conflicting with the main tree. These results indicate that the inclusion of exonic loci,

even if these constitute just 10% of the data and are restricted to those that pass the testing of model fit, can affect the most unstable parts of the tree. This finding can partially explain the different topologies reported in other studies using a high proportion of coding regions[2,11]. By contrast, exclusion of introns did not make a difference topologically in our analyses. Nevertheless, we treat as uncertain the five branches that differ between purely intergenic regions and these alternative trees (Fig. 3d).

## Discordance along chromosomes

Averaged over 500 kb windows, GT discordance levels were mostly consistent along chromosomes (31.4% normalized Robinson–Foulds distance to the main tree; Fig. 4c). However, we observed some notable troughs and peaks of GT discordance, particularly around the telomeres and some centromeres (relative to the chicken genome), agreeing with previous findings regarding telomeres[1]. Gene trees inferred from macrochromosomes (below 50 Mb) were slightly less distant to the main tree than intermediate chromosomes (12–40 Mb) and microchromosomes (average size 12 Mb; Extended Data Fig. 10a). The higher discordance near telomeres and across microchromosomes may be related to their elevated richness of genes, variation in GC content and higher recombination rates (Fig. 4c and Extended Data Fig. 10b–d) leading to higher local effective population size and challenging phylogenetic reconstruction. The Z chromosome had the lowest discordance (Extended Data Fig. 10a), consistent with its lower recombination rate. Species trees inferred from individual chromosomes resulted in topologies with 1–3% difference to the main tree, with most differences observed in microchromosomes followed by intermediate chromosomes (Extended Data Fig. 10a).

## Implications for avian diversification

We next evaluated how well the new phylogenetic tree reflects avian morphology, testing the expectation that closely related species should resemble one another. We found that our main tree fits morphological traits better than the topology of ref. 2, even when controlling for taxon sampling (Fig. 5a), including the larger number of Passeriformes in our study (supplementary results given in Supplementary Information). Simulations considering the misplacement of taxa and convergent scenarios suggested that the higher phylogenetic signal in this comparison was more probably attributed to topological differences (Extended Data Fig. 11a).

Next we compared branch lengths in time units and coalescent units, which should be proportional to population size, ignoring the effect of varying generation time (Methods). We found a strong signal of increased population sizes on nearly half of the branches 0–2 Ma following the K–Pg transition (Fig. 5b), in agreement with an earlier analysis of insertions and deletions[44]. This pattern could be indicative of lineages undergoing density compensation, a transient increase in population size in response to ecological opportunity and release that may be associated with adaptive radiation[45]. Birds would have been well positioned to exploit landscapes newly devoid of competitors and predators following the K–Pg mass extinction because of their flight capabilities. Vagile insectivores and marine species such as Strisores and Aequornithes could have rapidly expanded into early-succession habitats. A less marked spike was also observed around the end of the Palaeogene (Fig. 5b). There was also an apparent gradual decline in the ratio of time and coalescent unit branch lengths by close to an order of magnitude over 60 Ma. A reduction in generation times could plausibly produce this result, possibly reflecting an increase in numbers of passerine families through time. There has also been a trend toward reduced inferred body sizes over this time (Fig. 5c), and it has long been appreciated that taxa with small body size have short generation times[46].

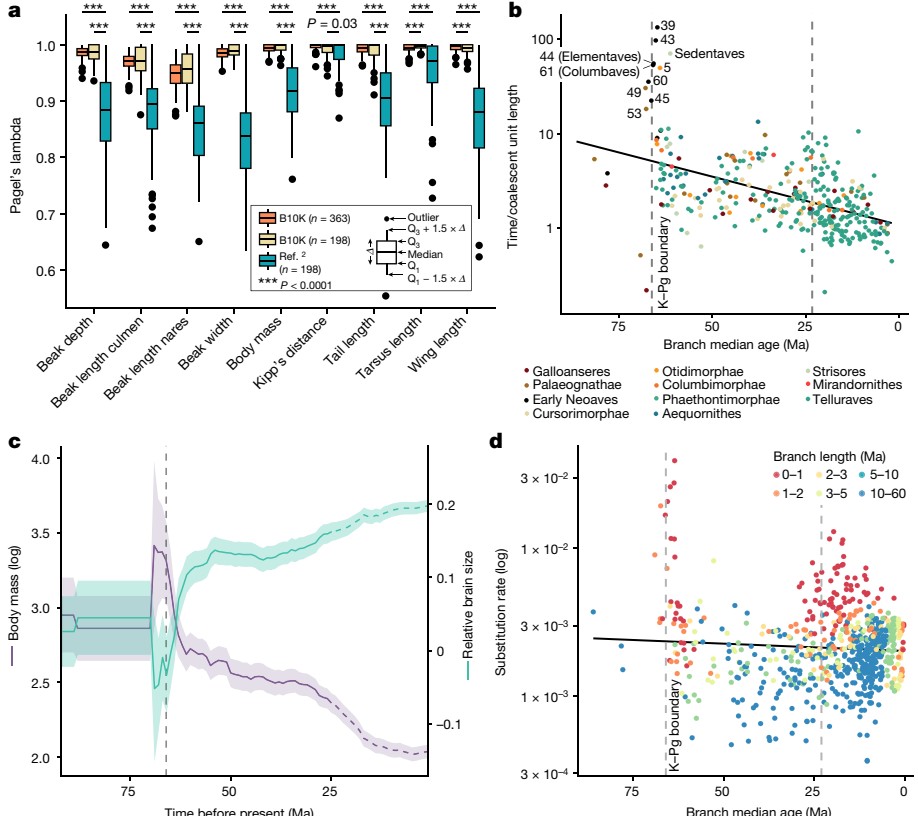

**Fig. 5 | Biological implications of the new time tree. a**, The main tree fits morphological traits well. We measured phylogenetic signal (Pagel's lambda) for nine traits over 100 replicates and compared the fit based on (1) the main tree, (2) the ref. 2 topology and (3) the main tree with random species sampling to match the sample size used in ref. 2 (one-sided *t*-test with Bonferroni correction). **b**, The K–Pg and Palaeogene–Neogene transitions were associated with increased effective population sizes of some lineages. Shown are the midpoint ages of each branch compared with the ratio between its length in time units and in coalescent units, which is proportional to the effective population size of that branch and its generation time. Numbers correspond to selected nodes from Fig. 2a. **c**, Variations in body mass and relative brain size over time changed in different directions following the K–Pg event. Solid lines indicate mean values and ribbons mark 95% confidence intervals. The dashed parts of the reconstruction (from 25 Ma) indicate possible uncertainty due to the lack of within-family sampling (Extended Data Fig. 11g). **d**, Substitution rates increased around the K–Pg boundary. Estimated molecular rates for the intergenic regions are plotted against the midpoint age of each branch.

Substitution rate estimates for the intergenic regions also showed a strong increase at and shortly after the K–Pg boundary (Fig. 5d), and a more diffuse increase near the boundary to the Neogene. The rate increase near the K–Pg boundary has been noted for other data types and attributed, at least in part, to the 'Lilliput effect' (refs. 47,48). This refers to decreases in body size in the wake of mass extinctions; those changes in body size would affect other life history traits, such as generation time. Consistent with this explanation, we found a decrease in reconstructed body size following the K–Pg event (Fig. 5c). This was accompanied by an increase in inferred relative brain size shortly before the K–Pg event, suggestive of strong selection for adaptability or behavioural flexibility, consistent with previous findings[49]. Shortly after the K–Pg event, the continuous changes of inferred relative brain size appear to have ceased (Fig. 5c). From around 35 Ma the reduction in reconstructed body mass does not seem to have been accompanied by an increase in relative brain size.

Across the tree we found that rapid evolutionary change occurred at the origin of major clades, throughout the diversification of some clades and along some isolated branches. Passeriformes exhibited a burst of body mass evolution at their most recent common ancestor (Extended Data Fig. 11b). Rates of evolution in relative brain size were more variable, with rapid evolutionary change in some clades (for example, Telluraves, vocal learning lineages such as parrots, corvids and hummingbirds)[49]. In addition, our data showed that the early burst was followed by sustained varied rates within these groups, especially in Passeri (Extended Data Fig. 11c).

## Conclusions

Relationships along the backbone of Neoaves have long been contentious, with various analyses yielding incongruent results. At the heart of the disagreements has been a long-standing question: is it better to sample many taxa at a few loci (typically conserved regions, such as exons and UCEs) or sample many loci widely across the genome, even if available from fewer species? We can finally answer this question because our data provide both dense taxon sampling and many loci across the whole genome. We observed that the number of loci, in addition to sequence types (for example, exons, introns, intergenic regions or chromosome type), had a much greater effect on the inferred tree than taxon sampling. Nevertheless, increased taxon sampling was crucial in inferring more precise dates, and for studying traits, trajectories of population size and substitution rates. By focusing on intergenic regions, a source of data largely unused in the past, we minimized model violations and increased phylogenetic resolution. Nonetheless, our results also showed that several recalcitrant relationships remain, even with this wealth of data, due to challenges imposed by biological processes such as hybridization that are hard to model in deep time using phylogenetics. Overall, our results underscore the

complexity of genome evolution and show methodologies that are likely to be useful for future phylogenomic studies focused on deep relationships.

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

[1]Section for Ecology and Evolution, Department of Biology, University of Copenhagen, Copenhagen, Denmark. [2]Center for Evolutionary & Organismal Biology, Liangzhu Laboratory & Women's Hospital, Zhejiang University School of Medicine, Hangzhou, China. [3]Department of General Surgery, Sir Run-Run Shaw Hospital, Zhejiang University School of Medicine, Hangzhou, China. [4]Innovation Center of Yangtze River Delta, Zhejiang University, Jiashan, China. [5]School of Life and Environmental Sciences, University of Sydney, Sydney, New South Wales, Australia. [6]Bioinformatics Research Centre, Aarhus University, Aarhus, Denmark. [7]Center for Evolutionary Hologenomics, The Globe Institute, University of Copenhagen, Copenhagen, Denmark. [8]BGI Research, Shenzhen, China. [9]BGI Research, Wuhan, China. [10]Computational Molecular Evolution Group, Heidelberg Institute for Theoretical Studies, Heidelberg, Germany. [11]Institute of Computer Science, Foundation for Research and Technology Hellas, Heraklion, Greece. [12]Institute for Theoretical Informatics, Karlsruhe Institute of Technology, Karlsruhe, Germany. [13]Department of Ecology and Evolutionary Biology, University of Toronto, Toronto, Ontario, Canada. [14]Department of Natural History, Royal Ontario Museum, Toronto, Ontario, Canada. [15]College of Science and Engineering, Flinders University, Adelaide, South Australia, Australia. [16]Australian Museum Research Institute, Sydney, New South Wales, Australia. [17]Department of Biological Sciences and Museum of Natural Science, Louisiana State University, Baton Rouge, LA, USA. [18]Department of Biology, New Mexico State University, Las Cruces, NM, USA. [19]Department of Ornithology, American Museum of Natural History, New York, NY, USA. [20]Bioinformatics and Systems Biology Graduate Program, University of California San Diego, La Jolla, CA, USA. [21]Computer Science and Engineering, University of California San Diego, La Jolla, CA, USA. [22]College of Life Sciences, University of Chinese Academy of Sciences, Beijing, China. [23]Department of Computer Science, Rice University, Houston, TX, USA. [24]Natural History Museum Denmark, University of Copenhagen, Copenhagen, Denmark. [25]Center for Global Mountain Biodiversity, Globe Institute, University of Copenhagen, Copenhagen, Denmark. [26]Centre of Excellence for Omics-Driven Computational Biodiscovery, Faculty of Applied Sciences, AIMST University, Bedong, Malaysia. [27]Department of Life Sciences, Imperial College London, Silwood Park, Ascot, UK. [28]Milner Centre for Evolution, University of Bath, Bath, UK. [29]ELKH-DE Reproductive Strategies Research Group, University of Debrecen, Debrecen, Hungary.

[30]Center for Macroecology, Evolution, and Climate, The Globe Institute, University of Copenhagen, Copenhagen, Denmark. [31]HUN-REN-PE Evolutionary Ecology Research Group, University of Pannonia, Veszprém, Hungary. [32]Behavioural Ecology Research Group, Center for Natural Sciences, University of Pannonia, Veszprém, Hungary. [33]Bird Group, Natural History Museum, Tring, UK. [34]CIIMAR/CIMAR, Interdisciplinary Centre of Marine and Environmental Research, University of Porto, Porto, Portugal. [35]Department of Biology, Faculty of Sciences, University of Porto, Porto, Portugal. [36]NABU, Berlin, Germany. [37]Centre for Zoo and Wild Animal Health, Copenhagen Zoo, Frederiksberg, Denmark. [38]Key Laboratory of Zoological Systematics and Evolution, Institute of Zoology, Chinese Academy of Sciences, Beijing, China. [39]College of Life Science, University of Chinese Academy of Sciences, Beijing, China. [40]Institute of Ecology, Peking University, Beijing, China. [41]Danish Institute for Advanced Study, University of Southern Denmark, Odense, Denmark. [42]Department of Vertebrate Zoology, National Museum of Natural History, Smithsonian Institution, Washington, DC, USA. [43]University of Illinois Urbana-Champaign, Champaign, IL, USA. [44]Department of Biology, University of Florida, Gainesville, FL, USA. [45]University Museum, NTNU, Trondheim, Norway. [46]Vertebrate Genome Lab, The Rockefeller University, New York, NY, USA. [47]Howard Hughes Medical Institute, Durham, NC, USA. [48]University of California, San Diego, San Diego, CA, USA. [49]Villum Center for Biodiversity Genomics, Department of Biology, University of Copenhagen, Copenhagen, Denmark. ✉e-mail: josefin.stiller@bio.ku.dk; smirarabbaygi@ucsd.edu; guojiezhang@zju.edu.cn

## Methods

Further details on methods are given in Supplementary Information. No statistical methods were used to predetermine sample size. The experiments were not randomized and investigators were not blinded to allocation during experiments and outcome assessment.

### Selection of genomic regions for phylogenomic inference

For the main tree, we used putatively intergenic regions extracted from a Cactus whole-genome alignment[4,51]. We converted the HAL alignment to MAF format using chicken as the reference and extracted the best-aligned synteny blocks from each query species using 10 kb windows (https://github.com/Secretloong/Cactus_Alignments_Tools, using HALtools[52] v.2.3), skipping regions that were repetitive in chicken or those present only in Galliformes. Among the first 2 kb of each window, the 1 kb portion with the most site-wise occupancy was selected to avoid portions with few sequences. The decision to use 1 kb loci from which to estimate GTs was made following preliminary assessments (Extended Data Fig. 1d). Therefore loci were 8–9 kb apart, reducing the risk of strong linkage[53]. We excluded fragmentary sequences (under 50% of the median length of all sequences of the locus) and loci with fewer than four sequences. This resulted in 94,402 loci for which we estimated GTs. Based on the chicken genomic annotation, we identified 1 kb loci which had overlap with exons (14,355 loci) or introns (16,617 loci) and created smaller datasets without these regions (Extended Data Fig. 1b). Subtraction of these from the total loci resulted in 63,430 purely intergenic loci, which were used to construct the main tree.

We also extracted loci of other data types and applied the filtering described above. This resulted in 44,846 intronic, 14,972 exonic and 4,985 UCE loci. Introns were extracted from the Cactus alignment following previously described procedures[4], reconstructing individual GTs for each intron of the same gene. Protein-coding regions were obtained from genome annotations[4] and all exons of the same gene were analysed as one locus; these were further filtered and aligned. This was done with an iterative PASTA[54] v.1.8.5 pipeline that included Tree-Shrink[55] v.1.3.1 to remove outlier sequences, alignment with MAFFT[56] v.7.149b G-INS-i with a variable scoring matrix[57] to isolate potentially unrelated segments and removal of these blocks. We excluded third codon positions because these were previously shown to be problematic[1]. UCE loci were extracted using PHYLUCE[58] v.1.6.3 (commit 185b705) targeting 5,060 UCEs and 1,000 base pair flanking regions. After filtering, 5,006 UCE loci remained. Alignment and exclusion of outliers was conducted similar to the protein-coding regions but using MAFFT L-INS-i without removal of alignment segments.

### Generation of GTs and species trees

A total of 159,205 GTs were estimated using maximum likelihood tree inference with Pargenes[59] v.1.1.0, which uses substitution model selection through Modeltest-NG[60] v.0.1.3 and RAXML-NG[61] v.0.9.0, with ten random and ten parsimony starting trees and scaled branch lengths. To identify and collapse poorly supported branches before running ASTRAL we used IQTREE[62] v.1.6.12 to perform parametric approximate likelihood ratio tests (aLRT), which are rapid tests of the three possible nearest-neighbour resolutions around a branch[63] and are more computationally efficient than bootstrapping. Outputs from Pargenes were used for computing aLRT scores. Poorly supported branches were contracted to polytomies using newick-utilities[64] v.1.6 if their aLRT value was below 0.95.

Collapsed GTs were summarized into a coalescent-based species tree using ASTRAL-MP[65] v.5.14.5. Support was assessed using posterior probability. We also performed gene-only, multilocus bootstrapping (globalBS) for cases in which uncertainty is not local (for example, two placements many branches away both resulting in high quartet support), a scenario that can mislead local posterior probability support[66]. In addition we tested polytomy null hypotheses[35] and evaluated

the quartet score of the three alternative nearest-neighbour interchanges around each branch[66]. Quartet scores were visualized using DiscoVista[67]. We evaluated alternative species trees (for example, moving Phaethontimorphae) by scoring these trees against the same input GTs using ASTRAL.

For a concatenated analysis of the 63,430 loci under maximum likelihood we used RAXML-NG v.1.0.1, partitioning by locus (63,430 partitions) with their previously determined substitution models. We ran 20 independent searches from random starting trees and picked the highest-scoring tree. We then ran 50 tree searches on bootstrapping pseudo-replicate alignments, judged sufficient according to the extended majority rules (MRE) bootstrap convergence criterion[68]. To save time and energy we used a topological constraint for all tree searches (maximum likelihood and bootstrapping). This was a strict consensus of the ASTRAL trees (63,430 loci, exons, introns and UCEs) and of an initial maximum likelihood run on the 63,430 loci (based on ten tree searches with five random plus five parsimony starting trees, no bootstrapping). This consensus left the backbone nodes free to be inferred with constraining uncontroversial nodes within orders (317 nodes resolved, 45 collapsed).

### Fossil calibration and molecular dating

We performed molecular dating using a Bayesian sequential-subtree approach[69]. This involved using date estimates from an initial analysis of a backbone tree (56 tips) containing two representatives of each of 11 subtrees. This provided secondary calibrations for subsequent dating analyses of 11 subtrees (19–42 tips each). The subtrees were then attached to the backbone to assemble a timetree of all 363 taxa.

We performed molecular dating using a subset of the 63,430 loci. For all loci we estimated phylograms in IQTREE[70] v.2.0.4 under GTR + F + R4, fixed to the main topology and rooted with FastRoot[71]. We selected 10,494 loci with the lowest coefficient of variation in root–tip distances, thereby retaining the most clocklike loci. For locus partitioning we randomly divided loci into two groups of 5,247 within which we partitioned based on their macro-, intermediate and microchromosomal origin. The two locus groups were used for dating. Half of the loci were used to date the backbone tree and the other half to date the subtrees, thus avoiding data duplication in the likelihood.

For node-based calibrations we identified 34 clades with fossils fulfilling best-practice criteria[72] (Supplementary Information). We used CladeDate[73] to generate calibration densities empirically based on fossil occurrences (187 fossils) and estimators of distributions in which the truncation was the estimated age of the clade[23,74]. We used the Strauss and Sadler[75] estimator for uniformly distributed fossil occurrences; otherwise, we excluded the Quaternary record or used estimators that do not assume sample uniformity[73]. The resultant distributions of clade ages were used to fit Student-skew distributions to parameterize calibration priors.

The posterior distributions of the ages of the 11 nodes in the backbone tree that corresponded to the root nodes of the subtrees were fitted with skew-$t$ densities using the R function sn::st.mple v.2.0.0, under the BFGS method for parameter optimization[76]. The skew-$t$ parameters were then used to specify the prior distributions of root ages for dating analyses of the subtrees.

Bayesian molecular dating was conducted using MCMCtree[77] v.4.9h, with approximate likelihood calculation[78] and under the GTR + G model. The analyses included all calibration priors, a minimum bound on root age based on an uncontroversial neornithine fossil[79] and a soft maximum bound at 86.5 Ma. Nodes without calibrations followed a birth–death process prior[80] ($\lambda = \mu = 1$, sampling fraction $\rho = 0.1$), which gives an approximately uniform kernel. We used a relaxed clock with lognormally distributed rates across branches and a gamma-Dirichlet prior on rates across the three subsets of loci[81]. During Markov chain Monte Carlo sampling, samples were drawn every 2,500 steps over a total of $5.5 \times 10^7$ steps following $5 \times 10^6$ burn-in, run twice.

We performed four additional analyses with alternative settings (Extended Data Fig. 6): (1) uniform calibration priors with ranges spanning the 95% probability density of the original calibration prior, adding a soft maximum bound with a 2.5% tail of probability; (2) a Jurassic age bound with a relaxed maximum age bound of 201.3 Ma on the root; (3) a calibration subset of 23 calibrations that were considered to be the most reliable (Supplementary Information); and (4) a set of 10,494 loci randomly selected from the 63,430 set, split into two equal groups of 5,247 and randomly partitioned into three subsets of 1,749 loci.

## Subsetting analyses

**Taxon sampling.** To investigate the effect of sampling multiple species across orders (which represent the most contentious branches), we successively reduced taxon sampling to 50, 25, 10, … 2 or 1 species per order. We randomly selected species from the existing GTs of the 63,430 locus set, retaining all if fewer than the desired number were available. We then scored the main tree against the taxon-reduced GTs to compute the normalized quartet support for the three topologies around each branch. These analyses showed substantial impact only for Accipitriformes, in which fewer than 50 species were required to recover the main relationship. Because only Passeriformes had fewer than 50 taxa, we inferred that their sampling affected the position of Accipitriformes. To test this we removed 1, 3, … 171 of the 173 Passeriformes in random order and computed quartet scores with GTs restricted to that subset. Two replicates produced indistinguishable results.

**Data quantity.** Of the 63,430 loci included in the main analysis we randomly selected subsets of increasing numbers of GT up to maximally half of the available GTs (1,000, 2,000, … 32,000). Each subset was repeated 50 times and an ASTRAL tree was estimated for each. The subset topology was compared to the main tree by counting the number of differing branches (Robinson–Foulds distance/2) using TreeCmp[82] v.2.0 and calculating the proportion of highly supported branches (posterior probability ≥ 0.95). We recorded whether each clade of the main tree was present in subset trees and counted how many different sister groups were present across the 50 replicates of each subset. We performed the same analyses for the other data types, maximally sampling about half of the available loci. This included exons (50 times sampling 1,000, 2,000, … 8,000 GTs), introns (1,000, 2,000, … 32,000) and UCEs (1,000, 2,000). We also performed the analyses using all non-coding (80,047 windows, introns and UCEs totalling 129,878 loci) GTs (1,000, 2,000, … 64,000).

**Data type.** We compared topological differences between trees for each data type, also controlling for the number of GTs used. We subsampled loci at random (50 times). The highest number of GT subsets present across all data types was 2,000 (limited by the number of UCEs). To show the effect of increasing loci we also performed the analysis for 8,000 loci, omitting comparisons with UCEs. We calculated mean pairwise Robinson–Foulds distances between resulting species trees.

**Genomic characteristics.** For GTs we calculated taxa number, tree length, tree diameter, stemminess, clocklikeness, mean branch support and proportion of branches with aLRT above 95 and above 99. For gene alignments we calculated locus length, total coverage, number and proportion of parsimony-informative sites and mean and s.d. of GC content (with seqkit[83] v.2.2.0). We predicted the difficulty of phylogenetic estimation under maximum likelihood using Pythia[84] v.1.0.0, which estimates whether the alignment is likely to result in multiple, topologically highly distinct yet statistically indistinguishable topologies. We divided loci into four equal-sized quantiles based on their values for each metric (20,011 loci based on 80,047 loci). We then estimated an ASTRAL tree for each quantile and calculated Robinson–Foulds distances to the main tree.

**Analysis by chromosomes and chromosomal category.** We built 16 species trees from GTs of the 80,047 loci according to their chromosomal assignment in chicken, excluding small chromosomes (fewer than 1,000 GTs, chr15, chr16, upwards from chr21). We also built species trees for each of the chromosome size categories of birds[85]—that is, macrochromosomes (49,686 GTs), intermediate chromosomes (11,592), microchromosomes (12,740) and the Z chromosome (5,672). To investigate discordance within and across chromosomes we calculated Robinson–Foulds distances to the main tree for each of the collapsed GTs from the 94,402 set, normalized to the numbers of nodes in each GT. We investigated potential genomic colocalization with the s.d. of GC content, because high deviations violate common model assumptions, and with recombination rates estimated for chicken[86]. We estimated mean values using the same bins as that study[85] (approximately 500 kb).

## Phylogenetic model adequacy

We tested for excessive amounts of non-stationary base composition using Foster's posterior predictive simulations method[87], adapted to maximum likelihood using a parametric bootstrap[88]. We also tested for misleading inferences due to substitution saturation using entropy tests on parsimony-informative sites[89]. For both tests, loci were defined as having a high risk of misleading inferences under scenarios in which all simulations yielded inaccurate inferences. We built an ASTRAL tree based on all loci that passed both tests (153,789 loci remaining).

## Investigation of specific nodes

**CoalHMM.** CoalHMM was used to estimate ILS levels of two clades that were difficult to resolve in our main analyses, Rheiformes and Strigiformes + Accipitriformes. We filtered and split alignment blocks into 1 Mb chunks on which CoalHMM was run[90]. We tested potential placements of Rheiformes within Palaeognathae using one representative for each order (using the most contiguous genome) and for all chromosomes. CoalHMM was also run for potential placements of Strigiformes and Accipitriformes, using Passeriformes as the outgroup and Bucerotiformes to represent the remaining Afroaves. The best-fitting topology was chosen based on posterior probabilities. Under an ILS model and in the absence of phenomena such as ancient hybridization, the proportion of sites supporting topologies different from the species tree should be equal.

**GC content within Palaeognathae.** Because we suspected that convergent GC content between Tinamiformes and Rheiformes may affect GT estimation, we defined a measure of GC similarity (ΔGC; Supplementary Information). This should be zero under the stationary models of evolution used for phylogenetic inference. Positive values deviate from the model uniting Tinamiformes + Rheiformes and negative values have the reverse effect. For 54,651 of the 63,430 loci that had all relevant species present, we calculated ΔGC and created nine subsets of loci. We ran ASTRAL on each subset, and all of them united Tinamiformes + Rheiformes. We computed a normalized quartet score around the branch to investigate whether subsets without high ΔGC had lower quartet support for Tinamiformes + Rheiformes.

## Inference of effective population size

We compared the time tree with the coalescent unit lengths estimated by ASTRAL. For each internal branch we computed the ratio of the branch length in time units to coalescent unit length:

$$\frac{\text{time unit}}{\text{coalescent unit}} = \frac{\text{generation time} \times \text{number of generations}}{\text{number of generations}/(2N_e)}$$

$$= 2 \text{ generation time} \times N_e.$$

Higher values are indicative of higher population size ($N_e$) or longer generation time. Ignoring changes to generation time, higher

time to coalescent unit ratios can be attributed to larger $N_e$. Around the K–Pg boundary the generation times are presumed to have decreased, which makes the increases in our measured quantity indicative of even larger $N_e$ growth than what would be inferred if generation times were assumed constant. Note that summary methods such as ASTRAL are known to underestimate coalescent unit length in the presence of high GT estimation error. However, we compare branches only to each other without claiming to estimate the true $N_e$. Thus, estimation error, if it is not particularly concentrated on specific nodes, should not affect the relative values.

### Analysis of molecular evolutionary rates

Genome-wide evolutionary rates were estimated for each branch using the 63,430 loci. To minimize the estimation bias in substitution rates arising from discordance between the species tree and GTs[91], we considered only those GT branches that were concordant with the main tree[92]. Each concordant branch length was divided by the time duration of the branch from the main time tree analysis, leading to a rate estimate for each species-tree branch for each locus.

### Analysis of phylogenetic signal

Pagel's lambda ($\lambda$)[93] was measured for nine continuous morphological traits from AVONET[94] on the main tree, the topology in ref. 2 and the main tree randomly subsampled to the sample size used in ref. 2 ($n$ = 198). We also performed a comparison between trees pruned to the 124 families present in both studies. To account for the high proportion of Passeriformes in our study we also excluded all but one passerine from both trees. We calculated $\lambda$ for each trait using 100 simulations using phylolm[95]. To investigate the potential effects of an incorrect tree topology we simulated traits on the main tree under a Brownian motion model using fastBM[96] with $\lambda$ = 0.96. We then randomly changed the position of 1, 5, 10 and 20% of taxa to represent incorrect relationships, repeated each 100 times, and estimated $\lambda$. To investigate the effect of convergent evolution we randomly selected species pairs consisting of one passeriform and one non-passeriform, representing 1, 5, 10 and 20% of taxa. We gave each species pair the same trait value, repeated 100 times, and estimated $\lambda$.

### Analysis of body mass and brain size evolution

We obtained body mass data (log-transformed) for 363 species[94,97] and estimated brain size (volume of the brain case) for 228 species based on endocast volume, or back-calculated it using brain volume = brain mass/1.036 (ref. 98). We used the average of males and females or mean unsexed values when available. For brain size we used missForest[99] to impute missing values based on phylogenetic relationships. Relative brain size was calculated as the residual from a log–log phylogenetic generalized least-square regression of absolute brain size against body mass. Ancestral states of both traits were reconstructed by Evomap using a multiple-variance Brownian motion approach[100]. Variations were summarized by dividing the phylogeny into bins of 1 Ma and averaging in each over all branches.

The rates of evolution in both traits were analysed using BayesTraits[101] v.4 with variable-rates models and default priors. Each analysis ran for 110 million iterations with a burn-in of 10 million in triplicates. We used the convergence diagnostic test of coda[102] and selected the run with the highest mean marginal likelihood. We also compared the fit of three single-process models (Brownian motion, early burst and Ornstein–Uhlenbeck) using Geiger[103] v.2. To compare model fit using Akaike information criterion (AIC) (Extended Data Fig. 11e), we used the mean of the rate-scaled trees of BayesTraits and calculated the likelihood of a Brownian motion model on this tree with the same trait data[104]. To investigate whether sampling one species per family could affect ancestral reconstructions, we modified tip values to reflect the family's range in body size[94] across 100 replicates (Extended Data Fig. 11f). We also confirmed that inclusion of the imputed brain size

values did not change the shape of ancestral reconstructions (Extended Data Fig. 11g).

### Reporting summary

Further information on research design is available in the Nature Portfolio Reporting Summary linked to this article.

## Data availability

The genome assemblies analysed in this study and their whole-genome alignment were part of a previous study[4], and accession numbers are given as part of the Supplementary Data. Alignments, GTs and species trees, in addition to data files produced for their analysis and scripts to plot the figures, are available at https://doi.org/10.17894/ucph.85624f66-c8e5-4b89-8e8a-fe984ca89e4a (ref. 105). This repository also contains a file detailing contents and commands to use for individual and batch download of files. The study analysed morphological trait data from AVONET[94] (https://figshare.com/s/b990722d72a26b5bead)[106] and Dryad (https://doi.org/10.5061/dryad.fbg79cnw7)[97], recombination rates for chicken[86] and time-calibrated species trees from ref. 1 (http://gigadb.org/dataset/101041)[107] and ref. 2 (Avian-TimeTree.tre from https://zenodo.org/records/28343)[108]. Source data are provided with this paper.

## Code availability

Code used for producing the figures in this paper is available at https://doi.org/10.17894/ucph.85624f66-c8e5-4b89-8e8a-fe984ca89e4a (ref. 105). The pipeline for extraction of synteny blocks from the whole-genome alignment is available under https://github.com/Secretloong/Cactus_Alignments_Tools. The pipeline for filtering and alignment of loci is available under https://github.com/uym2/TreeShrink/tree/master/related_scripts.

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

**Acknowledgements** Foremost we thank the individuals and institutions worldwide who have collected and preserved the tissue resources used to generate the genomic sequence data analysed here[4]. This dataset includes samples obtained from 66 institutions and 56 countries, spanning the years 1982–2015. Sample contributors were either authors of the original family-phase data release[4] or acknowledged in that study according to their choosing and were invited to contribute to the development of this manuscript. We are grateful for computing resources made available through GenomeDK and to the Science Faculty at the University of Copenhagen for access to Computerome2.0. We furthermore acknowledge the Gauss Centre for Supercomputing e.V. for funding computing time on the GCS Supercomputer SuperMUC at the Leibniz Supercomputing Centre. Computations were also performed on the San Diego Supercomputer Center through the Extreme Science and Engineering Discovery Environment supported by US National Science Foundation grant no. ACI-1548562, as well as high-performance computational resources provided by Louisiana State University (http://www.hpc.lsu.edu). This work was supported by the New Cornerstone Science Foundation through the XPLORER PRIZE and other grants to G.Z. from Kunpeng Fellowship at Zhejiang Province, the Strategic Priority Research Program of the Chinese Academy of Sciences (no. XDB31020000), a Villum Investigator grant (no. 25900) and the International Partnership Programme of the Chinese Academy of Sciences (no. 152453KYSB20170002). S.F. was funded by the National Natural Science Foundation of China (grant no. 32170626). D.A.D. was supported by a European Commission Marie Sklodowska-Curie Action (no. H2020-MSCA-IF-2019-883832). The contributions of E.D.J. were supported by the Howard Hughes Medical Institute. A.K., B.M., J.H. and A.S. were financially supported by the Klaus Tschira Foundation, by DFG grant no. STA 860/6-2 and by the European Union under grant agreement no. 101087081 (Comp-Biodiv-GR). S.C. is supported by the Natural Sciences and Engineering Research Council of Canada Discovery (grant no. RGPIN-2018-06747). P.A.H. and C.R. were supported by research grant no. 25925 from Villum Fonden. J.M.T.N. is supported by an Australian Research Council Discovery Early Career Researcher Award (no. DE200101222). S.Y.W.H. was supported by the Australian Research Council (no. FT160100167). The US National Science Foundation funded B.C.F. (no. DEB-1655624) and E.L.B. (no. DEB-1655683). T.S. was funded by The Royal Society (Wolfson Merit Award no. WM170050, APEX APX\R1\191045), by the National Research, Development and Innovation Office of Hungary (ÉLVONAL KKP-126949, no. K-116310) and by the Eötvös Loránd Research Network, ELKH (Debrecen University Reproductive Strategies Research Group, no. 1102207). This work was facilitated by a workshop supported by a Carlsberg Foundation grant (no. CF19-0810) to J.S.

**Author contributions** G.Z., J.S. and S.M. conceived and designed the study. J.S., S.F., A.-A.C., I.R.-G., D.A.D., Q.F., Y.D., A.K., A.S., S.Y.W.H., B.C.F., J.H., P.A.H., M.B., U.M., G.C., R.G., C.Z., Y.X., Z.H., Z.C., Z.Y., H.A.O., L.N., B.M., R.R.d.F., M.S., A.A., E.L.B. and S.M. performed genomic analyses and phylogenetic analyses. S.C., J.M.T.N., P.H., J.C., B.L. and J.F. developed fossil-based temporal calibrations. J.A.T., T.S., J.D.K., A.L. and C.R. contributed to trait data collection. J.S., S.F., A.-A.C., I.R.-G., D.A.D., A.K., A.S., S.C., J.M.T.N., S.Y.W.H., B.C.F., P.H., J.C., J.F., P.A.H., R.R.d.F., B.P., J.A.T., T.S., A.H.R., A.L., M.S., A.A., D.T.T., M.F.B., G.R.G., M.H.S., T.W., E.L.B., M.T.P.G., E.D.J., S.M. and G.Z. contributed to data interpretation. F.L., C.R., G.R.G., M.T.P.G., E.D.J. and G.Z. initiated the B10K project. J.F. contributed the bird drawings used in the figures. J.S., S.M. and G.Z. wrote the manuscript with input from all coauthors.

**Competing interests** M.T.P.G. serves on the Science Advisory Board of Colossal Laboratories & Biosciences. All other authors declare no competing interests.

**Additional information**
**Correspondence and requests for materials** should be addressed to Josefin Stiller, Siavash Mirarab or Guojie Zhang.

**a**

| Dataset | Data type | Description | Loci | Base pairs |
|---|---|---|---|---|
| 94K | Intergenic regions | All intergenic loci | 94,402 | 94,402,000 |
| 80K | Intergenic regions | Excluding overlap with exons | 80,047 | 80,047,000 |
| 63K | Intergenic regions | Excluding overlap with exons or introns | 63,430 | 63,430,000 |
| Intron | Introns | All intronic loci | 44,846 | 136,940,000 |
| UCE | UCEs | All Ultraconserved Element (UCE) loci | 4,985 | 25,579,810 |
| Exon | Exons | All exonic loci | 14,972 | 18,975,346 |
| 128K | Total Evidence | All 63K intergenic loci + introns + UCEs + exons | 128,233 | 244,925,156 |
| 159K | Total Evidence | All 94K intergenic loci + introns + UCEs + exons | 159,205 | 275,897,156 |

**b**

| Study | Species | Loci | Base pairs | Size of alignment | Species trees |
|---|---|---|---|---|---|
| Hackett et al. 2008, Science | 169 | 19 | 0.03 Mb | 5,070,000 | 1 |
| Jarvis et al. 2014, Science | 48 | 14,446 | 41.8 Mb | 2,006,400,000 | 35 |
| Prum et al. 2015, Nature | 198 | 259 | 0.4 Mb | 79,200,000 | 12 |
| Kuhl et al. 2021, MBE | 429 | 5,127 | 2.7 Mb | 1,158,300,000 | 13 |
| This study | 363 | 159,205 | 276 Mb | 99,825,000,000 | 1435 |
| Increase from Jarvis | 7x | 10x | 6x | 50x | 41x |

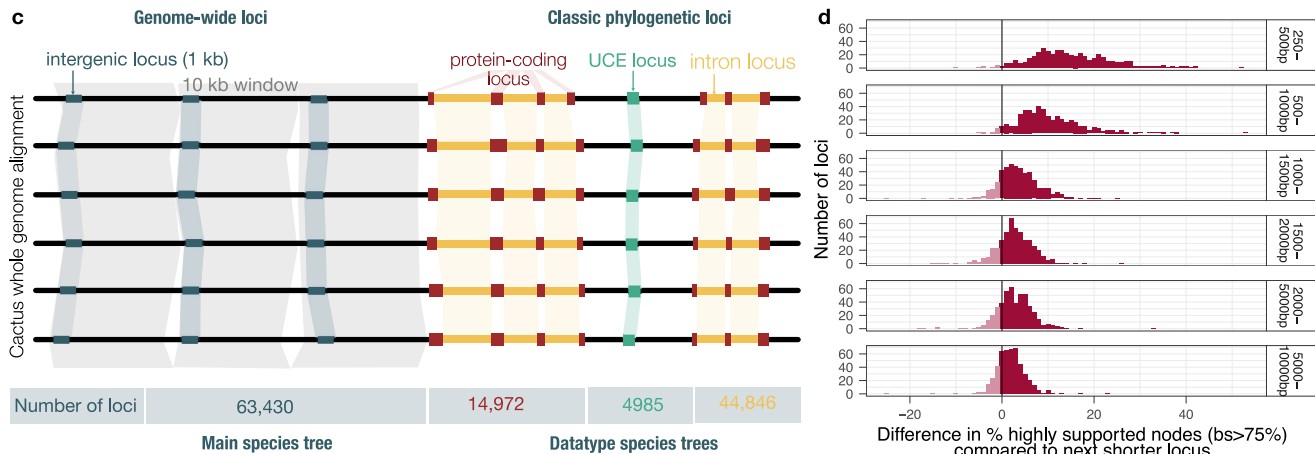

**Extended Data Fig. 1 | Overview of the phylogenomic dataset. a**, Overview of the datasets by different data types in terms of number of loci and base pairs analyzed. **b**, Comparison of dataset size to previous studies focused on avian relationships. **c**, Schematic overview of the extraction of different genomic data types (intergenic regions, exons, UCEs, introns). **d**, Choice of the length of intergenic loci. To evaluate the impact of locus length of intergenic regions, we used 500 alignments of 10 kb length and extracted subregions of increasing length (0.25 kb to 5 kb) to build gene trees for each. We then calculated the number of well-supported nodes of each locus compared to the next shorter version of the locus. We found that gene tree support increased up to 1 kb length for most loci indicating that phylogenetic signal increased. At lengths greater than 1 kb an increasing number of gene trees had fewer well-supported nodes than at shorter locus lengths (values below 0 in the plot), perhaps due to increasing propensity to include recombinations in a locus. We therefore chose 1 kb as the locus length for our analyses to balance high signal and reduced chance of recombination.

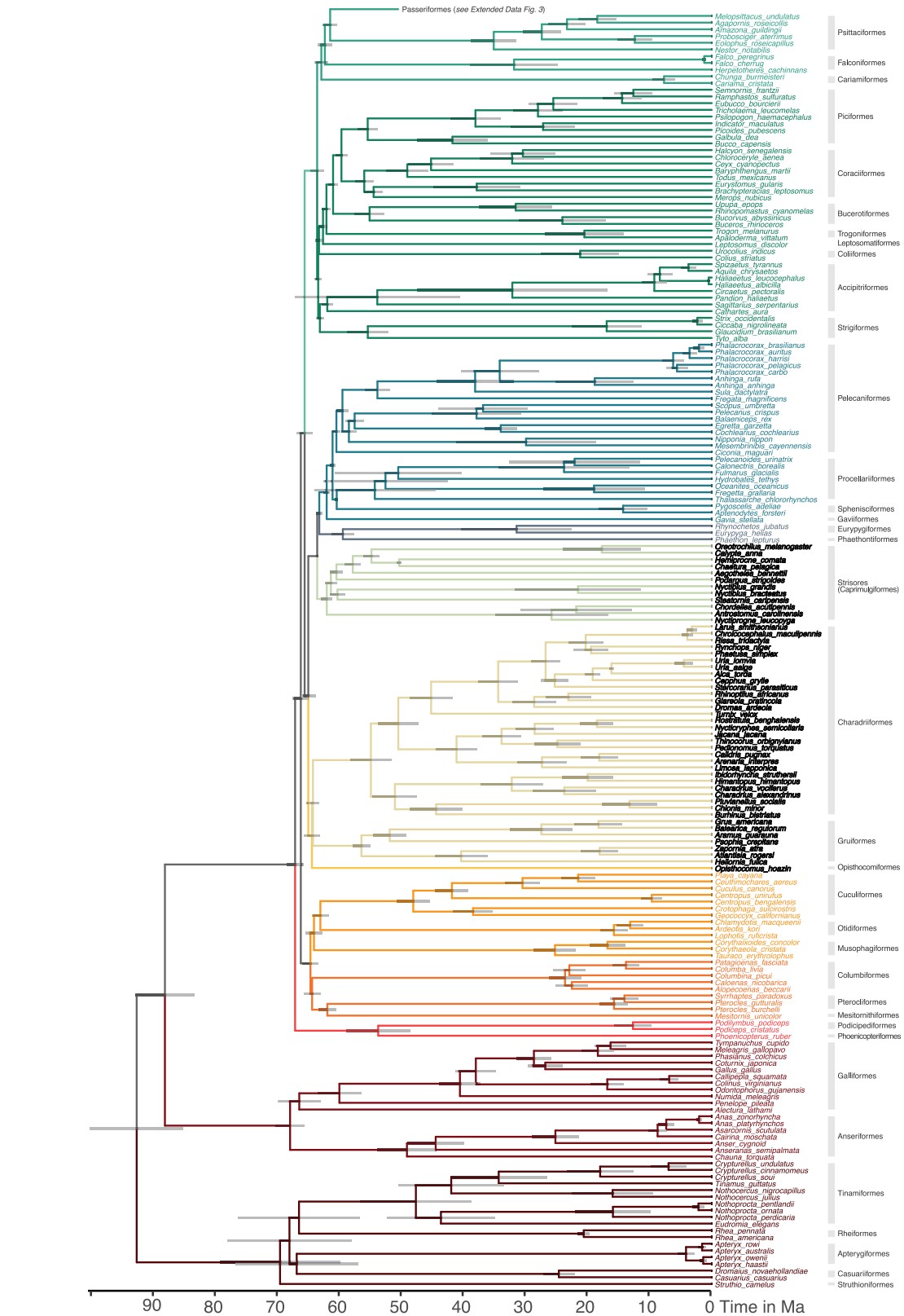

**Extended Data Fig. 2 | The main dated tree with tip labels for all groups except Passeriformes.** Taxonomic orders are annotated to the right of the tree. Colors of the branches follow those used in Fig. 1. The Passeriformes portion of the tree is shown in Extended Data Fig. 3.

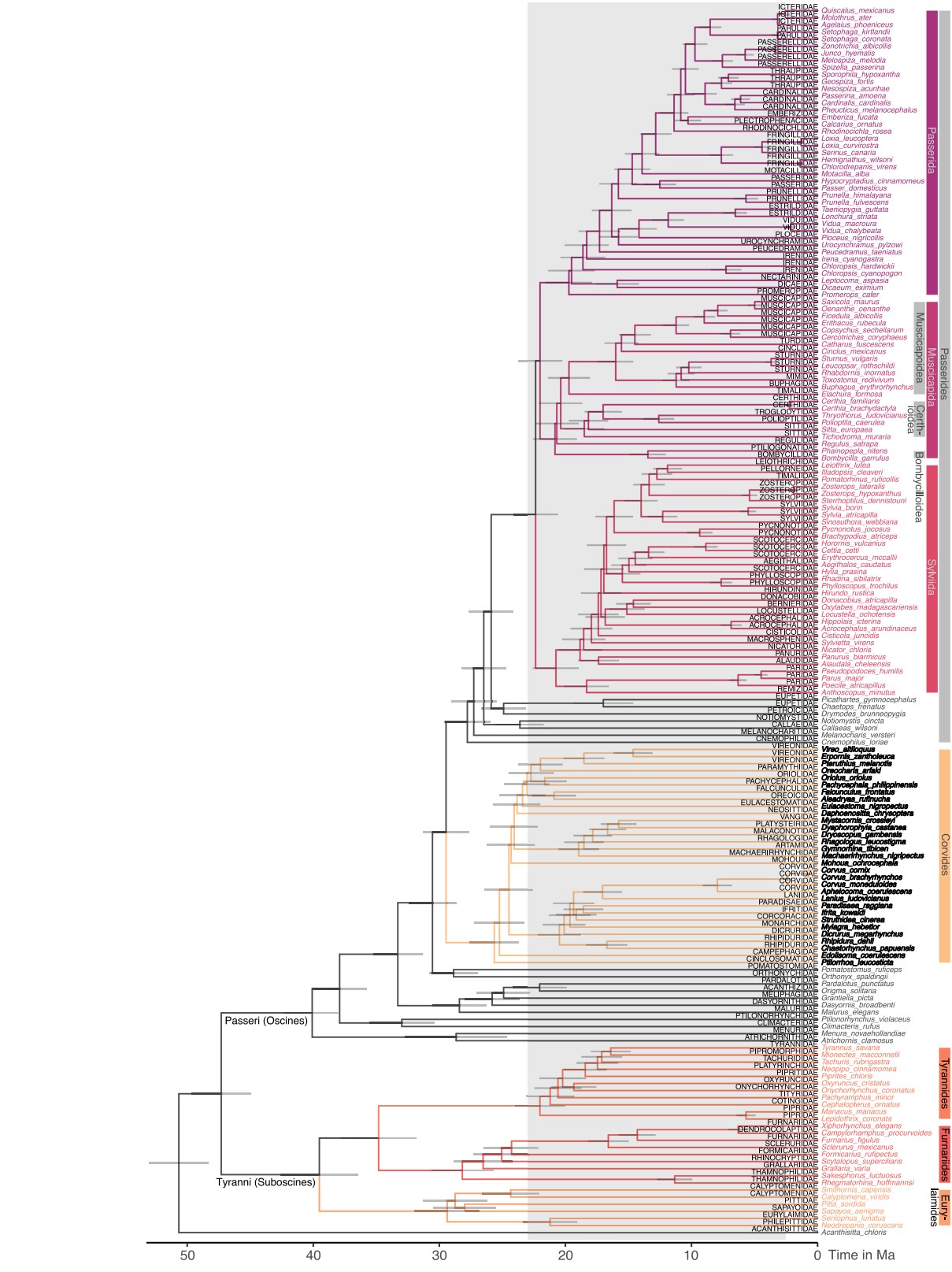

**Extended Data Fig. 3 | The main dated tree with tip labels for Passeriformes.** Taxonomic family names are given on the branches. Major clades as discussed in the text are annotated to the right following[24].

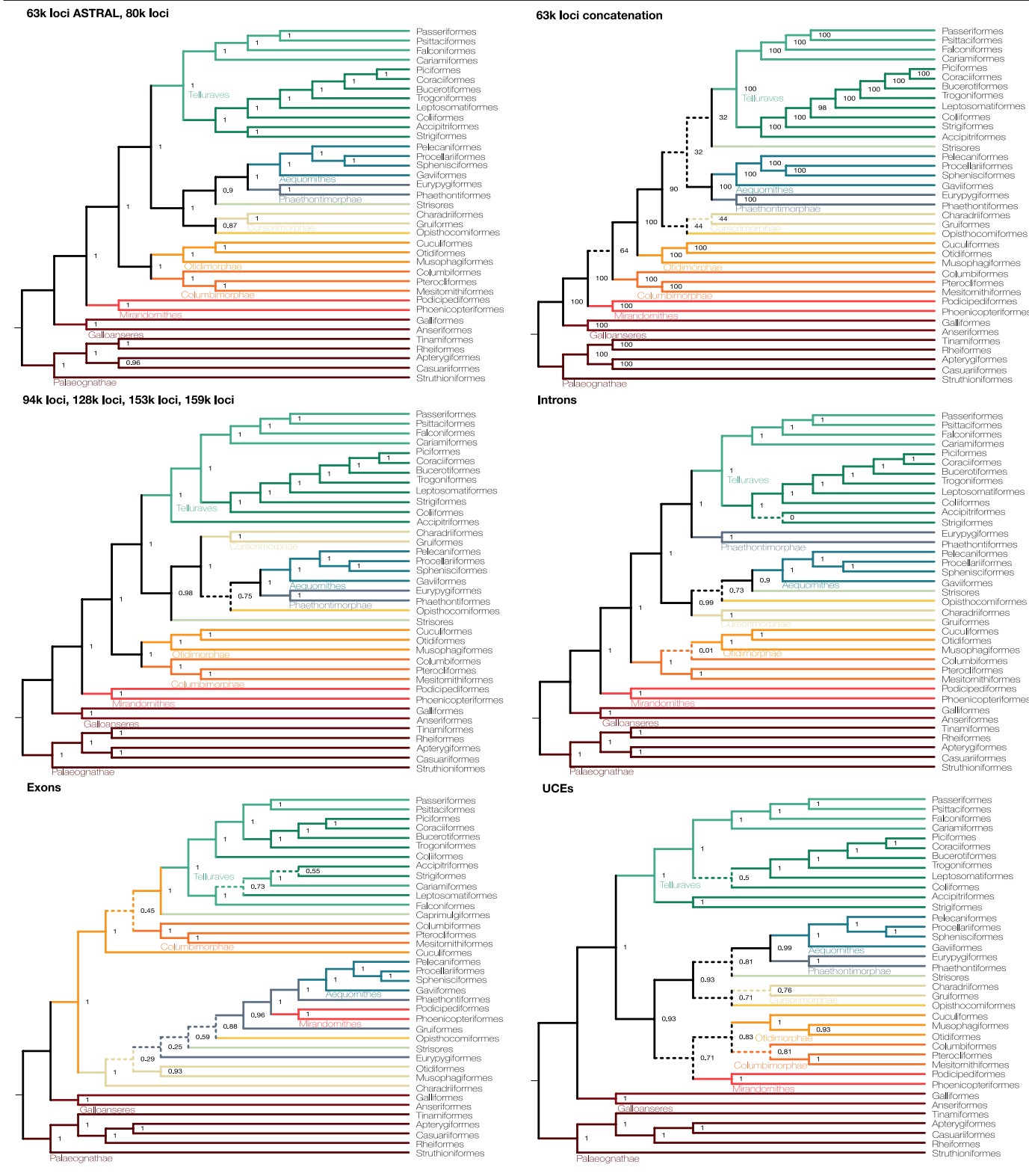

**Extended Data Fig. 4 | Overview of topologies for the species trees obtained for different data types.** Each tree is simplified to taxonomic orders, colors follow those used in Fig. 1. All analyses are coalescent-based species trees obtained from ASTRAL with support being local posterior probabilities, with the exception of the values on the panel showing the topology obtained from concatenated analysis using RAxML-NG with support values resulting from bootstrapping. Poorly supported branches (bootstrap<0.8, local posterior probabilities<0.9) are dashed.

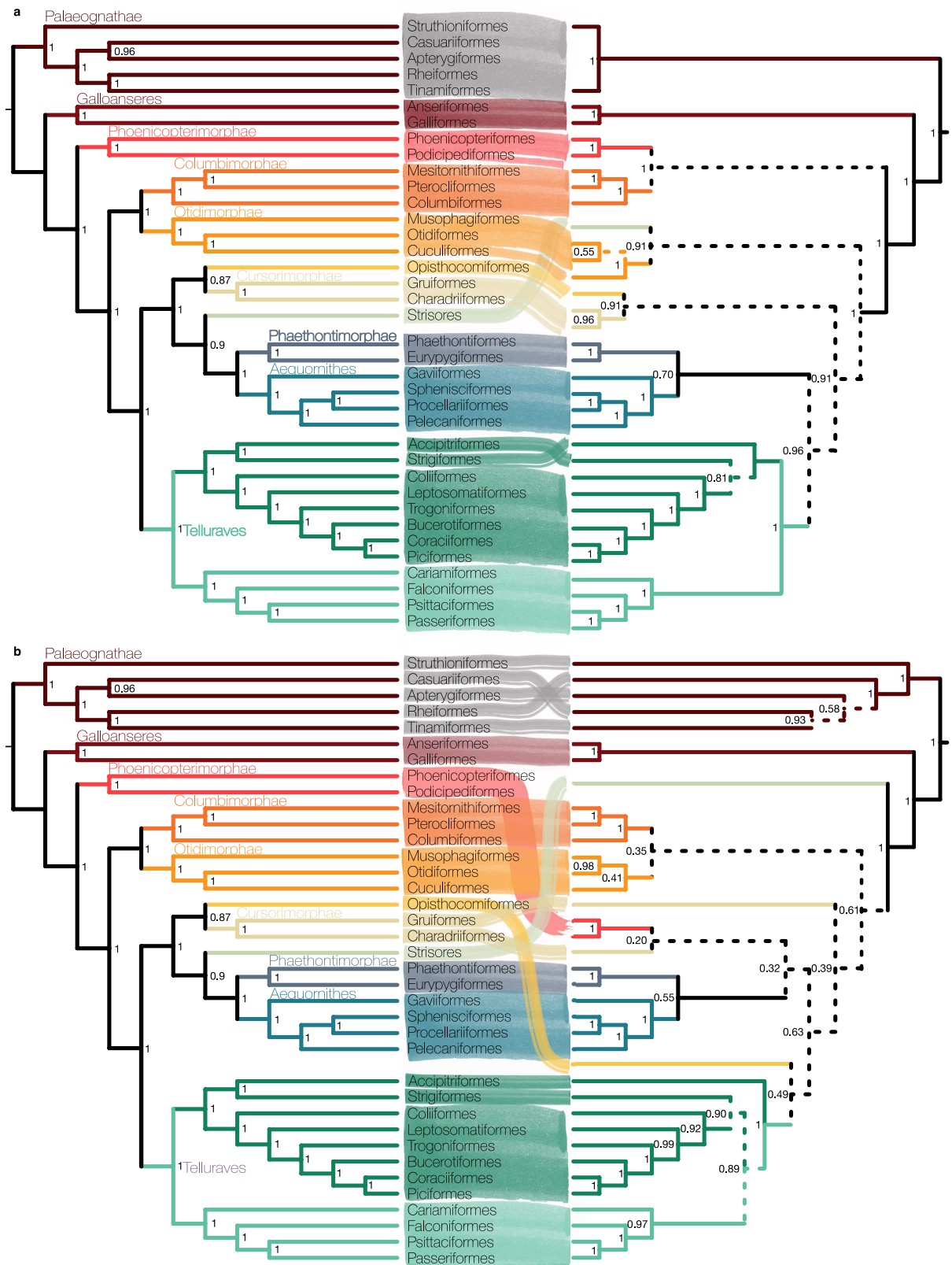

**Extended Data Fig. 5 | Comparison of the main tree with previous studies simplified to taxonomic orders.** Top, comparison to Jarvis et al.[1] 'TENT' on the right. Bottom, comparison with Prum et al.[2] on the right. Bands connect the same tips, dashed branches on the right tree indicate nodes not present in the main tree.

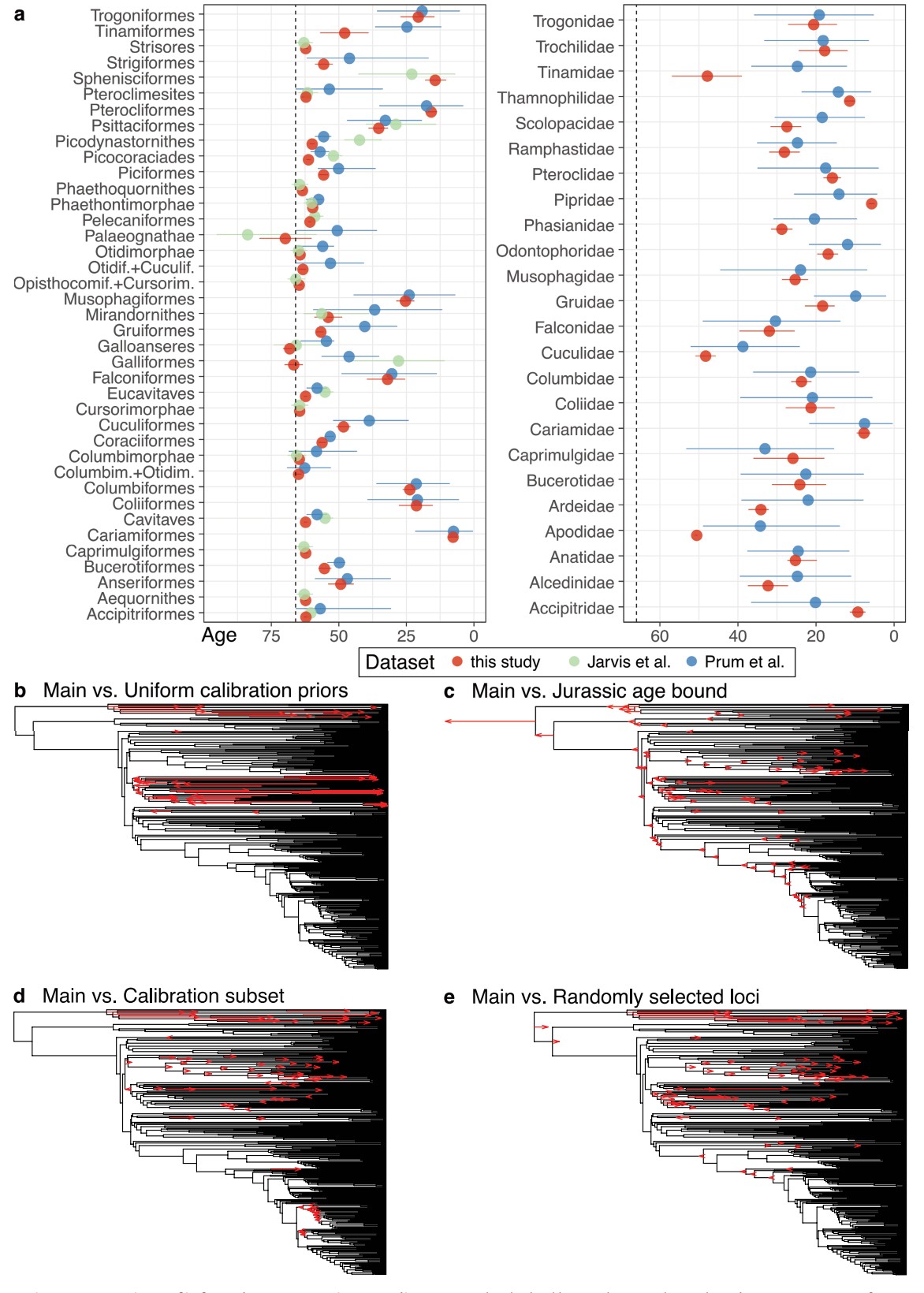

**Extended Data Fig. 6 | Comparison of inferred ages to previous studies and across alternative analyses. a**, Age estimates in comparison to previous studies for major clades and orders (left) and for families (right). Shown are median age estimates (points) and 95% credible intervals (whiskers) derived from MCMC sampling for clades that were present in at least two studies.

The dashed line is the K–Pg boundary. **b-e**, Comparison of age estimates between the main analysis and alternative analyses. Red arrows indicate the amount of displacement in the date estimates from the main analysis compared with each alternative analysis. For a description of each analysis, refer to the Methods.

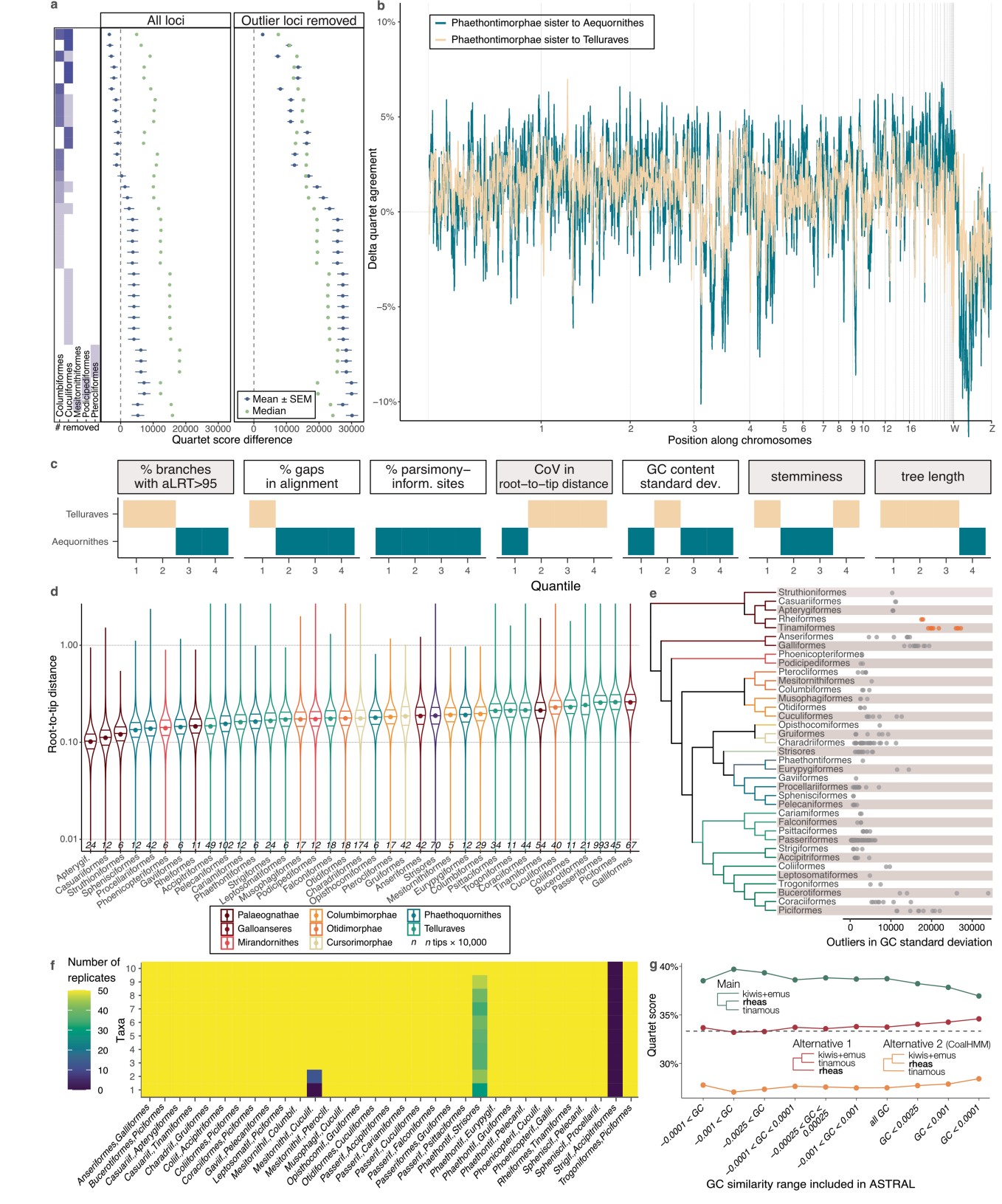

**Extended Data Fig. 7 |** See next page for caption.

**Extended Data Fig. 7 | Exploration of difficult nodes. a**, Removing species one by one from Columbea and Otidimorphae (rows, heatmap) changed the support for Columbea in the gene trees as measured by the difference between the quartet score of the tree placing Columbea or Mirandornithes at the base. Columbea was not recovered unless all but one Columbiformes or Cuculiformes was removed. Large differences between mean (blue; $n = 63,430$; shown with s.e.m.) and median (green) show the impact of outlier genes: While the mean score (akin to what is used by ASTRAL) favored Columbea in some cases, the median never favored it. **b**, Genome-wide scan for the competing topologies for Phaethontimorphae. The main (blue) and the alternative (brown) topology had a normalized quartet score difference of 0.000537%. Chromosomes with <100 windows were excluded. The $y$ axis shows the quartet support for a bipartition in each gene tree minus the mean support for that topology across all gene trees, calculated as a moving average over 100 loci. If a genomic region was strongly in favor of either topology, the two lines would be diverging, but this was not observed. **c**, The two competing positions (colors as in **b**) for Phaethontimorphae were responsive to selecting subsets of the intergenic regions that targeted long branches (panels with gray background). Species trees were generated from gene trees split into four quartiles according to their values for seven metrics. For each resulting species tree, the position of Phaethontimorphae is shown (posterior probability=1 throughout). **d**, Comparison of root-to-tip distances across 21,154,875 gene tree tips as an indicator of susceptibility to long-branch attraction. The violin plots show

distributions grouped by orders as well as mean (dots) and three quartiles (horizontal lines). **e**, Comparison of GC content outliers across birds. For each species grouped by orders, the number of loci that were outliers (defined using the interquartile range) in their GC s.d. from the remaining taxa is shown. The outliers were counted across 159,205 loci from all data types. Rheiformes and Tinamiformes had many loci with a different GC content compared to the remaining birds, which may artificially attract these two taxa. **f**, Effect of taxon sampling on topology. We sampled 1–10 taxa for each order and investigated the effect on specific nodes, given as the most recent common ancestor (MRCA) of two taxa. Colors indicate the number of replicates that recovered the clade. Most clades were supported irrespective of the number of taxa sampled (yellow), while Columbaves (Mesitornithiformes, Cuculiformes) was only found across all replicates when at least 3 taxa were sampled per order. The MRCA of Phaethontiformes + Strisores was only found when at least 10 taxa were sampled. Strigiformes and Accipitriformes were only recovered as a clade when more than 10 taxa were sampled (discussed in the main text). **g**, GC-content similarities between Tinamiformes and Rheiformes cause topological changes in gene trees. Positive values of the relative GC similarity indicate that Tinamiformes and Rheiformes are similar to each other but not to Apterygiformes and Casuariiformes, and negative values indicate the opposite. Using this quantity, we divided loci into bins and calculated the quartet score for each bin.

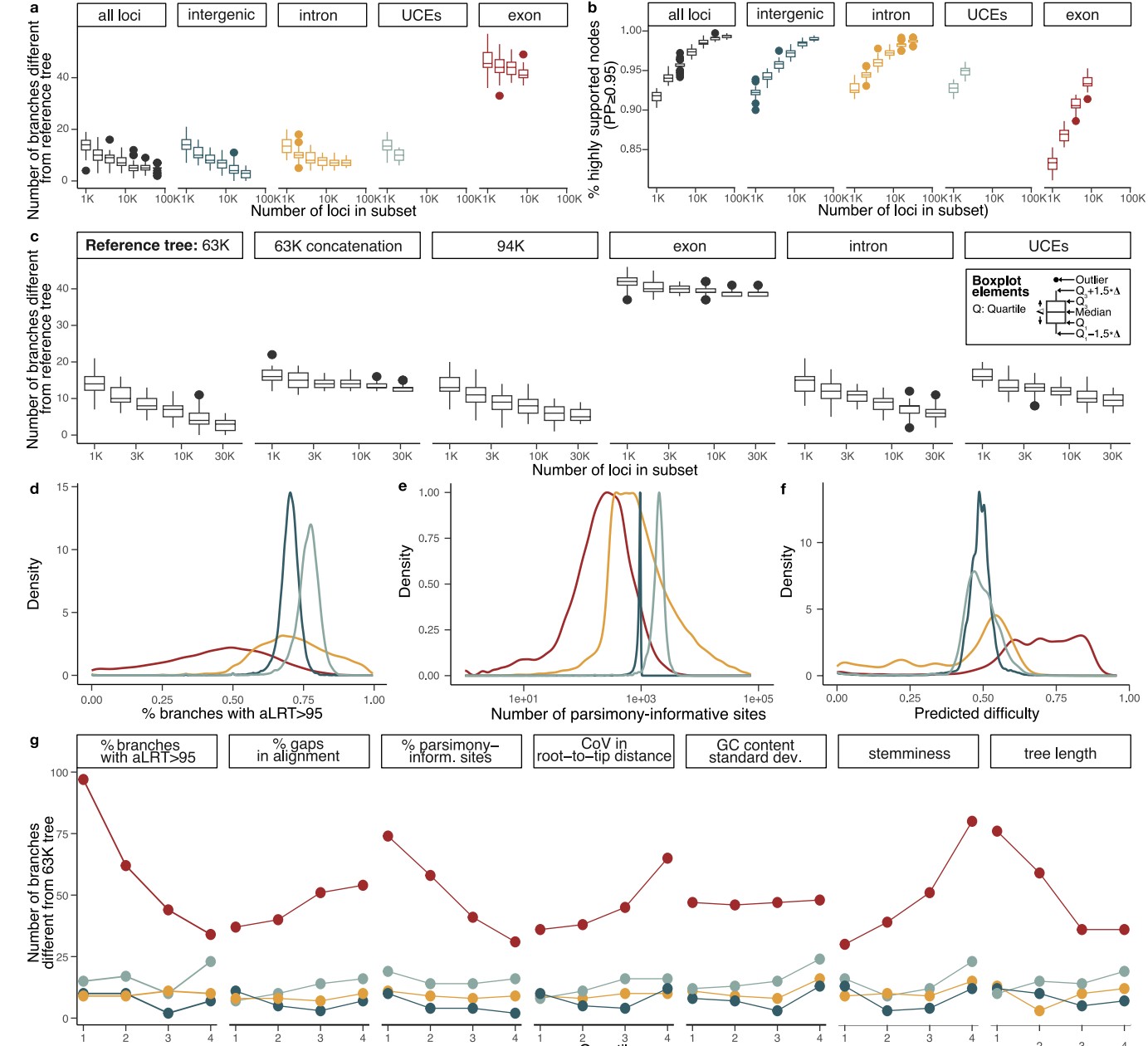

**Extended Data Fig. 8 | Comparisons between different data types.** Colors are the same for each data type across panels. In panels **a**–**c**, 50 subsets were drawn and summarized into species trees for each data type and each subset of *n* loci. Boxplot components are the same as in **c**. **a**, Greater dataset size resulted in increased similarity to the main tree across all data types. **b**, Greater dataset size resulted in an increased proportion of highly supported nodes of the resulting species tree across all data types. **c**, Response to increasing dataset size in comparison to different reference species trees. Each panel compares the same subsets of the 63,430 dataset to the reference trees (obtained from summarizing all loci of a data type), showing that increasing gene tree sampling consistently improved similarity. The increase in similarity to the species tree from concatenation and from analyzing exons is less pronounced, indicating more sustained differences despite large numbers of loci. **d-f**, Density

distribution of phylogenetic signal measured as **d**, the percentage of branches in each gene tree with more than 95% posterior probability support, **e**, the number of parsimony informative sites (PIS) in a locus, **f**, the predicted difficulty of each alignment using Pythia. Exons have the lowest signal and are more difficult. UCEs are longer than intergenic regions and thus have more PIS and slightly higher support on average, while the predicted difficulty of estimating trees for both is similar. Introns are heterogenous, ranging from easy to difficult. **g**, For each data type, loci were sorted according to their magnitude in seven metrics and split into four quantiles. The gene trees of each quantile were summarized into a species tree and compared to the main tree. Exons generally responded the strongest to subsetting, while effects were less pronounced but present in the other data types.

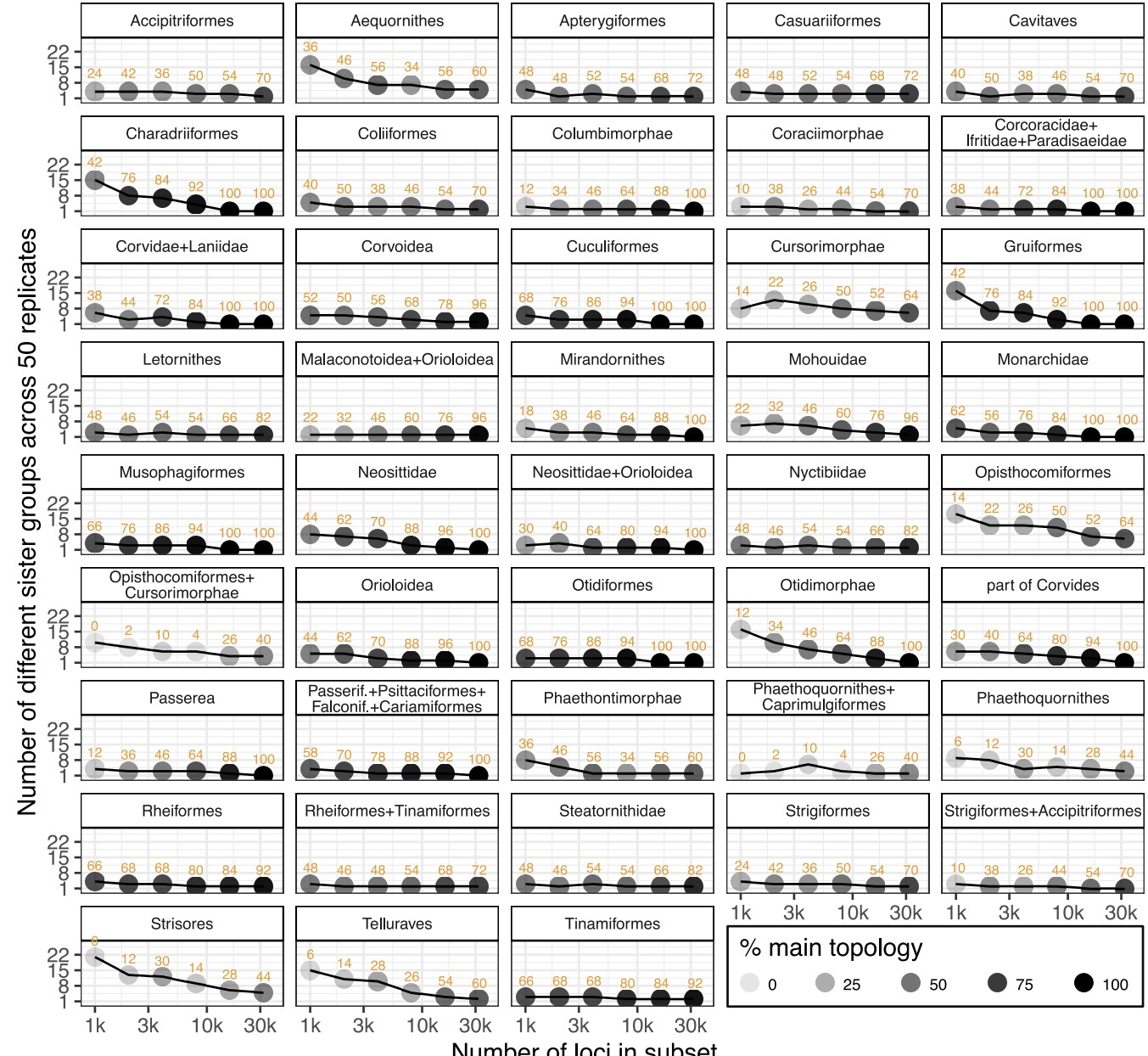

**Extended Data Fig. 9 | The number of potential sister groups decreases with increasing number of loci.** Only those nodes that still had multiple sister group proposals at 8,000 loci are shown. Points show the number of different sister group proposals obtained across 50 subsets of *n* loci. Shading of the nodes and orange numbers indicate the proportion with which the main topology was obtained.

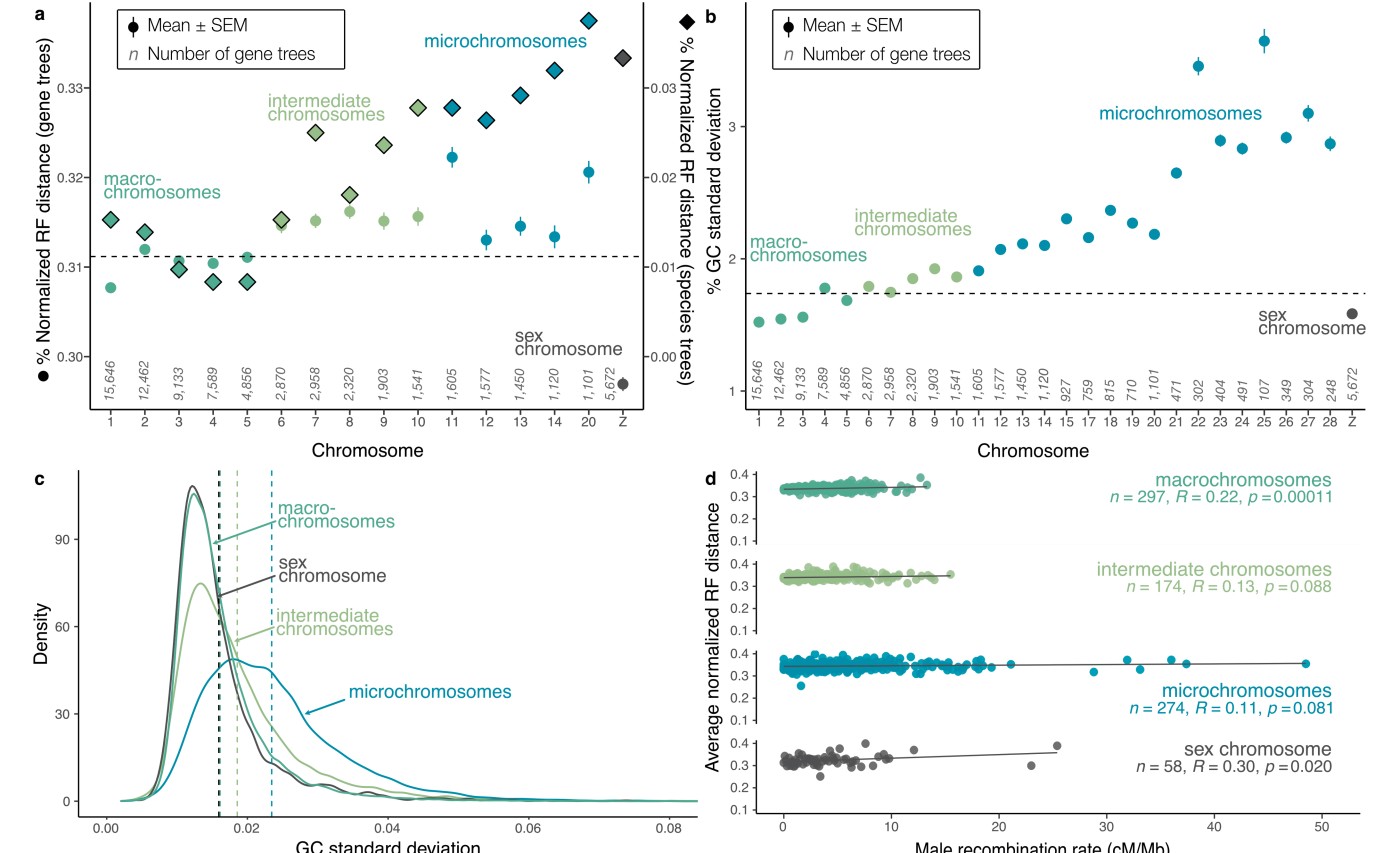

**Extended Data Fig. 10 | Comparison of different chromosomes and chromosomal categories. a**, Discordance across chromosomes. Mean ± s.e.m. of percent normalized Robinson-Foulds (RF) distance for gene trees from the 80,047 locus set derived from individual chromosomes (circles, left y-axis) and absolute RF distance to species trees (diamonds, right y-axis). Dashed line: mean gene tree distance across all chromosomes. Chromosomes with less than 1000 gene trees were not used to construct species trees. **b**, Mean ± s.e.m. of

the GC s.d. of gene trees from the 80,047 locus set for each chromosome, showing a general increase in GC s.d. in shorter chromosomes. Dashed line: mean across all chromosomes. **c**, Density plot for distribution of GC s.d. for alignments, showing higher deviation for microchromosomes. **d**, Pearson correlation of mean normalized RF distance and recombination rate for loci of different chromosome types binned over 500 kb. No adjustments for multiple comparisons were made.

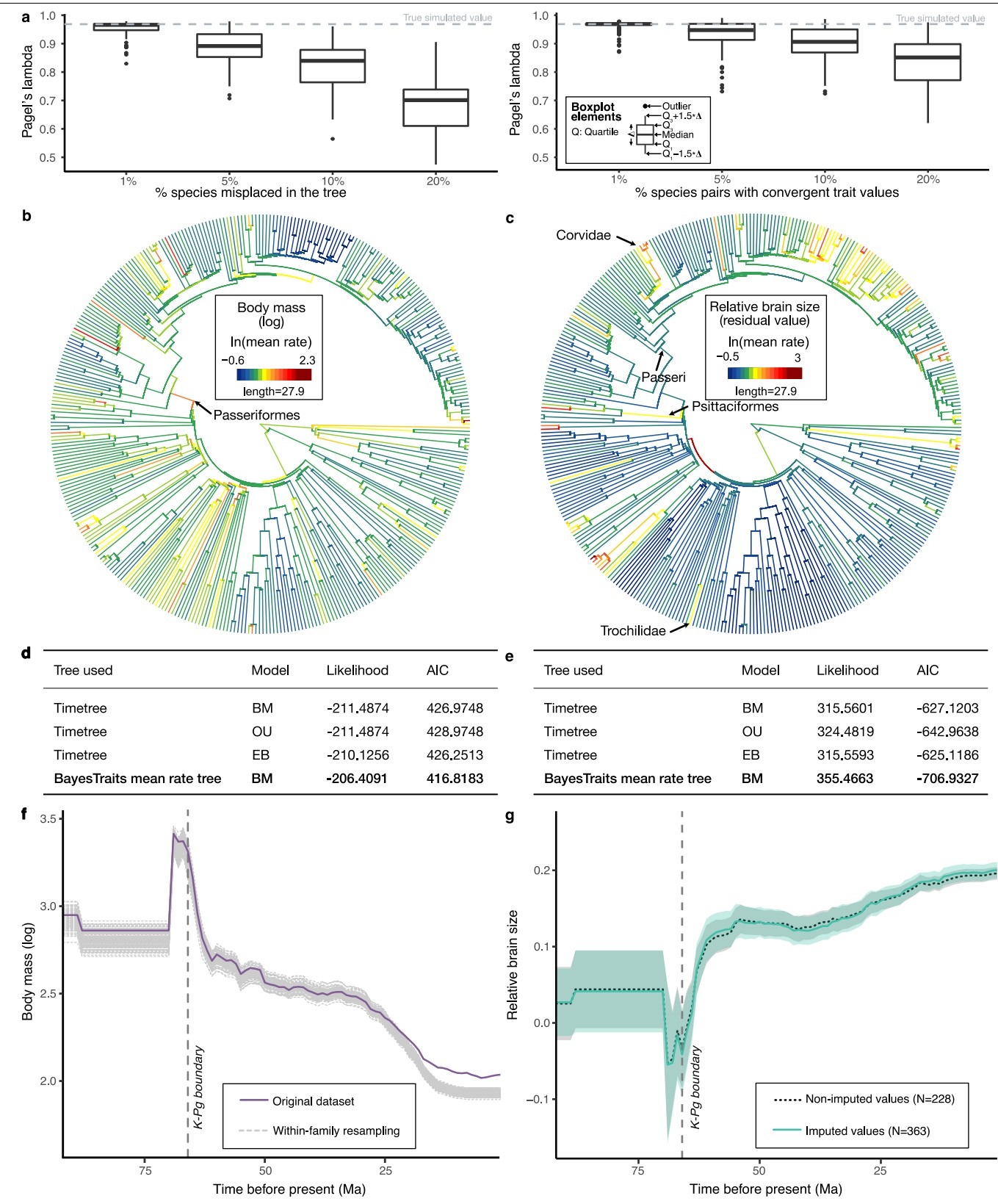

**Extended Data Fig. 11 | See next page for caption.**

Extended Data Fig. 11 | Trait evolution. a, Simulations on inferred Pagel's lambda ($\lambda$) values. To simulate topological error (left), continuous traits were simulated and an increasing proportion of species were randomly misplaced in the phylogeny ($n = 100$). To simulate the effect of convergence in trait values (right), continuous traits were simulated on a phylogeny and an increasing proportion of species pairs were randomly given the same trait value to simulate the action of convergence ($n = 100$). Compared to the effects of topological inaccuracies, the influence of convergently similar trait values on $\lambda$ estimates was weaker. b, Reconstruction of rate changes in body mass evolution (log-transformed). Branches are colored by estimates of the mean rate (log-transformed); rate changes can occur in both directions, either an increase or a decrease. c, Reconstruction of rate changes in relative brain size evolution (residual). Branch colors as in b. Taxa with pronounced rate changes as mentioned in the main text are annotated. d, Model comparisons between variable-rate and single-process models (BM: Brownian motion, EB: early burst, OU: Ornstein–Uhlenbeck) for body size. e, Model comparisons as in d for relative brain size. f, Impact of taxon sampling on ancestral reconstruction of body size. The solid purple line is the result of the ancestral reconstruction of the full dataset. The gray lines are ancestral reconstructions from analyses in which each species' trait values were randomly drawn from the range of values across their family ($n = 100$). The chosen values did not impact the reconstructions at deep timescales but estimates diverged more from 25 million years ago to the present, indicating that increased taxon sampling within families may lead to a different trajectory in more recent times. g, Impact of imputation on ancestral reconstructions of relative brain size. The non-imputed dataset contained only values based on the literature, while the imputed dataset included some values inferred using phylogenetic information. Solid lines indicate mean values and ribbons mark 95% confidence intervals. The two ancestral reconstructions are almost indistinguishable.

# Reporting Summary

## Statistics

For all statistical analyses, confirm that the following items are present in the figure legend, table legend, main text, or Methods section.

| n/a | Confirmed | |
|---|---|---|
| ☐ | ☒ | The exact sample size (*n*) for each experimental group/condition, given as a discrete number and unit of measurement |
| ☐ | ☒ | A statement on whether measurements were taken from distinct samples or whether the same sample was measured repeatedly |
| ☐ | ☒ | The statistical test(s) used AND whether they are one- or two-sided<br>*Only common tests should be described solely by name; describe more complex techniques in the Methods section.* |
| ☒ | ☐ | A description of all covariates tested |
| ☐ | ☒ | A description of any assumptions or corrections, such as tests of normality and adjustment for multiple comparisons |
| ☐ | ☒ | A full description of the statistical parameters including central tendency (e.g. means) or other basic estimates (e.g. regression coefficient) AND variation (e.g. standard deviation) or associated estimates of uncertainty (e.g. confidence intervals) |
| ☐ | ☒ | For null hypothesis testing, the test statistic (e.g. *F*, *t*, *r*) with confidence intervals, effect sizes, degrees of freedom and *P* value noted<br>*Give P values as exact values whenever suitable.* |
| ☐ | ☒ | For Bayesian analysis, information on the choice of priors and Markov chain Monte Carlo settings |
| ☒ | ☐ | For hierarchical and complex designs, identification of the appropriate level for tests and full reporting of outcomes |
| ☒ | ☐ | Estimates of effect sizes (e.g. Cohen's *d*, Pearson's *r*), indicating how they were calculated |

*Our web collection on statistics for biologists contains articles on many of the points above.*

## Software and code

Policy information about availability of computer code

| | |
|---|---|
| Data collection | All open source code and custom code used to collect the data is described in detail with versions in the methods section. Specifically, we used https://github.com/Secretloong/Cactus_Alignments_Tools, https://github.com/uym2/TreeShrink/tree/master/related_scripts, HAL v.2.3, PASTA v.1.8.5, TreeShrink v.1.3.1, MAFFT v7.149b, PHYLUCE v.1.6.3, Pargenes v.1.1.0, Modeltest-NG  v.0.1.3, RAXML-NG v.0.9.0,  RAXML-NG v.1.0.1, IQTREE v.1.6.12, IQTREE v2.0.4, newick-utilities v.1.6, ASTRAL-III v.5.14.5, FastRoot, CladeDate, MCMCtree v.4.9h, TreeCmp v.2.0, seqkit v.2.2.0, Pythia  v. 1.0.0, PhyloMAd, CoalHMM, BayesTraits v.4, . We used the following R packages and functions: sn::st.mple v.2.0.0, phylolm, fastBM, evomap, missForest. |
| Data analysis | All open source code and custom code used to analyze the data is described in detail with versions in the methods section. Specifically, we used DiscoVista and functions implemented in base R for statistical analysis. Plotting for figures was done in R with dependencies contained in the scripts deposited in the data repository at https://doi.org/10.17894/ucph.85624f66-c8e5-4b89-8e8a-fe984ca89e4a |

For manuscripts utilizing custom algorithms or software that are central to the research but not yet described in published literature, software must be made available to editors and reviewers. We strongly encourage code deposition in a community repository (e.g. GitHub). See the Nature Portfolio guidelines for submitting code & software for further information.

## Data

Policy information about availability of data

All manuscripts must include a data availability statement. This statement should provide the following information, where applicable:
  - Accession codes, unique identifiers, or web links for publicly available datasets
  - A description of any restrictions on data availability
  - For clinical datasets or third party data, please ensure that the statement adheres to our policy

The genome assemblies analyzed in this study and their whole genome alignment were part of the study by Feng et al. Nature 2020 and accession numbers are given as part of the Supplementary Data. Alignments, gene trees and species trees, in addition to data files produced for their analysis and scripts for plotting figures are available at https://doi.org/10.17894/ucph.85624f66-c8e5-4b89-8e8a-fe984ca89e4a. This repository also contains a file detailing contents and commands to use for individual and batch download of files. The study analyzed morphological trait data from AVONET (https://figshare.com/s/b990722d72a26b5bfead) and from https://doi.org/10.5061/dryad.fbg79cnw7, recombination rates for chicken (https://static-content.springer.com/esm/art%3A10.1186%2F1471-2156-11-11/MediaObjects/12863_2009_758_MOESM5_ESM.XLS), and time-calibrated species trees from Jarvis et al. Science 2014 (http://gigadb.org/dataset/101041) and Prum et al. Nature 2015 (Avian-TimeTree.tre from https://zenodo.org/records/28343).

## Research involving human participants, their data, or biological material

Policy information about studies with human participants or human data. See also policy information about sex, gender (identity/presentation), and sexual orientation and race, ethnicity and racism.

| | |
|---|---|
| Reporting on sex and gender | The study does not involve human participants or human data. |
| Reporting on race, ethnicity, or other socially relevant groupings | The study does not involve human participants or human data. |
| Population characteristics | The study does not involve human participants or human data. |
| Recruitment | The study does not involve human participants or human data. |
| Ethics oversight | The study does not involve human participants or human data. |

Note that full information on the approval of the study protocol must also be provided in the manuscript.

# Field-specific reporting

Please select the one below that is the best fit for your research. If you are not sure, read the appropriate sections before making your selection.

☐ Life sciences          ☐ Behavioural & social sciences          ☒ Ecological, evolutionary & environmental sciences

For a reference copy of the document with all sections, see nature.com/documents/nr-reporting-summary-flat.pdf

# Ecological, evolutionary & environmental sciences study design

All studies must disclose on these points even when the disclosure is negative.

| | |
|---|---|
| Study description | The study investigates phylogenetic relationships among bird species using whole genome sequences, spanning 363 species of birds. |
| Research sample | The loci for phylogenetic analysis were extracted from an existing whole genome alignment (https://doi.org/10.1038/s41586-020-2873-9) and analyzed using phylogenetic methods. |
| Sampling strategy | Sampling targeted at least one member for each taxonomic family of extant birds. |
| Data collection | We collected 159205 genetic loci from the whole genome alignment across the bird species using bioinformatic methods. For each locus we built a gene tree, which were summarized into species trees. |
| Timing and spatial scale | The data were extracted from the whole genome alignment at a single time point. |
| Data exclusions | We only excluded minimal amounts of data. We excluded fragmentary sequences, i.e. sequences shorter than 50% of the median length of all sequences of the locus because these fragmentary sequences can impact alignment accuracy and contain fewer parsimony informative sites than the remaining sequences. Secondly, we removed loci with fewer than 4 sequences as this is the minimum number of sequences needed to construct a tree. |
| Reproducibility | We performed bootstrapping to estimate statistical support on nodes of the best estimated tree. For subsetting analyses sampling a certain fraction of all gene trees, we performed 50 replicates to estimate amount of variation in the replicates. |

| Randomization | Decisions on groupings were based on bioinformatic cutoffs, therefore randomization was not relevant. |
| Blinding | Decisions on groupings were based on bioinformatic cutoffs, therefore blinding was not relevant. |

Did the study involve field work? ☐ Yes ☒ No

# Reporting for specific materials, systems and methods

We require information from authors about some types of materials, experimental systems and methods used in many studies. Here, indicate whether each material, system or method listed is relevant to your study. If you are not sure if a list item applies to your research, read the appropriate section before selecting a response.

## Materials & experimental systems

| n/a | Involved in the study |
|-----|----------------------|
| ☒ ☐ | Antibodies |
| ☒ ☐ | Eukaryotic cell lines |
| ☒ ☐ | Palaeontology and archaeology |
| ☒ ☐ | Animals and other organisms |
| ☒ ☐ | Clinical data |
| ☒ ☐ | Dual use research of concern |
| ☒ ☐ | Plants |

## Methods

| n/a | Involved in the study |
|-----|----------------------|
| ☒ ☐ | ChIP-seq |
| ☒ ☐ | Flow cytometry |
| ☒ ☐ | MRI-based neuroimaging |

## Plants

| Seed stocks | Not applicable. |
| Novel plant genotypes | Not applicable. |
| Authentication | Not applicable. |

