## [Peer Review File · Nature]

Manuscript Title: Complexity of avian evolution revealed by family-level genomes

Reviewer Comments & Author Rebuttals

Reviewer Reports on the Initial Version:

Referees' comments:

Referee #1 (Remarks to the Author):

This paper is the next in a series of high impact submissions from the B10K group. I was pleased to see that this paper is reporting a comprehensive set of complex phylogenetic analyses that incorporate a substantial amount of data from the hundreds of genomes produced by this group. I will also note that the list of authors, although a long way from gender parity, includes a minimal 25% female scientists (including the 2 lead authors). Many large-scale collaborations tout the number of people in their networks but are often extremely male and Western dominated, which further perpetuates the disparity in publications between these groups and female and Global South scientists. I urge the B10K group to continue to consider the gaps in gender, diversity, and career status and look beyond their immediate colleagues to expand their network. Additionally, how are all the people across the globe who contributed rare samples for this analysis to be made possible being acknowledged or supported in this paper? There is no mention of the valuable resources (tissue collections in natural history museums) or people who helped acquire samples from across the world in this paper. The B10K group likes to promote itself as doing a service to the community but it is not doing its part towards uplifting members of its community or building a stronger community by inviting a larger group to participate.

I agreed to review this manuscript because it met my minimum standard for the inclusion of female authors. So although I made the important points about the author team that produced this paper and hope they will acknowledge and work towards addressing these issues, I will focus on the scientific merits of the paper in the rest of my review.

This manuscript is a remarkable feat incorporating several analyses. They managed to align and extract a large number of homologous loci across the genome – more than any other study for birds – and for nearly all families across the avian radiation. For the most part, I believe the choice of methods and their interpretation is logical. This paper merits publication given the size of the dataset and the novel strategies taken to conduct their phylogenetic analyses. Nevertheless, the resulting phylogeny is the strongest to date, not because it uncovered any novel relationships but because it has the most data. The authors are trying hard to convince the reader of the robustness of their phylogeny by doing so many analyses. However, what is most remarkable is how much discordance still exists and is genome-wide for many early branching events. This is a key pattern in birds that has been highlighted across many studies and should be acknowledged more clearly in this paper. Otherwise it becomes yet another bird phylogenomic analysis bloviating about seemingly robust relationships, which may later change in subsequent papers as small amounts of data can lead to substantial topological changes in the extremely short branches at the base of Neoaves.

There are a number of details that need to be addressed to improve the manuscript and make it more understandable to a wider audience.

First, the explanation of focusing on loci with minimal model violation is obviously a good ideal. In this study, the authors chose a strategy of extracting intergenic regions on the expectation that these are less subject to issues that lead to systematic biases. Their strategy for extracting regularly spaced regions of the genome seems well rationalized. But the authors do not really test their assumption that intergenic regions are better fit to model assumptions. The results of their analyses of phylogenetic signal and discordance imply that for the most part intergenic regions do perform better but there is still variation within this class of data that hasn't fully been explored in this study and seems to be brushed over. I would have expected that at the minimum the authors could explore statistics like base composition or heterogeneity biases with the class of data defined as intergenic to make their case.

Second, the dating analysis is presented as strongly supporting Neoaves divergences being post-K-Pg. However, it seems a bit strange to me that the basal nodes in Neoaves have very tight CIs in the dating analysis while the intermediary nodes (between 50-20 Ma) are substantially wider. This seems to be a function of branch length rather than the uncertainty of the rate estimates. Also, this is not the typical pattern based on other published dating analyses which often show that nodes deeper in time (and distant from fossil calibration points) tend to have wider CIs, as estimates are more uncertain, than more recent divergences. Can the authors please explain this result?

Third, the issue of data type is touched on many times in this analysis and the main discussion of this in the text (lines 392-430) seems to indicate that even with a few loci of highly discordance signals, relationships within the bird tree can easily change. To me this indicates the precarious nature of some of the relationships uncovered by this study. Also, I feel like using the categories of data type might be somewhat simplistic as each of these (except maybe UCEs) exhibit a high amount of heterogeneity (or variance in terms of model fit). Going back to point one, introns seem to behave very similarly to intergenic regions (Fig.4a) so why did the authors feel the need to exclude them and focus on the 63K dataset rather than the 94K or a version of 94K that excludes highly biased loci?

Fourth, all figures are data and analysis heavy, yet can be quite hard to view or interpret. I would strongly recommend that the authors review their main points and limit the figures to 2-3 subsections rather than the 4-6 currently shown (and many with subsections with multiple parts). In almost all cases, the font is way too small to read without zooming in and the colors were not always explained. Below is a list of suggestions to improve figures or their interpretations:

Fig 2a -> the terms easy, medium, difficulty and "node difficulty" is really in reference to this particular dataset/analysis. I would recommend using terms that could be more informative into future analyses and comparable with other studies, like "levels of discordance" low/medium/high or something similar.

Fig.2b – This explanation of potential long-branch attraction is weak. The branch lengths of Phaethontiformes are no more similar to Telluraves than they are to others in Elementaves. I would recommend either adding more statistical evidence (actually testing for LBA) or removing this section. Also the colors of the bar plots were not explained.

Fig.2c – I really didn't understand what the numbers and text were referring to here. Also, was there

a test to show that gene tree frequency was significantly lower in the 3rd topology?

Fig.2d – why do the y-axis numbers go above 1.0? If the high value for the main topology in the 5th bin is interpreted as misleading, why is the low value for CoalHMM not misleading? In the text, the authors rationalize that the main topology support for Rheiformes as the sister to Tinamiformes is misled by G-C bias and favor the CoalHMM placement of Rheiformes as the sister to Apterygiformes+Casuariiformes instead. But shouldn't the low value in the 5th bin of this figure also be interpreted as bias? Also, is the yellow points/topology the concatenated result? Color legend is unclear. The yellow topology is least biased according to the logic presented here but was not discussed in the text. Finally, it seems like a very low number of loci that fail the G-C bias test; why not exclude these loci and re-run the analyses?

Fig.3a—This figure is interesting for pointing out that gene/species tree methods are also subject to additive effects similar to the hidden support/conflict idea in concatenated analyses. It is interesting that all subsets are relatively stable in signal until the 50 taxa mark. This definitely needs to be explored more – what is causing this? The difference between concatenated and species tree approaches is not discussed much in this paper but I think this figure points to why it would be worthwhile to inform phylogenetic methods in general.

Fig.3c – the branches are too small or thin to see the color differences

Fig.4a – colors are not explained

Fig. 4c – it would nice to see a similar figure specifically for the 63K and 94K loci?

In summary, there are several issues that the authors still need to address in this manuscript to make it clearer for readers. Most substantially, I would urge the authors that instead of focusing on a few novel, yet still tenuous, relationships in this analysis, they also highlight the extreme discordance at the early evolution of Neoaves. This is still one of the most remarkable patterns across the tree of life and one that doesn't seem to be resolvable even with enormous amounts of data using the standard phylogenetic tools.

Referee #2 (Remarks to the Author):

The manuscript by Stiller et al. examines the family-level relationships among birds using an impressive number of loci from 363 genomes representing (to a large extent) the diversity of bird taxonomic families. This large increase, relative to previous studies, in taxon and gene sampling enabled the authors to reconstruct a relatively stable phylogeny and timetree and evaluate the reasons for the conflicts observed in some parts of the bird phylogeny. The dataset is impressive (for vertebrates), the analyses generally well done, and the manuscript is clearly written and presented. I have a few items that I wish that the authors consider:

Lines 133-142: the authors propose the erection of a new clade named Elementaves but it was not clear to me what the strength of evidence for this clade was and why it was not recovered in previous studies.

The authors make a big deal on how exons were phylogenetic markers of lower quality than intergenic regions. While I see from the data presented that the phylogenetic information contained in exon markers differs from that of other regions, it was not as clear to me why was of poorer

quality. Take for example figure 4, panels A and B. The PCA in panel A simply shows that the information present in exons is different from that in other markers (the same can be said for UCEs), but the legend states that "protein-coding regions hold LESS CONSISTENT phylogenetic signal than non-coding regions" - where is the evidence that it is less consistent? I simply do not see it. In Panel B, the authors show that the number of branches different from their main (63K) tree is higher for protein-coding regions than for non-coding ones and interpret this as evidence that protein-coding regions are markers of lower quality (see lines 404-416). But their 63K tree is made of intergenic regions so the comparison simply shows that the trees inferred from protein-coding regions are DIFFERENT from those of non-coding regions (rather than the authors' inference that intergenic region trees are BETTER than protein-coding region trees). The only evidence I saw in the manuscript is that protein-coding regions contain fewer parsimony-informative sites than non-coding regions. However, this is not a measure of quality per se.

Lines 470-472 and Figure 4a: the authors finds that the fit of the new topology to morphological trait evolution is higher than that of a previous phylogeny from Prum et al. This is an interesting finding but I am not sure what the authors interpret this to mean. One explanation would be that the new phylogeny is more accurate and better captures trait evolution. The alternative hypothesis is that the tighter association of trait evolution and the new phylogeny signifies that the new phylogeny is less accurate (phylogenies of morphological data have time and again failed us because of convergent evolution). It would be good to discuss the implications of this given how widespread convergent evolution is for the traits examined.

The manuscript is peppered with a variety of subjective terms (e.g., "INCREDIBLY high number of loci" - really? "exons are not ideal phylogenetic markers" - is any phylogenetic marker ideal?, etc.) that were somewhat distracting. The authors should remove these terms and let the data speak for themselves.

Referee #3 (Remarks to the Author):

The study by Josefin Stiller and co-workers is an impressive piece of research, and their time-calibrated family-level phylogenetic hypothesis of birds will set the standard in bird phylogenomics for the next while, for example, as backbone for future species-level phylogenetic analyses. About ten years after some of the same authors presented a first genome-scale phylogenetic analyses of modern birds, it is about time for such an extended analyses based on many new genomes generated in the framework of the B10K initiative.

The group has put in quite a bit of effort into resolving hitherto doubtful branches in the bird tree of life. Their strategy of using more than 60000 intergenic loci resulted in a robust phylogenetic hypothesis, supporting - among other findings - the divergence of Neoaves after the K-Pg boundary.

Overall, the manuscript is well written. However, I feel that the authors tried to pack too many things into this manuscript. For example, while it is important to report to a general audience that the increase in genetic loci and the use of intergenic regions helped improving phylogenetic resolution in their case, I find the discussion about increasing taxon sampling versus increasing the

number (or type) of loci not very important (and would suggest moving it into the supplement). In the end, it is a matter of the research question whether or not one wants to increase taxon sampling, and in the genomic era, the number of loci is not per se a limiting factor in study design.

The authors should be consistent throughout in presenting confidence intervals when they report divergence times. It is somewhat counterintuitive that the confidence intervals around divergence events are small around the K-Pg boundary and become larger towards present times, suggesting a strong constraint in their time-calibrations around or towards this time point. Were there any cross-validations done regarding the fossil dates? Was there any reason why not using BEAST?

The section "Implications of the new time frame for avian diversification" (466-525) is weak in parts and often speculative, and some of the topics discussed in this section are not really an implication of the new time frame. There is little support provided (statistical and/or through simulations) for some of the macroevolutionary analyses (Lilliput effect, brain size evolution, body mass evolution), and it is not clear how much these analyses are influenced by the overall rather limited sample size, especially towards the tips of the tree.

While the display items are typically of high quality, I was somewhat disappointed with Fig. 1. For example, I find that the smaller panel is too close to the main figure. I also thought that it would be nice to have some silhouettes of birds, as e.g. in Extended Data Fig. 4. Perhaps the authors could come up with a better caption for Fig. 2?

Other comments/questions:

- Pagel's lambda (reported in Fig. 5): What is the interpretation of this pattern? Is the idea that a phylogeny is better if Pagel's lambda of a mapped trait gets closer to 1? But what if there is convergence then?

- Extended Data Fig. 5: Perhaps label it with "Jarvis et al." and "Prum et al." instead of "Jarvis" and "Prum"?

- Extended Data Fig. 10 and page 25: Did you also test other models than the variable-rates one? Does the variable-rates model explain the data better than e.g. a Brownian motion model? Was brain size used as volume or was it corrected for body size?

- I did not find a link to the computational scripts used for the different types of analyses and the accession numbers of the genomes used, but maybe I have overlooked this information.

Author Rebuttals to Initial Comments:

Referee expertise:

Referee #1: evolution of birds

Referee #2: phylogenomics

Referee #3: evolutionary genomics

Referees' comments:

Referee #1 (Remarks to the Author)

This paper is the next in a series of high impact submissions from the B10K group. I was pleased to see that this paper is reporting a comprehensive set of complex phylogenetic analyses that incorporate a substantial amount of data from the hundreds of genomes produced by this group. I will also note that the list of authors, although a long way from gender parity, includes a minimal 25% female scientists (including the 2 lead authors). Many large-scale collaborations tout the number of people in their networks but are often extremely male and Western dominated, which further perpetuates the disparity in publications between these groups and female and Global South scientists. I urge the B10K group to continue to consider the gaps in gender, diversity, and career status and look beyond their immediate colleagues to expand their network. Additionally, how are all the people across the globe who contributed rare samples for this analysis to be made possible being acknowledged or supported in this paper? There is no mention of the valuable resources (tissue collections in natural history museums) or people who helped acquire samples from across the world in this paper. The B10K group likes to promote itself as doing a service to the community but it is not doing its part towards uplifting members of its community or building a stronger community by inviting a larger group to participate.

>>> We could not agree more with the points made by the referee about the importance of gender balance and diversity in a large international community. The B10K is committed to closing gaps in gender, diversity, career stage and geographic representation, but this is an ongoing commitment and we can always do better. This paper has a female lead and two non-Western senior authors. The list of coauthors includes authors of all career stages (students, postdocs, early career researchers, established researchers). Our previous paper (Feng et al. 2020) that produced the genomic data credits as authors the original source of the samples. We now explicitly mention this in the revised paper, and cite the previous study for the release of the genomes. That study was also led by 4 PhD students/postdocs, 3 of whom are female.

We are always striving to give credit and recognition to the contributors to the B10K. We would like to outline the actions we are taking:

- Although ultimately we are constrained by the fact that certain established researchers have in their collections the valuable samples upon which we rely for our ongoing endeavor to sequence all bird species, we aim to place the highest emphasis on the collectors and museum personnel that curated these samples. This includes offering authorship in the publications releasing the genomes, highlighting the importance of contributing institutions and individuals in the publication, and providing detailed sampling information in the publication, Genbank submissions and our database.
- We provide annual updates on the progress to all our collaborators (providing both information on what has been generated so far and credit to the contributors).
- We naturally extend invitations to the entire community to contribute analyses.
- We have made dozens of unpublished assemblies available to researchers for their studies prior to their release, regardless of whether they have ever contributed samples or not, and only ask for an acknowledgement of the data sharing in resulting publications, illustrating the B10K's desire to facilitate studies using the genomes as widely as possible.
- Furthermore, as we recognise the need to support local research and collecting, B10K is directly providing funding to researchers in multiple countries to not only aid collecting, but additionally assist the training of our highly valued local collaborators. For example, we sponsored the PhD studies of Ms Zamekile Bhembe in Eswatini.

In light of the comments, in addition to citing the sample contributors in the Feng et al. 2020 data release, in the acknowledgements section of the current paper, we now express our appreciation for the people and institutions who have contributed samples.

Added to Acknowledgements:

“Foremost we express gratitude to the individuals and institutions worldwide who have collected and preserved the tissue resources used to generate the genomic sequence data analyzed here ⁴. This dataset includes samples obtained from 66 institutions and 56 countries, spanning the years 1982–2015. Sample contributors were either authors of the original family-phase data release 4 or acknowledged in that study according to their choosing and were invited to contribute to the development of this manuscript.”

I agreed to review this manuscript because it met my minimum standard for the inclusion of female authors. So although I made the important points about the author team that produced this paper and hope they will acknowledge and work towards addressing these issues, I will focus on the scientific merits of the paper in the rest of my review.

This manuscript is a remarkable feat incorporating several analyses. They managed to align and extract a large number of homologous loci across the genome – more than any other study for birds – and for nearly all families across the avian radiation. For the most part, I believe the choice of methods and their interpretation is logical. This paper merits publication given the size of the dataset and the novel strategies taken to conduct their phylogenetic analyses. Nevertheless, the resulting phylogeny is the strongest to date, not because it uncovered any novel relationships but because it has the most data. The authors are trying hard to convince the reader of the robustness of their phylogeny by doing so many analyses. However, what is most remarkable is how much discordance still exists and is genome-wide for many early branching events. This is a key pattern in birds that has been highlighted across many studies and should be acknowledged more clearly in this paper. Otherwise it becomes yet another bird phylogenomic analysis bloviating about seemingly robust relationships, which may later change in subsequent papers as small amounts of data can lead to substantial topological changes in the extremely short branches at the base of Neoaves.

>>> We thank the reviewer for the positive assessment of the study as a “remarkable feat”.

We have made the following changes to **highlight discordance and not overemphasize relationships**. Discordance was already emphasized in the title of the manuscript, several subsection headers, and the subject of two of the figures, and we hope that the new presentation is more balanced.

- Upon the first introduction of our results in the Summary, we now introduce discordance:

“Using intergenic regions and coalescent methods, we present a well-supported family tree but also remarkable degree of discordance across analyses.”

- We end the introduction by discussing how genome-wide data allowed us to uncover several major sources of phylogenetic incongruence and that there still remains some uncertainty about specific branches:

“Using this tree as a reference allowed us to test the impact of various factors, including adding introns and exons, describe the major sources of phylogenetic incongruence, and identify the remaining cases of uncertainty.”

- We edited the discussion of all specific branches to highlight where uncertainty remains and at least one case where we think our main tree is likely to be incorrect. These changes are further described below under the comments on Fig. 2.

- We removed explicit referrals to new relationships and left statements of improvement up to the reader’s interpretation (edits marked as strikethrough):

“Our analyses of phylogenetic relationships among Passeriformes (perching birds)

included 173 species in 121 families and revealed new relationships. We applied seven fossil calibrations that led to more precise age estimates.”

There are a number of details that need to be addressed to improve the manuscript and make it more understandable to a wider audience.

First, the explanation of focusing on loci with minimal model violation is obviously a good ideal. In this study, the authors chose a strategy of extracting intergenic regions on the expectation that these are less subject to issues that lead to systematic biases. Their strategy for extracting regularly spaced regions of the genome seems well rationalized. But the authors do not really test their assumption that intergenic regions are better fit to model assumptions. The results of their analyses of phylogenetic signal and discordance imply that for the most part intergenic regions do perform better but there is still variation within this class of data that hasn't fully been explored in this study and seems to be brushed over. I would have expected that at the minimum the authors could explore statistics like base composition or heterogeneity biases with the class of data defined as intergenic to make their case.

>>> In order to better justify the focus on intergenic regions, we reformulated the text to increase clarity and added additional evidence.

Where we first introduce the intergenic region tree, we add a statement to emphasize the theoretical expectation that these regions may better fit model assumptions based on previous evidence (addition highlighted in bold):

*“Among thousands of inferred trees, we focus on this tree because **of the a priori expectation that intergenic regions reduce systematic error due to model misspecifications**^{12,38} (further supported by findings below).”*

Regarding evidence that intergenic regions are a better fit to model assumptions:

We expanded our experimental evidence and discussion to justify the focus on intergenic regions as opposed to introns, UCEs or exons with regard to the points below.

1) **Phylogenetic informativeness:** We clarified that, among the data types considered, our choice of the intergenic regions was based on this data type having overall high values in measures of phylogenetic informativeness with low variance among them. These narrow distributions indicate that the degree of variation between intergenic loci is relatively small. This includes the proportion of highly supported branches (Extended Data Figure 7d), number of parsimony informative sites (Extended Data Figure 7e), and the Pythia difficulty scores (Extended Data Figure 7f). UCEs have somewhat higher informativeness in two metrics but have a broader

spread of values in phylogenetically informative sites and slightly broader distributions of Pythia difficulty scores. To conclude, our choice of intergenic regions was based on them having the **lowest variability of the considered data types**.

We clarify our choice of intergenic regions:

“We found that exons had less phylogenetic information and were more variable in their signal across genes than the other data types (Extended Data Fig. 7d-e). Exons also scored highest in a measure of phylogenetic estimation difficulty (Extended Data Fig. 7f), indicating that their gene trees are less reliable than those of other data types.”

2) **Model violations:** Newly added tests supported the intergenic regions as generally having a low risk of violating common assumptions of phylogenetic models.

a) Testing for **saturation** using the entropy saturation test

(<https://doi.org/10.1093/sysbio/syab075>), we found that intergenic regions had the lowest percentage of loci with high risk of saturation (0.07% of loci fail, 43/63430). This was followed by UCEs (0.34%, 17/4985) and introns (0.83%, 373/44846). Exonic regions, despite having the 3rd codon position removed throughout our analyses, had a high proportion of loci failing the test (30.83%, 4616/14972).

b) We tested for **non-stationarity of base composition** using Foster’s posterior predictive simulations test method (<https://doi.org/10.1080/10635150490445779>), adapted to maximum likelihood using parametric bootstrap (<https://doi.org/10.1093/molbev/msx092>). Overall, we found that few loci of any data type failed this test. Among these, UCEs performed best (0.02% of loci fail, 1/4984), followed by intergenic regions (0.07% of loci fail; 42/63430), and introns (0.08% of loci fail, 36/44846). Exons had more than an order of magnitude more failures, albeit still a small portion (2.45% of loci fail, 367/14972).

To conclude, the intergenic regions performed favorably in these tests for model violations with a low proportion of loci at risk of violating stationarity of base composition or saturation. These results are now added in the main text (see the next section).

c) **Recombination assumptions:** We include additional detail for the assumption that loci used for phylogenetics should be recombination free. It can be assumed that the longer the locus, the higher the probability that it contains recombination breaks. Most data types, except for the intergenic regions, have a variable length distribution with some very long loci (introns can be up to 92,882 bp long). While the loci of introns, UCEs and intergenic regions are contiguous, most exonic loci are not composed of contiguous sequence due to their exon-intron structure. This is because we joined individual exons of the same gene into one locus (otherwise, they will be too small to provide any signal to resolve gene trees). This makes the physical distance of exons

even larger and they can span up to 566 kb of the genome (mean 16964 bp, range=149-566199). This means that the average sequence lengths within an exonic loci are on average only 950 bp, the first and last exon of a locus are on average 16,964 bp apart. The controlled 1 kb length of intergenic regions therefore is the best available compromise to balance phylogenetic informativeness and the risk of recombination within a locus.

Added to Short Online Methods:

“Phylogenetic model adequacy

We tested for excessive amounts of non-stationary base-composition using Foster’s posterior predictive simulations method⁹⁶, adapted to ML using a parametric bootstrap⁹⁷. We also tested the data for misleading inferences due to substitution saturation using entropy tests on parsimony-informative sites⁹⁸. For both tests, loci were defined as having high risk of misleading inferences under simulation scenarios where all simulations yielded inaccurate inferences.”

Supplementary Methods (Supplementary File 3)

“Phylogenetic model adequacy

We tested for misleading inferences due to phylogenetic model violation. Specifically, we tested for excessive amounts of non-stationary base-composition using Foster’s posterior predictive simulations method⁴¹, adapted to maximum likelihood using a parametric bootstrap⁴². We also tested the data for misleading inferences due to substitution saturation using entropy tests on parsimony-informative sites⁴³. For both tests, we implemented thresholds of assessment based on simulations, as described in the original studies. These thresholds define loci as having high risk of misleading inferences under simulation scenarios where all simulations yielded inaccurate inferences.“

Results are described in section “Strong impacts of different types of genomic loci due to heterogeneous signal”:

“We found that data types differed in the risk of violating assumptions of phylogenetic models. A much higher proportion of loci of exons were found to be at risk of sequence saturation (30.83%) compared to the other data types (intergenic regions: 0.07%, UCEs: 0.34%, introns: 0.83%). The evidence for violating stationarity was low across datasets but highest among exons (exons: 2.45% of loci failing the test, UCEs: 0.02%, intergenic regions: 0.07%, introns: 0.08%). Moreover, because individual exons of the same gene were joined into one locus, the assumption that phylogenetic loci are recombination-free is expected to be more frequently violated by exonic regions. An exonic locus can span wide stretches of the genome because its individual exons are not contiguous (mean sequence length=16,964 bp, range=149-566199), as opposed to loci of other types of data (mean sequence length, intron: 2543 bp, UCEs: 2095 bp, intergenic regions: 897 bp). The increased length of exons increases the risk of within-locus recombination. Thus,

analyzing only intergenic regions minimizes the risk of recombination and model violations overall.”

Second, the dating analysis is presented as strongly supporting Neoaves divergences being post-K-Pg. However, it seems a bit strange to me that the basal nodes in Neoaves have very tight CIs in the dating analysis while the intermediary nodes (between 50-20 Ma) are substantially wider. This seems to be a function of branch length rather than the uncertainty of the rate estimates. Also, this is not the typical pattern based on other published dating analyses which often show that nodes deeper in time (and distant from fossil calibration points) tend to have wider CIs, as estimates are more uncertain, than more recent divergences. Can the authors please explain this result?

>>> The patterns are the outcome of using a greater density of fossil calibrations near these old nodes, and of the sequential subtree approach used. Within each subtree, there is a secondary calibration at the root but only a handful of fossil calibrations across the other nodes – so the date estimates would be more precise than we would normally expect for the deeper nodes. A similar pattern can be seen in the mammal dated tree that was inferred using a sequential subtree approach (Álvarez-Carretero et al. 2022), where the deepest nodes in each subtree have more precise date estimates than some of the shallower nodes. This result is therefore not unexpected for this method, and the method used remains the best option for molecular dating using genome-scale data. With increases in the number of genomes and with further refinement of fossil calibrations, we expect to see in future studies a corresponding improvement in the age uncertainty of nodes closer to the present.

We have added the following sentence to the section on the timetree:

“The widest 95% CI were observed for nodes positioned farthest from the calibration points, including the secondary calibrations involved in our sequential-subtree dating approach. However, the prospects for narrowing the intervals are promising through future refinement and addition of fossil-based age constraints.”

Third, the issue of data type is touched on many times in this analysis and the main discussion of this in the text (lines 392-430) seems to indicate that even with a few loci of highly discordance signals, relationships within the bird tree can easily change. To me this indicates the precarious nature of some of the relationships uncovered by this study. Also, I feel like using the categories of data type might be somewhat simplistic as each of these (except maybe UCEs) exhibit a high amount of heterogeneity (or variance in terms of model fit). Going back to point one, introns seem to behave very similarly to intergenic regions (Fig.4a) so why did the authors feel the need

to exclude them and focus on the 63K dataset rather than the 94K or a version of 94K that excludes highly biased loci?

>>> Thank you for the comments. We address the points raised individually below.

Regarding introns:

We would like to clarify which loci made up the 63k, 80k and 94k trees because our description in the previous version was confusing. The 63k, 80k and 94k tree sets resulted from the window-based scan across the entire genome and gene tree reconstruction from 1 kb windows. The 63k trees resulted from windows that were not overlapping with introns or exons (based on chicken genome annotations) and which were therefore designated as intergenic regions. The 80k trees contained the 63k intergenic region trees plus those from windows that had some overlap with introns (at least 1 bp was intronic). The 94k trees included the 80k trees plus those from windows that had some overlap with exons (at least 1 bp was exonic). We acknowledge that it was confusing to also refer to the 80k and the 94k trees as intergenic and we now reserve the term “intergenic regions” now exclusively to the 63k tree set.

We did not perform analyses where we mixed different data types (such as 63k intergenic regions + introns), except for a “total evidence” analyses that also included all data types (the 128K and 159K trees).

Comparing the 63k dataset and the 80k dataset shows that they are topologically identical (Extended Data Figure 3a), hence the inclusion of some overlap of the windows with introns does not make a topological difference. Nonetheless, we decided to be conservative and exclude these windows with intronic overlap for the main reference tree. As the results above show, introns are more frequently at risk of violating model assumptions (both in saturation and in base heterogeneity) and are more variable in their characteristics than the intergenic regions.

We now clarify early in the manuscript that we use the main reference tree to then sequentially test the impact of adding other data types:

“Using this tree as a reference allowed us to test the impact of various factors, including adding introns and exons, describe the major sources of phylogenetic incongruence, and identify the remaining cases of uncertainty.”

In order to emphasize that the exclusion of introns was a conservative decision, we also added a statement to clarify that the inclusion of some overlap with introns did not make a difference topologically (as opposed to including overlaps with exons, change highlighted in bold):

“These results indicate that the inclusion of exonic loci, even if they constitute just 10%

*of the total loci used for analysis, can impact the most unstable parts of the tree and that their careful exclusion from phylogenomic datasets is warranted. **In contrast, exclusion of introns did not make a difference topologically.***”

Regarding variability within data categories: As Extended Data Fig. 7d-f clarifies, intergenic regions have low heterogeneity in terms of their properties, and subsetting them by various properties has a relatively low impact on the results (Extended Figure 7e). This is expected as all loci have the same length and are randomly chosen. We believe the reviewer is referring to the previous Fig. 4a, where there is substantial disagreement between intergenic region trees. However, note that this figure was based on subsamples of only 2000 gene trees. Fig. 3b shows that 2000 gene trees is an insufficient number to obtain a robust phylogeny. We were forced to use 2000 gene trees here because the low number of UCEs does not allow further subsampling. To avoid similar confusion, we updated Figure 4a and also added results with 8000 gene trees sampled.

Regarding the general point of precarious nature of results: To the extent that the reviewer was concerned with highly varied results in subsets (Fig. 4a), our clarification that we used only 2000 genes in that figure should help. Beyond that, the reviewer is correct that a handful of branches are unstable and we highlight those in detail throughout the manuscript. However, the majority of the tree is stable once enough loci (only 1000 loci for 89% of the nodes, often many tens of thousands for the remaining) with low model violations are sampled. Extended Figure 7a makes this point clear.

In the light of these comments and those of another reviewer, we realized that the presentation of the subsetting analysis in Fig. 4a with a principal coordinate analysis may not have been the most effective way to visualize the data because the axes cannot be interpreted easily. We replaced this figure with a simpler version that shows absolute topological distances between data types. The data is shown as a heatmap with mean Robinson-Foulds distances between repeated subsets of 2000 or 8000 loci for each data type (new Fig. 4a shown below). The figure clearly shows that exons have the greatest RF distance, both to other data types but also when exons are compared to each other.

Replacement for Fig. 4a:

Caption for Fig. 4a:

“Protein-coding regions give more varied species trees when they are subsampled. Each heatmap cell shows the average Robinson-Foulds distance between 1250 (diagonal: 1225) pairs of species trees each built from 2000 gene trees, one from the data type specified on the row and the other on the column. The values in brackets give the same metrics for 8000 gene trees, omitting UCEs which had fewer than 8000 loci.”

Short Online Methods:

“Data type. We compared the topological differences between trees for each data type while controlling for the number of gene trees used. As before, we subsampled loci at random (50 replicates). The highest number of gene tree subsets present across all data types was 2000 (limited by the total number of UCEs). To show the impact of increasing loci, we also performed the analysis for 8000 loci, omitting comparisons with UCEs. For all resulting species trees, we calculated mean pairwise RF distances.”

Supplementary Methods (Supplementary File 3):

“By data type

In addition to investigating the effect of increasing data quantity for each data type, we also compared the magnitude of differentiation between the resulting trees for each data type. For this analysis, we controlled for the number of gene trees used. As before, we subsampled loci at random (50 replicates). The highest number of gene tree subsets present across all data types was 2000 (because the number of loci was limited by the total number of UCEs). To show the impact of increasing loci, we also performed the analysis for 8000 loci, omitting comparisons with UCEs. For all resulting species trees, we calculated mean pairwise RF distances computed using TreeCmp v.2.0³⁶.”

Fourth, all figures are data and analysis heavy, yet can be quite hard to view or interpret. I would strongly recommend that the authors review their main points and limit the figures to 2-3

subsections rather than the 4-6 currently shown (and many with subsections with multiple parts). In almost all cases, the font is way too small to read without zooming in and the colors were not always explained. Below is a list of suggestions to improve figures or their interpretations:

>>> We have revised all of the main text figures (and several Extended Data Figures) and have placed emphasis on making them easier to view and interpret. In addition to specific considerations addressed below, we have implemented the following general changes:

- We changed the font size throughout to the requirements of the journal (pt 5 to maximally 7).
- We have made sure all colors are explained either in the legend or in the caption (or both).
- Fig. 1 has been completely redesigned with two clearly separated panels and better spacing of text.
- Fig. 2 has been modified significantly (details are described below) to reduce its complexity.
- Fig. 4a has been replaced with a simpler version (as described in the previous comment).

Fig 2a -> the terms easy, medium, difficulty and “node difficulty” is really in reference to this particular dataset/analysis. I would recommend using terms that could be more informative into future analyses and comparable with other studies, like “levels of discordance” low/medium/high or something similar.

>>> As suggested, we changed the legend description to “Levels of discordance” and the scale was changed to Low, Medium and High, with the respective quartet frequencies for alternative topologies given in the legend of the panel.

Figure caption for Fig. 2a:

“a, Gene tree discordance across the backbone of the main tree. Nodes are colored and numbered according to the bar plots (to the right) showing quartet frequencies for three possible resolutions around each branch. Dashed lines are the expected quartet frequency of a polytomy.”

Fig.2b – This explanation of potential long-branch attraction is weak. The branch lengths of Phaethontiformes are no more similar to Telluraves than they are to others in Elementaves. I would recommend either adding more statistical evidence (actually testing for LBA) or removing this section. Also the colors of the bar plots were not explained.

>>> **Regarding possible long-branch attraction.**

In order to substantiate the point that Phaethontimorphae may be artificially attracted to the long-branched Telluraves, we added quantifications of branch length differences.

Phaethontimorphae had ~25% longer branches (mean=0.192, SD=0.057) than their presumed sister group Aequornithes (mean=0.156, SD=0.048), which was statistically significant (paired t-test across loci, $p < 2.2 \times 10^{-16}$). Telluraves had significantly longer branches (mean=0.257, SD=0.070) than Phaethontimorphae ($p < 2.2 \times 10^{-16}$), which may be attracting Phaethontimorphae.

It is true that Phaethontimorphae had branch length values between Telluraves and Aequornithes. But that does not exclude long branch attraction. The point we are making is that gene tree discordance levels are not ILS-like (the second and third topologies have very different frequencies). Thus, we argue that long branch attraction is distorting gene tree frequency patterns by attracting Phaethontimorphae away from its correct position in the gene tree (which could be discordant with the species tree).

Beyond the evidence from examining the branch length differences, performing explicit tests for long branch attraction is complex (<https://doi.org/10.1093/sysbio/syab001>). PMSF approaches (<https://doi.org/10.1093/sysbio/syx068>) were proposed to ameliorate LBA effects but they are only implemented for amino acid datasets and likely computationally prohibitive on a dataset of this size.

Additions to the main text in Section “Conflict in the placement of Phaethontimorphae and Strisores”:

“Alternatively, assuming that our main topology is correct, the high support for the alternative placement could be due to problems arising from long branches. Phaethontimorphae have ~25% longer terminal branches than Aequornithes (paired t-test across loci, $p < 2.2 \times 10^{-16}$), showing greater similarity than Aequornithes to Telluraves in this regard (Fig. 2b).”

Regarding colors in Fig. 2b.

The bar plots were removed from Fig. 2b because they were redundant with those shown in Fig. 2a and added unnecessary visual complexity. Instead, we add the node numbers to allow easier matching of nodes in Fig. 2b and Fig. 2a.

Fig.2c – I really didn’t understand what the numbers and text were referring to here. Also, was there a test to show that gene tree frequency was significantly lower in the 3rd topology?

>>> We apologize that this figure panel was confusing. Because it was a text-heavy figure panel, we decided to remove it entirely and instead detail the numbers in the text.

We added a chi-square test to support the observations in Fig. 2b that the gene tree frequency of the third topology was significantly lower:

Main = Alternative 1, cannot reject ($p=0.23$)

Main = Alternative 2, reject ($p<10^{-16}$)

These results have been added to the main text in section “Landbirds are confidently resolved, except owls and hawks”:

“This node also shows gene-tree-based quartet frequencies that are statistically indistinguishable for two topologies (35% vs. 34.6%, χ^2 test, $p=0.23$), while the third is significantly lower (30.5%, $p<10^{-16}$; node 26 in Fig. 2b), contradicting expectations of an ILS-only model. Since we found no evidence of long branch attraction (Extended Data Fig. 6d), the non-ILS patterns could be indicative of ancestral hybridization⁴⁹. In contrast to gene trees, direct analysis of alignment sites using CoalHMM 50 supported an ILS-like pattern, where the two alternative topologies had similar scores (31.2% vs. 29.6%).”

For consistency, we also added a chi-square test when comparing quartet frequencies for the placement of Strisores:

“Quartet frequencies did not follow an ILS-alone scenario, as moving Strisores to the base of Elementaves, had quartet frequencies similar to the main placement (node 39 in Fig. 2a, χ^2 test, quartet frequencies indistinguishable, $p=0.6155$), while the third alternative had lower frequency ($p=0.488 \times 10^{-11}$).”

Fig.2d – why do the y-axis numbers go above 1.0? If the high value for the main topology in the 5th bin is interpreted as misleading, why is the low value for CoalHMM not misleading? In the text, the authors rationalize that the main topology support for Rheiformes as the sister to Tinamiformes is misled by G-C bias and favor the CoalHMM placement of Rheiformes as the sister to Apterygiformes+Casuariiformes instead. But shouldn't the low value in the 5th bin of this figure also be interpreted as bias? Also, is the yellow points/topology the concatenated result? Color legend is unclear. The yellow topology is least biased according to the logic presented here but was not discussed in the text. Finally, it seems like a very low number of loci that fail the G-C bias test; why not exclude these loci and re-run the analyses?

>>> In the light of these useful comments, we have investigated this issue further in two regards. 1) As suggested by the reviewer, we removed the loci that were outliers of the GC similarity metric and performed species tree estimation. 2) We increased the number of bins to 10 to obtain more fine-scaled insight than with the previous 5 bins. We found that the current topology remained dominant. This was also the case when we applied a stringent threshold of excluding loci (removing up to half of the loci). We observed that omitting loci with similar GC content for Tinamiformes and Rheiformes, but not for others, tended to reduce (but not eliminate) support for this clade (see figure below, now Extended Data Figure 6g).

Our results indicate that while GC content contributes to differences, it does not fully explain the overall signal. As a result, we are not convinced anymore that GC content differences explain the entire observed inconsistencies in placing Rheiformes. We have hence decided to remove the figure (previous Fig. 2d) from the main text and instead place a modified version with the greater number of bins as Extended Data Figure 6g. We have accordingly toned down the discussion of the placement of Rheiformes, emphasizing that GC content is only part of the explanation.

The header of the section has been modified (edits shown as strikethrough):

“Rheas have conflicting placements ~~due to biased sequence composition~~”

Results:

“Outside of Neoaves, we found support for different placements of Rheiformes within Palaeognathae, a conflict previously attributed primarily to strong ILS⁵⁴. While our main topology supports Rheiformes as the sister to Tinamiformes, analysis with CoalHMM favored Rheiformes as sister to Apterygiformes+Casuariiformes (Fig. 2c), in agreement with the previous study⁵⁴. We found that both Rheiformes and Tinamiformes had a large

proportion of loci with high GC content, setting them apart from the remaining taxa (Extended Data Fig. 6e). We observed that omitting loci with similar GC content for Tinamiformes and Rheiformes, but not for others, tended to reduce (but not eliminate) support for this clade (Extended Data Fig. 6g). These results suggest that the strong support for this grouping in our main tree was enhanced by biased GC content, leaving the placement of Rheiformes as sister to Apterygiformes+Casuariiformes, as recovered by CoalHMM, plausible.”

Extended Data Figure 6g:

Caption for Extended Data Figure 6g:

“g GC-content similarities between Tinamiformes and Rheiformes cause topological changes in gene trees. Positive values of the relative GC similarity indicate that Tinamiformes and Rheiformes are similar to each other but not to Apterygiformes and Casuariiformes, and negative values indicate the opposite. Using this quantity, we divided loci into bins and calculated the quartet score for each bin.”

Short Online Methods:

“GC content differences within Palaeognathae. Because we suspected that convergent GC content between Tinamiformes and Rheiformes may impact gene tree estimation, we defined a measure of GC similarity (GC), described in Supplementary File 3. It should be zero under the stationary models of evolution used for gene tree inference. Either positive or negative values inculcate deviation from the model, with positive values uniting Tinamiformes and Rheiformes and negative values having the reverse effect. For 54651 of the 63k loci that had all relevant species present, we calculated GC and created nine subsets of loci. We ran ASTRAL on each subset, and all of them united Tinamiformes and Rheiformes. We computed a normalized quartet score around the branch to investigate whether subsets without high GC had lower quartet support for uniting Tinamiformes and Rheiformes.”

Supplementary Methods (Supplementary File 3) in section “GC content differences within Palaeognathae”:

“To interrogate the impact of convergent GC content on gene trees, we defined a measure of GC similarity. For any three groups of taxa, X, Y, and Z, we could use

$$\frac{1}{(|X|+|Y|)(|Z|)} \sum_{a \in XUY} \sum_{z \in Z} (a - z)^2 - \frac{1}{|X||Y|} \sum_{x \in X} \sum_{y \in Y} (x - y)^2$$

to measure the similarity of X and Y beyond their similarity to Z in terms of any measure, including the GC content. To use this approach, we computed GC content per species across the 63k intergenic loci, of which 54651 loci had all relevant species present. We averaged the values for each of the four clades: Tinamiformes (T), Rheiformes (R), Apterygiformes (A), and Casuariiformes (C). Then, we defined a quantity measuring how similar T and R are compared to how similar either T or R are to either A or C:

$$\Delta GC = \left((T - C)^2 + (T - A)^2 + (R - C)^2 + (R - A)^2 \right) / 4 - (T - R)^2$$

This quantity (ΔGC) should be zero under the stationary models of evolution used for gene tree inference. It will have high positive values when T and R are similar to each other but different from C and A. It will have low negative values when T and R are dissimilar compared to their similarity to A and C. Either positive or negative values inculcate deviation from the model, with positive values pulling T and R towards each other and negative values having the reverse effect.

We created subsets of loci by removing those that had ΔGC values that diverged substantially from 0. We created nine subsets of the loci: those with $\Delta GC < 0.0001$ (51731 loci), $\Delta GC < 0.001$ (62321 loci), $\Delta GC < 0.0025$ (63108 loci), $\Delta GC > -0.0001$ (38956 loci), $\Delta GC > -0.001$ (54969 loci), $\Delta GC > -0.0025$ (59867 loci), $-0.0001 < \Delta GC < 0.0001$ (27257 loci), $-0.001 < \Delta GC < 0.001$ (53860 loci), and $-0.0025 < \Delta GC < 0.0025$ (59545 loci). We ran ASTRAL on each subset, and all of them united R and T. We computed a normalized quartet score around the branch uniting R and T for each subset to investigate whether subsets with high ΔGC removed had lower quartet support for uniting R and T.”

Regarding the scale of the y-axis of previous Fig. 2d.

We note that the y-axis in the revised figure now reports a different metric. For completeness, the previous y-axis showed the mean score of each topology in a given bin divided by the mean score of the topology across all genes. So, 1 meant no difference compared to the average, >1 meant more support than average, and <1 meant less than average.

Fig.3a—This figure is interesting for pointing out that gene/species tree methods are also subject to additive effects similar to the hidden support/conflict idea in concatenated analyses. It is interesting that all subsets are relatively stable in signal until the 50 taxa mark. This definitely

needs to be explored more – what is causing this? The difference between concatenated and species tree approaches is not discussed much in this paper but I think this figure points to why it would be worthwhile to inform phylogenetic methods in general.

>>> We performed additional analyses to investigate this interesting pattern. Passeriformes is the only order where we have more than 50 terminals, therefore we assumed that the difference in topology may be due to sampling of Passeriformes. There are two hypotheses to explain the change in topology for Accipitriformes and Strigiformes: 1) The change is due to the inclusion of certain Passeriformes terminals. 2) The change is due to the number of included Passeriformes terminals.

To address this, we randomly shuffled the Passeriformes in two replicates. We then removed 1, 3, 5, 7, ..., 171 of the 173 Passeriformes according to the shuffled order and for each replicate. We then computed the quartet scores for gene trees restricted to that subset.

We found that

- The two replicates looked identical despite removing taxa in a different order.
- Contrary to hypothesis 1, the inclusion or exclusion of individual taxa was not what matters. Rather, the important fact seems to be how many of the species are included, lending support to hypothesis 2.
- When 138 Passeriformes were sampled, the main topology was obtained (now shown in Fig. 2c, see below).

While the exact reasons behind this pattern need further mathematical analyses, results suggest that Strigiformes were pulled into two different directions in the gene tree quartets. Specifically, Strigiformes were pulled towards being sister to remaining Telluraves in quartets that did not include Passeriformes and towards being sister to Accipitriformes when Passeriformes were sampled. When the number of Passeriformes exceeded 138 (which is incidentally close to 142, the number of non-Telluraves taxa in the tree), the pull towards Accipitriformes overwhelmed the other signal. These patterns underline that the impacts of taxon sampling of one group can extend to other groups and are not easily predictable.

Results are added as Fig. 2c:

Caption for Fig. 2c:

“ c, Impact of taxon sampling within Passeriformes on the relationship between Accipitriformes and Strigiformes. Passeriformes were randomly ordered and 1, 3, 5, ..., 173 taxa were included in the analyses. In each case, the quartet support for the three topologies was assessed. Dashed lines are the expected quartet frequency of a polytomy. With 138 or more Passeriformes (black arrow), the main topology received higher support, while with fewer samples an alternative topology dominated.”

Description of results in the main text:

“Additionally, we observed that the relationship between Accipitriformes and Strigiformes depended on the number of passeriform taxa sampled. The main topology was obtained only when at least 138 Passeriformes were included, whereas sampling fewer taxa of each order favored Accipitriformes as the sister to the remaining Afroaves (Fig. 2c). This case demonstrates that the impact of taxon sampling of one group can extend to others and that these sampling effects are not easily predictable”

Short Online Methods:

“These analyses showed substantial impact only for Accipitriformes, where more than 50 species were required to recover the main relationship. Since only Passeriformes had >50 taxa, we inferred that their sampling impacted the position of Accipitriformes. To test this, we randomly shuffled passeriform taxa. We then removed 1, 3, ... 171 of the 173 Passeriformes and used ASTRAL to compute the quartet scores with gene trees restricted to that subset. Two replicates produced indistinguishable results.”

Supplementary Methods (Supplementary File 3) under “Subsetting analyses by Taxon sampling”:

“ These analyses showed substantial impact only for Accipitriformes and that more than 50 species were required to recover the main clade. Since the only order with more than

50 taxa was Passeriformes, we inferred that their sampling impacted the position of Accipitriformes and Strigiformes. In order to test this hypothesis, we randomly shuffled the Passeriformes in two replicates. We then removed the first 1, 3, 5, 7, ..., 171 of the 173 Passeriformes and for each replicate, and used ASTRAL to compute the quartet scores with gene trees restricted to that subset. We drew the support for each topology for both replicates but note that the two replicates produced indistinguishable results.”

Fig.3c – the branches are too small or thin to see the color differences

>>> We have increased the branch thickness of the gray background branches (i.e. the majority of branches which are easy to resolve), from 0.4 to 0.7. For the more difficult branches highlighted in blues and purples, we increased line thickness to 1.0.

Fig.4a – colors are not explained

>>> Fig. 4a has been replaced as explained above under “Regarding the general point of precarious nature of results”.

Fig. 4c – it would nice to see a similar figure specifically for the 63K and 94K loci?

>>> We apologize that this was confusing. This figure is based on the 94K loci, averaging normalized Robinson-Foulds distances over regions of 500 kb (= ~50 gene trees). This was mentioned in the main text but is now also clarified in the caption.

Without a certain degree of binning gene trees from adjacent regions of the genome, the variation in the RF distances of individual gene trees is heterogeneous and a signal is difficult to distinguish (see plot below). We therefore consider the binned line average as shown in Fig. 4c easier to interpret.

In summary, there are several issues that the authors still need to address in this manuscript to make it clearer for readers. Most substantially, I would urge the authors that instead of focusing on a few novel, yet still tenuous, relationships in this analysis, they also highlight the extreme discordance at the early evolution of Neoaves. This is still one of the most remarkable patterns across the tree of life and one that doesn't seem to be resolvable even with enormous amounts of data using the standard phylogenetic tools.

>>>We thank the reviewer again for their constructive and helpful comments. We hope we have addressed these points above.

Referee #2 (Remarks to the Author)

The manuscript by Stiller et al. examines the family-level relationships among birds using an impressive number of loci from 363 genomes representing (to a large extent) the diversity of bird taxonomic families. This large increase, relative to previous studies, in taxon and gene sampling enabled the authors to reconstruct a relatively stable phylogeny and timetree and evaluate the reasons for the conflicts observed in some parts of the bird phylogeny. The dataset is impressive (for vertebrates), the analyses generally well done, and the manuscript is clearly written and presented. I have a few items that I wish that the authors consider:

Lines 133-142: the authors propose the erection of a new clade named Elementaves but it was not clear to me what the strength of evidence for this clade was and why it was not recovered in previous studies.

>>> **Regarding strength of evidence**

Elementaves was supported in all analyses of the intergenic regions (63k, 80k, 94k, 128k, 159k, PP=1.00), UCEs (PP=0.93) and exons (PP=0.98) in the coalescent-based framework (Extended Data Figure 3). Analyses of introns alone did not support Elementaves but instead placed Phaethontimorphae as sister to Telluraves. Concatenation of the 63k intergenic loci did not support an Elementaves clade (Strisores was supported as the sister group to Telluraves), but was compatible with it because the conflicting branch had only 32% support.

Modification of section “Genome-wide interrogation of intergenic regions reveals new relationships and improves support”:

“This clade was supported in coalescent-based analyses of the intergenic regions, UCEs, and exons, but not by the introns or in concatenated analysis of the intergenic regions (Extended Data Fig. 3).”

Regarding reasons for the clade not being recovered before

From a phenotypic perspective it is not surprising that the groups in Elementaves have not been recovered as monophyletic in the past since its members have been historically difficult to place (members of "the magnificent seven"). This clade includes 5 major groups (Aequornithes, Phaethontimorphae, Cursorimorphae, Opisthocomiformes, Strisores), and failure to unite even one of the groups with others would break the clade. In particular, Opisthocomiformes, Strisores, and Phaethontimorphae have all been difficult to place. The fact that Elementaves is recovered as monophyletic here appears to be related to the amount of data used (number of loci and perhaps taxon sampling) and the methodological approach used (random loci with coalescent-based method), which has not been attempted before.

The authors make a big deal on how exons were phylogenetic markers of lower quality than intergenic regions. While I see from the data presented that the phylogenetic information contained in exon markers differs from that of other regions, it was not as clear to me why was of poorer quality. Take for example figure 4, panels A and B. The PCA in panel A simply shows that the information present in exons is different from that in other markers (the same can be said for UCEs), but the legend states that "protein-coding regions hold LESS CONSISTENT phylogenetic signal than non-coding regions" - where is the evidence that it is less consistent? I simply do not see it. In Panel B, the authors show that the number of branches different from their main (63K) tree is higher for protein-coding regions than for non-coding ones and interpret this as evidence that protein-coding regions are markers of lower quality (see lines 404-416). But their 63K tree is made of intergenic regions so the comparison simply shows that the trees inferred from protein-coding regions are DIFFERENT from those of non-coding regions (rather than the authors' inference that intergenic region trees are BETTER than protein-coding region trees). The only evidence I saw in the manuscript is that protein-coding regions contain fewer parsimony-informative sites than non-coding regions. However, this is not a measure of quality per se.

>>> Thank you for this comment. In the revised version, we clarify our wording and provide additional evidence for differences in phylogenetic utility among different data types. We show that exons:

1. have less phylogenetic informativeness and hence higher *estimated* gene tree discordance (due to noise),
2. are less robust to both random and targeted subsampling,
3. are more prone to saturation and base heterogeneity
4. when merged together to represent a gene, sometimes span long physical distances and are less likely to be recombination free.

1. Regarding phylogenetic informativeness:

We also direct the referee to similar comments raised by Referee #1 and our responses. We have expanded on the arguments that exons are prone to misleading phylogenetic analysis, which would indicate a lower suitability for phylogenetic analysis. This includes **lower phylogenetic informativeness** of exons (as mentioned by the reviewer, Extended Data Figure 7e) and the fact that exons more frequently yield **multiple, distinct tree topologies** with the same likelihood (as measured with Pythia scores, Extended Data Figure 7f). Unlike the number of parsimony-informative sites, Pythia scores are an indicator of signal quality.

We expand upon this point in section "Strong impacts of different types of genomic loci due to heterogeneous signal":

"We found that exons had less phylogenetic information and were more variable in their signal across genes than the other data types (Extended Data Fig. 7d-e). Exons also

scored highest in a measure of phylogenetic estimation difficulty (Extended Data Fig. 7f), indicating that their gene trees are less reliable than those of other data types.”

2. Regarding sensitivity to subsampling:

Our formulation that exons are “less consistent” referred to the observation that exonic loci produced more varied species trees compared to other data types, even when controlling for the number of loci used. This could be seen in the **greater dispersion of species trees built from randomly selected exons** in the principal coordinates analysis (previous Fig. 4a). It can also be quantified in comparisons of topological distance between species trees built from subsets of each data type. We found that the average Robinson-Foulds distance between exonic species trees was 61.4 with a standard deviation of 9.32, a much higher mean and variance than when comparing the other data types (intergenic regions 25.2, SD 5.67; introns 24.9, SD 5.39; UCEs 13.9, SD 5.29). Similarly, we also found that species trees built from exons included fewer well supported branches than other data types while controlling for the number of loci used (Extended Data Figure 7b). This greater topological disparity between subsets indicates less consistency in topology in exons than in the other data types.

In the light of these comments and those of Referee #1, we realized that the presentation of the subsetting analysis in Fig. 4a with a principal coordinate analysis may not have been the most effective way to visualize the data because the axes cannot be interpreted easily. We replaced this figure with a simpler version that shows absolute topological distances between data types. The data is shown as a heatmap with mean Robinson-Foulds distances between repeated subsets of 2000 or 8000 loci for each data type (new Fig. 4a shown below). The figure clearly shows that exons have the greatest RF distances, both to other data types but also when exons are compared to each other.

Caption for Figure 4a:

“a, Protein-coding regions give more varied species trees when they are subsampled. Each heatmap cell shows the average Robinson-Foulds distance between 1250 (diagonal: 1225) pairs of species trees each built from 2000 gene trees, one from the data type specified on the row and the other on the column. The values in brackets give the same metrics for 8000 gene trees, omitting UCEs which had fewer than 8000 loci.”

We have rewritten the part on the variation due to subsampling:

“Trees inferred from exons were also more sensitive to subsampling than trees built from other data types (Extended Data Fig. 7a-c). Even when controlling for the number of gene trees used in species tree construction (N=2000), exons produced more dissimilar trees than other data types (Fig. 4a). Problems associated with analysis of exons have been extensively discussed^{1,12,13} and our results corroborate the idea that low signal and violations of sequence evolution models cause these differences.”

We agree that exons becoming more similar to the 63k tree is not a measure of their quality. Yet, Figure 4b (and Extended Data Figure 7g) shows that exons **respond much more strongly to targeted subsampling** according to different metrics than the other data types. In the most extreme case (% of branches with aLRT>95), we found that the tree from quantile 1 differed from that of quantile 4 in 63 branches (Extended Data Figure 7g, leftmost panel). While the other data types also responded to subsetting, the magnitude was much smaller. This indicates that there are qualitative differences in the exons that go beyond their phylogenetic informativeness.

We have modified the corresponding section:

“To examine if exons had misleading signal, we restricted species tree inference to gene trees with more signal, less gappy alignments, greater clocklikeness, and greater total length. Unlike intergenic regions, where subsampling did not systematically change the species trees, using more informative, less gappy, and more clocklike exons reduced the incongruence between the resulting species trees and the main tree (Fig. 4b; Extended Data Fig. 7g). Thus, exons yield phylogenetic trees that are less reliable. This conclusion is consistent with earlier analyses based on fewer genomes that concluded exons have properties that are problematic for phylogenetics^{1,12,13,38}.”

3. Regarding potential violations of phylogenetic model assumptions:

We have added analyses that aimed to identify the risk that loci may **violate common phylogenetic model assumptions** as described in our response to Reviewer 1. While risk of non-stationarity of base composition was overall low among the data types, exons fared the worst (2.45% of all loci failed compared to <0.11% of loci of other data types failing). Tests for saturation found that exons were at the highest risk of being saturated (30.83% of loci failed) despite not analyzing 3rd codon positions. Introns (0.83%), UCEs (0.34%) and particularly intergenic regions (0.07%) had much lower risks of sequence saturation. These tests highlight

that a **significant proportion of exons are at risk of violating models**, indicating that there is a qualitative difference in the exons compared to other data types. These results have been added to the main text (see next point).

4. Regarding the assumption of loci being recombination free: Another assumption of phylogenetic analysis is commonly that loci are recombination free. For each exonic locus, we merged all exons of the same gene. This was motivated by their functional coherence as a gene. As a consequence, an exonic locus may span large distances on the genome, raising the risk of spanning recombination breaks. We found that while the average length of the sequences within an exonic locus is 950 bp, the physical distance between the first and the last base is on average 16964 bp (range=149-566199 bp). This is a much greater physical distance than spanned by most other loci we analyzed. We therefore consider exons **more prone to recombination**.

These findings outlined under point 3. and 4. have been added to the main text:

“We found that data types differed in the risk of violating assumptions of phylogenetic models. A much higher proportion of loci of exons were found to be at risk of sequence saturation (30.83%) compared to the other data types (intergenic regions: 0.07%, UCEs: 0.34%, introns: 0.83%). The evidence for violating stationarity was low across datasets but highest among exons (exons: 2.45% of loci failing the test, UCEs: 0.02%, intergenic regions: 0.07%, introns: 0.08%). Moreover, because individual exons of the same gene were joined into one locus, the assumption that phylogenetic loci are recombination-free is expected to be more frequently violated by exonic regions. An exonic locus can span wide stretches of the genome because its individual exons are not contiguous on the genome (mean sequence length=16,964 bp, range=149-566199), as opposed to loci of other types of data (mean sequence length, intron: 2543 bp, UCEs: 2095 bp, intergenic regions: 897 bp). This may increase the risk of within-locus recombination. Thus, analyzing only intergenic regions minimizes the risk of recombination and model violations overall.”

Lines 470-472 and Figure 4a: the authors finds that the fit of the new topology to morphological trait evolution is higher than that of a previous phylogeny from Prum et al. This is an interesting finding but I am not sure what the authors interpret this to mean. One explanation would be that the new phylogeny is more accurate and better captures trait evolution. The alternative hypothesis is that the tighter association of trait evolution and the new phylogeny signifies that the new phylogeny is less accurate (phylogenies of morphological data have time and again failed us because of convergent evolution). It would be good to discuss the implications of this given how widespread convergent evolution is for the traits examined.

>>>Pagel's lambda is the degree to which shared history of taxa has driven trait distributions at tips. A value close to 1 indicates high phylogenetic signal suggesting that the diversification of the trait is influenced by shared evolutionary histories, while a lambda value of 0 indicates no phylogenetic influence on trait distributions.

While Pagel's lambda can be low due to convergent evolution, we use it here to indicate accuracy with the following logic. None of the morphological data mapped here onto the tree are used to infer the tree (neither in our analyses nor in Prum's). Thus, we are faced with two hypotheses: H1: Traits are convergent, but our tree shows more agreement with morphology despite never using the traits for inference just by random chance, or H2: Our tree is more accurate and the traits are not extremely convergent. We believe H2 is more likely.

In order to better disentangle these options, we performed additional experiments.

Simulation to evaluate the impact of an incorrect species tree topology. We simulated a list of continuous traits (sample size = 363) on our timetree under a Brownian motion model using fastBM. The measured Pagel's lambda of the simulated traits was 0.96. Then, we randomly changed the position of 1%, 5%, 10%, and 20% of the species on the phylogeny to represent misinferred species relationships. We repeated each analysis 100 times. After that, the Pagel's lambda values were estimated under these incorrect topologies.

The results are shown in the figure below (new Extended Data Figure 10a), with the dotted line representing the true lambda value of the simulation (0.96). The simulations showed that the higher the topology error level, the smaller the Pagel's lambda value. Even if only 1% of the species were incorrectly placed on the phylogeny, the Pagel's lambda values were significantly lower (t-test, $p=2.969 \times 10^{-7}$). These results show that the incorrect topology does decrease the Pagel's lambda value.

Caption for Extended Data Figures 10a and 10b:

“a, Simulation of the effect of topological error on inferred Pagel's lambda values. Continuous traits were simulated on a phylogeny and an increasing proportion of species were randomly misplaced in the phylogeny to simulate topological error. Incorrect inference of the phylogeny strongly reduced the inferred Pagel's lambda. b, Simulation of the effect of convergence in trait values on inferred Pagel's lambda values. Continuous traits were simulated on a phylogeny and an increasing proportion of species pairs were

randomly given the same trait value to simulate the action of convergence. Compared to the effects of topological inaccuracies, the influence of convergently similar trait values on Pagel's lambda estimates was weaker."

Simulation to account for the possibility of convergent evolution. To detect the effect of convergent evolution on the estimation of Pagel's lambda, we introduced convergence to the trait values simulated on the correct phylogeny.

Convergent evolution is a phenomenon in which distantly related species independently evolve similar traits. In order to simulate this, we randomly selected species pairs, each consisting of one passeriform and one non-passeriform. We selected an increasing number of species pairs, representing 1%, 5%, 10% and 20% of the 363 species. Each species pair was given the same trait value to simulate the action of (complete) convergent evolution. Again, we repeated the analysis 100 times and estimated Pagel's lambda. In this way, an increasing number of Passeriformes and non-Passeriformes had identical trait values due to convergence. The results shown above (now Extended Data Figure 10b) that increasing convergence decreases Pagel's lambda values.

However, compared to the effects of topological inaccuracies from the first simulation, the influence of convergently similar trait values on Pagel's lambda estimates was weaker. This can be seen when viewing the two panels next to each other (see figure above, now Extended Data Figure 10ab). Therefore, incorrect inference of the phylogeny appears to have a stronger impact on the estimates of Pagel's lambda than potential convergence of traits.

The manuscript is peppered with a variety of subjective terms (e.g., "INCREDIBLY high number of loci" - really? "exons are not ideal phylogenetic markers" - is any phylogenetic marker ideal?, etc.) that were somewhat distracting. The authors should remove these terms and let the data speak for themselves.

>>> We have revised the manuscript to remove instances of subjective terms.

Referee #3 (Remarks to the Author)

The study by Josefin Stiller and co-workers is an impressive piece of research, and their time-calibrated family-level phylogenetic hypothesis of birds will set the standard in bird phylogenomics for the next while, for example, as backbone for future species-level phylogenetic analyses. About ten years after some of the same authors presented a first genome-scale phylogenetic analyses of modern birds, it is about time for such an extended analyses based on many new genomes generated in the framework of the B10K initiative.

>>> We thank the reviewer for the careful comments and for the assessment of the work as an “impressive piece of research” and that they believe that the new hypothesis will “set the standard for the next while”.

The group has put in quite a bit of effort into resolving hitherto doubtful branches in the bird tree of life. Their strategy of using more than 60000 intergenic loci resulted in a robust phylogenetic hypothesis, supporting - among other findings - the divergence of Neoaves after the K-Pg boundary.

Overall, the manuscript is well written. However, I feel that the authors tried to pack too many things into this manuscript. For example, while it is important to report to a general audience that the increase in genetic loci and the use of intergenic regions helped improving phylogenetic resolution in their case, I find the discussion about increasing taxon sampling versus increasing the number (or type) of loci not very important (and would suggest moving it into the supplement). In the end, it is a matter of the research question whether or not one wants to increase taxon sampling, and in the genomic era, the number of loci is not per se a limiting factor in study design.

>>>We would like to argue that the results presented on the importance of taxa vs. loci are important. In the case of the Neoavian phylogeny, taxon sampling vs. choice of markers have been **central in the discussion of the different topologies** obtained in phylogenomic studies (Jarvis et al., Prum et al., Hackett et al.). Indeed, one of the most important debates in the phylogenomic era has been about the relative impact of models, data types, and taxon sampling. This debate has typically been framed in terms of the trade-off between sampling sites vs sampling more taxa, and it has been a very important debate in avian phylogenomics (e.g., Reddy et al.). Thus, we feel that an explicit discussion is vital to the manuscript.

Overall, these are **important variables in phylogenomic study design**, which makes the insights applicable to future phylogenomic studies of other organisms:.

- The number of taxa to be sampled is often guided by the scientific question, but it is also often limited by sample availability and funds for their sequencing. Taxon sampling is one primary component to resolving a reasonably accurate phylogeny.
- While it is correct that with whole genomes the number of loci is theoretically not the primary limiting factor, the decision of how many loci to analyze is crucial because of financial (\$/CPU hour) and environmental (CO2 footprint) costs involved with computation and analysis. Our tests of how many loci are needed to resolve nodes of various phylogenetic difficulty may be informative for other studies and avoid unnecessary computation on “easy” datasets where fewer loci may sometimes be sufficient.
- The comparison of different data types is informative for study design. Many studies focus on individual data types, e.g. UCEs through targeted enrichment or exons obtained from targeted enrichment or transcriptome sequencing. When planning such a study, it is important to be aware that the choice of targeted marker may impact the phylogenetic results.
- We also include a relatively new class of phylogenetic markers, the intergenic regions. Our way of extracting and filtering them, in addition to the various analyses demonstrating their utility and their often favorable characteristics in terms of phylogenetic model assumptions, provides useful new phylogenetic insight.
- With whole genome data, different data types are more readily available. Nonetheless, UCEs or exons are sometimes chosen even when whole genomes are available because of the greater complexity of extracting introns or intergenic regions. We therefore consider it important to compare the utility of these data types for phylogenomics in order to inform data type selection when whole genomes are available.

The authors should be consistent throughout in presenting confidence intervals when they report divergence times. It is somewhat counterintuitive that the confidence intervals around divergence events are small around the K-Pg boundary and become larger towards present times, suggesting a strong constraint in their time-calibrations around or towards this time point. Were there any cross-validations done regarding the fossil dates? Was there any reason why not using BEAST?

>>>Regarding consistency of presenting CI.

We have revised all instances of reported divergence times and found that CI of divergence times were reported consistently. It is possible that the reviewer refers to instances where instead of the absolute age estimate, we reported the length of a branch (i.e. the difference between the mean inferred age of an older and a younger branch). These branch lengths are given to highlight particularly short branches, which are presumed to cause phylogenetic difficulties (in sections “Mirandornithes as sister to remaining Neoaves dissolves Columbea” and in “Genome-level insights into the passerine radiation”). In two cases, we report the sum of various branch lengths

(section “Conflict in the placement of Phaethontimorphae and Strisores”). We have revised all instances in the text to ensure that any reference to branch lengths versus node age CI are clearly separated.

Regarding variability of CI.

This variability was also highlighted by Referee #1. The patterns are the outcome of using a greater density of fossil calibrations near these old nodes, and of the sequential subtree approach used. Within each subtree, there is a secondary calibration at the root but only a handful of fossil calibrations across the other nodes – so the date estimates at deeper nodes arise from greater prior information and are thus more precise than shallower nodes. A similar pattern can be seen in the mammal dated tree that was inferred using a sequential subtree approach (Álvarez-Carretero et al. 2022), where the deepest nodes in each subtree have more precise date estimates than some of the shallower nodes. This result is therefore not unexpected, and the method used remains the best option for molecular dating using genome-scale data. With increases in the number of genomes and with further refinement of fossil calibrations, we expect to see in future studies a corresponding improvement in the age uncertainty of nodes closer to the present.

We have added the following sentence to the section on the time tree:

“The widest 95% CI were observed for nodes positioned farthest from the calibration points, including the secondary calibrations involved in our sequential-subtree dating approach. However, the prospects for narrowing the intervals are promising through future refinement and addition of fossil-based age constraints.”

Regarding cross-validation analysis.

We did not perform a cross-validation analysis regarding the fossil dates, but did examine the impacts of using two different sets of fossil calibrations (either using all age constraints, or only those that could be assigned more confidently), which resulted in overall similar age estimates (Extended Data Figure 5b-e). We also ran several versions of the dating analysis while varying some of the key settings.

Regarding using BEAST for analysis.

Unfortunately the size of the data set means that it is not feasible to run a more exhaustive set of analyses, such as simultaneously inferring tree topology and divergence times as is the cornerstone of BEAST analyses. Specifically, dating analysis using BEAST would not have been feasible given the number of taxa, leading to a massive parameter space, unless we had reduced the data set to a few dozen loci. The approach used, implemented in MCMCtree, balances the benefits from Bayesian divergence times estimation with computational efficiency. This is done by assuming a tree topology and by using approximate likelihood calculation, which is highly

efficient and performs well under relaxed-clock models when the alignment is large (dos Reis and Yang 2011).

The section "Implications of the new time frame for avian diversification" (466-525) is weak in parts and often speculative, and some of the topics discussed in this section are not really an implication of the new time frame. There is little support provided (statistical and/or through simulations) for some of the macroevolutionary analyses (Lilliput effect, brain size evolution, body mass evolution), and it is not clear how much these analyses are influenced by the overall rather limited sample size, especially towards the tips of the tree.

>>> In order to strengthen this section, we implemented the following modifications:

- We changed the title to "*Implications of the new tree for avian diversification*" to account for the fact that the analyses of Pagel's lambda were not implied by the new time frame.
- We removed speculation from the discussion of the observed patterns of effective population sizes, specifically (removed parts in strikethrough):

"Birds would have been well-positioned to exploit landscapes newly devoid of competitors and predators following the K–Pg mass extinction because of their flight capabilities. Thus, it is noteworthy that the branch to Elementaves is among those with elevated Ne since this clade bears the hallmarks of adaptive radiation in its diversity. Vagile insectivores and marine species such as Strisores and Aequornithes could have rapidly expanded into early-succession habitats. A less dramatic spike was also observed around the end of the Paleogene (Fig. 5b), which could be associated with ecological opportunity during the late Oligocene warming, after the glaciations and potential extinctions of the early Oligocene 32. After quickly expanding their populations for several million years before ecological conditions stabilized, the population sizes returned to normal."

- We removed an interpretation that was not necessarily following the observed pattern of body mass evolution in passerines:

"Passeriformes exhibited a burst of body mass evolution at their most recent common ancestor, indicating rapid phenotypic diversification"

- We removed an example of brain size in a flightless bird (*Atlantisia rogersi*) because the brain volume of this species was imputed in our dataset. While we found no major differences between the imputed dataset and a dataset using only data points with available brain volume metrics (Extended Data Figure 10h), highlighting this species would be misleading.
- We have added simulations to address the impact of topological inaccuracy and convergence on Pagel's lambda. These are described in detail in the response to the second to last comment by Referee #2.

Regarding the impact of taxon sampling on ancestral reconstruction of body size

It is correct that our sampling of more recent divergences (within families) was limited. We would therefore expect that increased sampling would most likely impact reconstructions on more recent branches, while the older events are more likely to be stable irrespective of the sampling. Even though recent divergence events are not the focus of the paper, we performed a resampling analysis to assess the impact of limited within-family sampling.

We first simplified the timetree to the family level by keeping only one species in each family. We used body size values from the AVONET database, this time using the entire database, which covers most bird species. For each family, we randomly selected the trait value of one species to represent its family and used it as the input value for the ancestral reconstruction. This way, the sample size remained constant in each run, while the trait values reflected the variation of the trait within each family. We repeated this 100 times.

In the figure below (now Extended Data Figure 10g), the solid purple line represents the result of the ancestral reconstruction of the full dataset, and the 100 gray lines are the resulting ancestral reconstructions of the replicates. We find that from ca. 70 million years, through the K-Pg boundary, and to ca. 25 million years the full dataset and the replicates were highly congruent and support the spike around the K-Pg boundary and a subsequent overall declining trend of body size. From ca. 25 million years onwards, the estimate from the full dataset and of the replicates start to diverge increasingly, with the full dataset supporting a slightly less severe decline than the replicates. These findings suggest that the effect of increasing sample size on ancestral reconstruction has a larger impact at recent time scales than on the deeper events. We therefore believe that the discussion in the manuscript on the inferred changes in body size around the K-Pg event and the subsequent decline is valid, while we do not focus on any more recent events.

Added Extended Data Figure 10g:

Caption of Extended Data Figure 10g:

“g, Impact of taxon sampling on ancestral reconstruction of body size. The solid purple line is the result of the ancestral reconstruction of the full dataset. The 100 gray lines are ancestral reconstructions from analyses in which each species’ trait values were randomly drawn from the range of values across their family. The chosen values do not impact the reconstructions at deep time scales but estimates diverge more from 25 million years ago to the present, indicating that increased taxon sampling within families may lead to a different trajectory on more recent time scales.”

In the main text, we modified Fig. 5c and its caption to acknowledge the potential variability in the reconstructions from 25 Ma on as dashed lines:

Caption of Fig. 5c:

“c, Variations in body mass and relative brain size over time changed in different directions after the K–Pg event. Solid lines indicate mean values and ribbons mark 95% confidence intervals. The dashed parts of the reconstruction (from 25 Ma) indicate possible uncertainty due to the lack of within-family sampling (Extended Data Fig. 10g).”

Short Online Methods:

“To investigate whether our sampling of one species per family could impact ancestral reconstructions, we performed 100 replicates, each with tip values modified to reflect the family’s range in body size calculated from AVONET (Extended Data Fig. 10g). We also confirmed that inclusion of the imputed brain size values did not change the shape of the ancestral reconstruction (Extended Data Fig. 10h).”

Supplementary Methods (Supplementary File 3) in section “Analysis of body mass and brain size evolution”:

“To investigate whether our sampling of one species per family could impact ancestral reconstructions, we performed 100 ancestral reconstructions each with tip values modified to reflect the family’s range in body size. To achieve this, we simplified our timetree to one species per family and drew a body size value for each tip from the range within its family (calculated using the entire AVONET database). This way, the sample size remained the same in each run, while the trait values reflected the variations of the trait within each family. (Extended Data Fig. 10g). We also confirmed that inclusion of the imputed brain size values did not change the shape of the ancestral reconstruction compared to a dataset containing only values from the literature (Extended Data Fig. 10h).”

Regarding the impact of taxon sampling on ancestral reconstruction of relative brain size

Unfortunately, we could not apply the same resampling scheme for the relative brain size because data was not available for sufficient bird species outside of the 363 species sampled here. Instead we compared the results of the ancestral reconstruction using two sets with different sample sizes.

The first dataset contained the brain size (volume of the brain case) of 228 species collected from the literature, which we corrected for body mass. This relative brain size was calculated as the residual from a log-log phylogenetic Generalized Least Square regression (a single regression slope and the lambda transformation) of absolute brain size against body mass. For the second dataset, we imputed the missing values of brain size using the ‘missForest’ package in R to obtain a dataset of brain size covering all 363 species. The Out of Box error (OOBerror) was 0.0003, which means the normalized root mean squared error was 0.0003. These values were again transformed to relative brain size as above. This is the dataset that was presented in the main text of the original submission.

Using the non-imputed and the imputed dataset for the ancestral reconstruction with BayesTraits, we found that the ancestral reconstructions of relative brain sizes were highly similar between the two datasets (see figure below, now Extended Data Figure 10h).

This additional analysis is now included in the manuscript as part of the Short Online Methods

“We also confirmed that inclusion of the imputed brain size values did not change the shape of the ancestral reconstruction (Extended Data Fig. 10h).”

Supplementary Methods (Supplementary File 3) in section “Analysis of body mass and brain size evolution”:

“We also confirmed that inclusion of the imputed brain size values did not change the shape of the ancestral reconstruction compared to a dataset containing only values from the literature (Extended Data Fig. 10h).”

We added the results as Extended Data Figure 10h:

Caption of Extended Data Figure 10h:

“h, Impact of imputation on ancestral reconstructions of relative brain size. The non-imputed dataset contained only values based on the literature, while the imputed dataset included some values inferred using phylogenetic information. Solid lines indicate mean values and ribbons mark 95% confidence intervals. The two ancestral reconstructions are almost indistinguishable.”

While the display items are typically of high quality, I was somewhat disappointed with Fig. 1. For example, I find that the smaller panel is too close to the main figure. I also thought that it would be nice to have some silhouettes of birds, as e.g. in Extended Data Fig. 4. Perhaps the authors could come up with a better caption for Fig. 2?

>>>Thank you for these suggestions.

Regarding Fig. 1:

We have re-designed Fig. 1 in a manner that we believe will make this figure more accessible. Specifically:

- We have separated the previous smaller panel into a full width panel (now Fig. 1a) that shows the topology simplified to the orders and the major higher-level classification. This panel emphasizes the relationships among the orders and higher clades.

- Fig. 1b now shows the full, time-calibrated tree in a different plotting scheme (half circle as opposed to previous rectangular). This tree emphasizes the tight time scale of the diversification of Neoaves, in addition to the breadth of taxon sampling.
- We have added representative bird drawings by Jon Fjeldså for the major groupings as suggested by the reviewer.
- The figure maintains the consistent color coding of major clades that follows throughout other figures of the manuscript.

Regarding Fig. 2:

Figure 2 was changed in the course of the revision in response to reviewers' remarks. This involved removing previous panel c altogether and significantly simplifying panel b and d (now Fig. 2c). This allowed us to shorten and simplify the caption.

Other comments/questions:

- Pagel's lambda (reported in Fig. 5): What is the interpretation of this pattern? Is the idea that a phylogeny is better if Pagel's lambda of a mapped trait gets closer to 1? But what if there is convergence then?

>>> These questions were also posed by Reviewer 2 (second to last comment). We refer the reviewer to the discussion there.

- Extended Data Fig. 5: Perhaps label it with "Jarvis et al." and "Prum et al." instead of "Jarvis" and "Prum"?

>>>Fixed.

- Extended Data Fig. 10 and page 25: Did you also test other models than the variable-rates one? Does the variable-rates model explain the data better than e.g. a Brownian motion model? Was brain size used as volume or was it corrected for body size?

>>>As addressed in one of the earlier responses to the reviewer, we used **relative brain size**, i.e. the volume of the brain case corrected for body mass. We clarified this throughout the manuscript (short online methods and Supplementary Methods, main text and Figure 5 caption).

Regarding testing of additional models:

In addition to the previously presented variable-rates BayesTraits analysis, we now compare three single-process models (Brownian motion (BM), early burst (EB) and Ornstein–Uhlenbeck (OU)) using the 'fitContinuous' function and default settings in the R package Geiger v2.0.

Regarding model fit:

Assessing model fit between the BayesTraits variable-rates model and the single-process models is complicated by the fact that BayesTraits and the single-process models are not fit in a common framework with consistent likelihood calculations. BayesTraits provides a marginal likelihood, while the single-process model provides a maximum likelihood. Thus, we followed the approach described in Cooney et al. (2017) (<https://doi.org/10.1038/nature21074>) to compare the fit of the alternative evolutionary models. This approach uses the mean of the rate-scaled trees output by BayesTraits and calculates the maximum likelihood of a BM model fit to this tree with the same

trait data. The idea here is that if there is evidence for non-Brownian trait evolution (e.g. rate heterogeneity) within the tree, a BM model applied to the rate-scaled tree should return a higher likelihood than a BM model applied to the time tree.

Based on the model comparisons using delta AIC, the BayesTraits variable-rate model had the best fit to the data across all analyses (see tables below, now included as Extended Data Figure 10ef).

Extended Data Figure 10e,f:

e				f			
Tree used	Model	Likelihood	AIC	Tree used	Model	Likelihood	AIC
Timetree	BM	-211.4874	426.9748	Timetree	BM	315.5601	-627.1203
Timetree	OU	-211.4874	428.9748	Timetree	OU	324.4819	-642.9638
Timetree	EB	-210.1256	426.2513	Timetree	EB	315.5593	-625.1186
BayesTraits mean rate tree	BM	-206.4091	416.8183	BayesTraits mean rate tree	BM	355.4663	-706.9327

Caption for Extended Data Figure 10e,f:

“e, Model comparisons between variable-rate and single-process models (BM: Brownian motion, EB: early burst, OU: Ornstein–Uhlenbeck) for body size. f, Model comparisons as in (e) for relative brain size.”

Short Online Methods:

“In addition to BayesTraits, we compared the fit of three single-process models (BM, early burst (EB), Ornstein–Uhlenbeck (OU)) using the `fitContinuous` function in Geiger v2.0¹¹¹. We used the mean of the rate-scaled trees of BayesTraits and calculated the likelihood of a BM model on this tree with the same trait data, assuming that under non-BM trait evolution, a BM model applied to the rate-scaled tree should return a higher likelihood than a BM model applied to the timetree¹¹². Model fit was assessed using AIC (Extended Data Fig. 10ef).”

Supplementary Methods (Supplementary File 3) in section “Analysis of body mass and brain size evolution”:

“In addition to BayesTraits, we compared the fit of three single-process models (Brownian motion (BM), early burst (EB) and Ornstein–Uhlenbeck (OU)) using the `fitContinuous` function in the R package Geiger v2.0⁵⁹. As the BayesTraits and the single-process models are not fitted in a common framework with consistent likelihood calculations, we used the mean of the rate-scaled trees output by BayesTraits and calculated the likelihood of a BM model fit to this tree with the same trait data⁶⁰. If there is evidence for non-BM trait evolution (e.g. rate heterogeneity) within the tree, a BM model applied to the rate-scaled tree should return a higher likelihood than a BM model applied to the timetree⁶⁰. Model fit was assessed using AIC (Extended Data Fig. 10ef).”

- I did not find a link to the computational scripts used for the different types of analyses and the accession numbers of the genomes used, but maybe I have overlooked this information.

>>>In addition to all underlying raw data files (alignment, tree files, csv files with metrics for alignments and gene trees), we have now added all files needed to reproduce the main figures and those of the Extended Data Figures onto our data repository. This includes csv files with source data, commented R code for calculations and plotting, and any relevant resulting data files. This data repository will be frozen and equipped with a DOI after potential future revisions of the datasets. Until then, all material is available under the link:

<https://sid.erda.dk/sharelink/ENhZODU9YE>

We have added the additional resources to the Data Availability section.

The accession numbers of the genomes used have been added to Supplementary File 1.

Reviewer Reports on the First Revision:

Referees' comments:

Referee #1 (Remarks to the Author):

I appreciate that the authors took the time to respond to each of the comments made by me and other previous reviewers. While the responses seem to indicate major changes, I find the revisions to the manuscript are minimal except for some re-vamped figures and some re-wording of the text. The main strength of this paper is the large size of the data generated by this group (already previously published in Feng et al. Nature 2020). The authors have chosen to conduct a similarly large number of different analyses to examine various aspects of the bird tree and the behavior of the genomic data. In all of these, their methods are hard to understand (see more below) and their interpretations are debatable and mostly lead to conclusions that 'it's complicated...' While I don't have a problem with publishing studies that honestly show the complexities of empirical data, my concern is that it is unclear what, if anything, we have learned about the evolution of birds or their genomes from this paper. There is little novelty in this paper and many of the findings lead to more confusion than clarity about the bird tree. Moreover, it is unclear what readers, especially students, are supposed to take away from this paper that can be applicable to other studies.

The authors place a lot of value in stating that their tree is well-resolved and well-supported. But they also show that there is a high level of discordance. This discordance is largely at the nodes at the base of Neoaves - the same nodes that have been intractable in previous studies. This study has not brought us closer to resolving these nodes with any confidence, despite what the authors state, as the levels of discordance within their preferred loci set and the disagreement across different data sets is high. For example, I am still unconvinced that their newly coined clade, Elementaves, is warranted given that this clade is not recovered using different analytical methods or slightly different data sets.

The time-calibration analyses could be novel, however the methods applied in this paper are very hard to understand and would be impossible to replicate. The description of the time-calibration is simply stated as "we empirically generated calibration densities for 34 nodes using 187 fossil occurrences" [218-19 lines] with reference to a supplemental file that describes the fossil evidence but not how they conducted their analyses. Furthermore, a few lines down [lines 224-225] there is mention of a sequential-subtree approach but little information for the reader to understand this, either in the main text or in the supplemental methods.

The use of intergenic regions is still not clearly justified and at times the paper seems to put more emphasis on showing that the 63K loci set is superior rather than evaluating the strengths (or model fit) of loci regardless of their genic category. I understand the desire to sample loci across the genome and the strategy of choosing regularly spaced loci. However, the explanation of why intergenic loci are superior is simply an a priori assumption [lines 189-191] and "supported" by their findings, which seems a bit circular to me. Furthermore, given the availability of whole genomes I am a bit surprised that the authors are not trying to maximize the amount of data they are using. What is the rationale for favoring the 63K dataset rather than the 80K (including introns but not exons)

datasets? Is it because with the 80K set, the novel clade Elementaves is not recovered [again pointing to the tenuous nature of this clade]? Subsequent studies examine model violations and systematic biases across loci. The 63K loci set is generally the best one, but there are perfectly good loci in the other sets, as well as not good intergenic loci. Why did the authors not attempt to build a tree using all loci excluding the ones that violate model fit? The emphasis on conducting analyses by separating by gene type seems to overemphasize the one unique data type in this paper in contrast to previous papers but also undervalues the vast genomic data they have access to.

The most interesting aspect of this paper is the strange behavior of taxon sampling. The general sentiment in phylogenetics is that increasing taxon sampling should improve phylogenetic estimates but in this study there is a strong evidence that increased taxon sampling can also lead to increased discordance. In some of the illustrated examples, like the sister relationship of Accipitriformes and Strigiformes, the main topology and an alternative topology have similar quartet scores and only using the whole dataset recovers the 'main topology'. The authors interpret this as meaning that their tree is robust to taxon sampling (line 429) but again, this feels circular to me because their 'test' is whether the main topology is recovered.

There are still several extraneous analyses that do not contribute much to the paper. The analysis of phylogenetic signal in terms of the fit of the main tree to morphological traits (AVONET) is suspicious. All of the traits used have high incidences of convergence across birds. Pagel's lambda measures phylogenetic signal by examining how closely related taxa resemble each other. The sampling in this study has a larger representation of passerines than any previous study that they are comparing to. The result is that the Pagel's lambda is higher for the 63K tree than the Prum tree is really a function of the number of passerines represented in the 63K tree. The authors claim to account for taxon sampling by randomly pruning their tree to be the same number of terminals and the Prum tree but again, this will still overrepresent passerines in the pruned trees. A real test is if they pruned their tree to match the same taxa as in the Prum tree, which I suspect will lead to no difference in Pagel's lambda score to fit these linear measurements. Moreover using these morphological measurements in a univariate, rather than multivariate, context additionally oversimplifies this analysis.

Line 94 – "phylogenetic loci" – this is unclear; do you mean phylogenetically informative loci?

The updated Fig 1 has less information than the previous version. I suppose it is a matter of preference which one you use but the smears of color behind the bird icons are distracting and not very aesthetic.

Lines 244-245 – "In particular, six nodes had quartet support below 35% and another eight nodes had support below 37%." Why is it important to distinguish these percentages differently? Why not just say 14 nodes below 37%?

Lines 260-261 – "Our companion paper..." – the citation for this is incomplete and if I recall correctly, this paper was declined by Nature. This should be reworded.

Fig. 2c is related to Fig 3a; why not keep those together? I would recommend one figure (2) for

discordance and one figure (3) that deals with taxon sampling.

Is GlobalBS the same as multi-locus bootstrapping (MLBS)? This is not a term that is used often and needs better explanation in this paper.

Referee #2 (Remarks to the Author):

The authors have satisfactorily addressed my concerns.

Referee #3 (Remarks to the Author):

I have now gone through the revised version of the manuscript and the point-by-point response from the authors. I stand by my original assessment that this is an impressive piece of research and that the time-calibrated family-level phylogeny of birds will set the standard in bird phylogenomics for some time. In the revised version, and/or in the rebuttal letter, the authors have clarified the major open points; they have improved the figures (I particularly like the new Figure 1 and what is now Figure 2); they improved the "implications" section; and they provide further details (e.g. on evolutionary models) and, importantly, underlying raw data and code to reproduce their findings.

I still find that the manuscript is a bit overloaded and believe that the discussion on increasing taxon sampling versus increasing the number/type of loci is not key to the main story line. However, I also see their arguments in keeping this part as is (If the manuscript had to be shortened, this would be a section that could be moved to the supplement). Also, I am still of the opinion that a cross-validation would have been an elegant strategy with regard to the fossil-calibrations. However, both of these issues are minor.

As a side note, I would like make one additional comment (referring to their response in the rebuttal letter): I am happy to see that the authors agree with me that the number of loci is not the limiting factor anymore. I am less happy about how they responded to this (\$/CPU, CO₂ footprint). First, I think we should do science as best as we can. When spending millions of dollars on a research project (just think about the salaries of all people involved in that project), the CPU hours shouldn't matter. Second, \$/CPU hour is constant (and low), irrespective of how long an analysis will take. Third, if the CO₂ footprint is an issue, then one should think about using CO₂-neutral computing facilities running with renewable energy (It would be very interesting to know how much the computing contributed to the CO₂ footprint of this research project compared to all the other parts such as sampling birds in remote locations, shipping the samples, sequencing them, consortia meetings [in person and via zoom], all authors going through that manuscript on their computers, etc.)...

Minor comments:

L90: Why does it need reference 4 here, where the authors report what they have done in this study.

L111: Does it really need "widespread" here? If something is incongruent, it is incongruent... Perhaps just say "incongruences".

L130, "through the Chicxulub impact": I would argue that not only the impact itself matters, but also what it caused on a global scale.

L158-159: I might be a bit biased here, but working on closely related taxa, it is not really an issue to "create comprehensive datasets ... across many taxa"; perhaps add the notion that this is challenging when dealing with deeper time scales?

L171: maybe better "94,402 1-kb loci"? By adding "-" it is clear that "1" has nothing to do with the number of loci...

Author Rebuttals to First Revision:

Referees' comments:

Referee #1 (Remarks to the Author):

I appreciate that the authors took the time to respond to each of the comments made by me and other previous reviewers. While the responses seem to indicate major changes, I find the revisions to the manuscript are minimal except for some re-vamped figures and some re-wording of the text. The main strength of this paper is the large size of the data generated by this group (already previously published in Feng et al. Nature 2020). The authors have chosen to conduct a similarly large number of different analyses to examine various aspects of the bird tree and the behavior of the genomic data. In all of these, their methods are hard to understand (see more below) and their interpretations are debatable and mostly lead to conclusions that 'it's complicated...' While I don't have a problem with publishing studies that honestly show the complexities of empirical data, my concern is that it is unclear what, if anything, we have learned about the evolution of birds or their genomes from this paper. There is little novelty in this paper and many of the findings lead to more confusion than clarity about the bird tree. Moreover, it is unclear what readers, especially students, are supposed to take away from this paper that can be applicable to other studies.

Response: We appreciate the time and effort that the reviewer took for this second round of comments. We address them point by point below. We revised the methods to make them clearer. We also tried to make our interpretations and conclusions clearer.

In terms of novelty, the paper does present novel discoveries on the causes of incongruence for hard-to-resolve branches of the tree, and helps resolve debates on why previous estimates have differed. In addition, we present new phylogenomic methods and insights into bird evolution. First, this is the first time that entire genomes (including intergenic regions) are interrogated to resolve the deep branches of the avian phylogeny. We believe that this is not only an insightful approach for other studies on biodiversity genomics but also interesting for students in phylogenetics or bioinformatics. Second, the manuscript identifies the specific causes of incongruence and uncertainty for specific branches of the bird tree. These findings on the phylogenetic incongruences across bird families provide important insights into why the neoavian radiation has been a long-standing challenge in phylogenetics. The patterns observed will be of value to other systematic analyses. Rather than focusing on a single tree, the paper aims to show the outcome of various analyses on different difficult nodes. Such comprehensive analyses would provide text-book examples that phylogenetic patterns can differ along the genomes and that some evolutionary divergences are much more challenging to resolve than others.

The authors place a lot of value in stating that their tree is well-resolved and well-supported. But they also show that there is a high level of discordance. This discordance is largely at

the nodes at the base of Neoaves - the same nodes that have been intractable in previous studies. This study has not brought us closer to resolving these nodes with any confidence, despite what the authors state, as the levels of discordance within their preferred loci set and the disagreement across different data sets is high. For example, I am still unconvinced that their newly coined clade, Elementaves, is warranted given that this clade is not recovered using different analytical methods or slightly different data sets.

Response: We would like to argue against the notion that the study has not brought us closer to confidence regarding the previously intractable nodes in the tree. We demonstrate the reasons why some methods and some data sets give different answers at previously intractable nodes, and the rationale for selecting ones to reflect the true phylogeny. These are not slightly different data sets, but greatly different in their results, like exons vs introns of the same genes or vs intergenic regions. We demonstrate that exons are not appropriate for avian phylogenomics. For the majority of difficult clades, we were able to demonstrate consistency within datasets and were supported with high statistical confidence. Elementaves was supported across several datasets, including all coalescent-based analyses that included intergenic regions. The same was true for Columbaves and Cursorimorphae. Recovering Mirandornithes as sister to the remaining Neoaves was also consistent. There are a few nodes where consistent relationships were not obtained with different datasets, namely the placements of Strisores, Strigiformes (with respect to Coraciimorphae and Accipitriformes) and the always challenging Opisthocomiformes. For each of these, we discuss them in detail and provide evidence for why they remain difficult.

In order to better reflect consistency and differences between our different analyses, we added a graphic metatable to Fig. 3 as panel d. It summarizes the level of support (bootstrap or PP support) of species trees obtained from different datasets for specific clades. For the clades of interest, we selected difficult nodes defined as nodes that required at least 2000 loci to be consistently recovered across subsetting experiments. These difficult nodes contain all the challenging nodes of the backbone of the neoavian phylogeny, in addition to some challenging nodes in Palaeognathae and Passeriformes. The main takeaway message is that several relationships have in fact been consistently recovered, especially if intergenic regions are used.

Caption for Fig. 3d:

“Ten selected species trees, data types used in each, and the support for all challenging branches (labeled in panel b). Asterisks indicate relationships in Passeriformes that differ from previous studies.”

The time-calibration analyses could be novel, however the methods applied in this paper are very hard to understand and would be impossible to replicate. The description of the time-calibration is simply stated as “we empirically generated calibration densities for 34 nodes using 187 fossil occurrences” [218-19 lines] with reference to a supplemental file that describes the fossil evidence but not how they conducted their analyses. Furthermore, a few lines down [lines 224-225] there is mention of a sequential-subtree approach but little information for the reader to understand this, either in the main text or in the supplemental methods.

Response: We apologize if the methods were difficult to understand. In the main text, we refrain from including detail on these methodological descriptions due to space constraints but we now refer to the Methods to guide the reader to the further detail on the applied approaches (addition underlined).

Lines 221-223 (line numbers refer to the revised version with tracked changes):

“To time-calibrate our main tree, we empirically generated calibration densities for 34 nodes using 187 fossil occurrences (Supplementary File 2) and applied these in a Bayesian sequential subtree framework (Methods).”

We have also modified the description of the sequential-subtree dating approach and the empirically generated distributions for fossil calibrated nodes in the Methods to provide a clearer overview of the procedure (in addition to more detail given in the supplementary methods in Supplementary File 3):

Lines 986-1015 (line numbers refer to the revised version with tracked changes):

“To place an evolutionary timescale on our main tree, we performed molecular dating using a Bayesian sequential-subtree approach 78. This involved using date estimates from an initial analysis of a backbone tree (56 tips), containing two representatives of each of 11 subtrees. This provided secondary calibrations for subsequent dating analyses of 11 subtrees (19–42 tips each). The subtrees were then attached to the backbone tree to assemble a timetree of all 363 taxa.

For node-based calibrations, we identified 34 clades with fossils fulfilling best practice criteria 72 (Supplementary File 2). We used the CladeDate 79 method to generate calibration densities empirically based on fossil occurrences (187 fossils) and estimators of distributions in which the truncation was the estimated age of the clade 22,73 We used the Strauss and Sadler 75 estimator for uniformly distributed fossil occurrences; otherwise, we excluded the Quaternary record or used estimators that do not assume sample uniformity 74. The resultant distributions of clade ages were used to fit Student-skew distributions to parameterize calibration priors.”

To better explain how calibration densities were obtained, the relevant section in Supplementary File 3 was rewritten for clarity and now gives additional detail:

Page 6-7 of Supplementary File 3:

“For node-based fossil calibrations, we derived calibration distributions empirically using the recently described CladeDate method 22,23. We first selected crown clades that were well supported across phylogenomic analyses and that had a good-quality fossil record. Within each clade, we identified the oldest fossil that fulfilled best practices criteria for fossil calibrations 24. In particular, we identified fossils that were composed of multiple bones, were chronologically well constrained, and whose phylogenetic position within the crown-clade was supported by explicit phylogenetic analyses or strong apomorphy-based evidence.

To generate calibration densities, CladeDate requires a sample of the fossil record of each calibration clade. For widespread clades, we generated this sample by identifying the oldest fossil in each continental landmass. Restricting the sample to one fossil per landmass minimizes problems of spatial non-independence and confounding histories of dispersal 22. For clades restricted to single landmasses, we used the oldest fossils of different species in different geological formations.

With a set of fossil ages for each clade, we then used CladeDate to generate distributions representing the uncertainty of the age of the calibration clades. CladeDate uses various estimators of the location of the upper bound (age) of truncated distributions to generate estimates of the age of clades based on the sample of fossils. When the distribution of fossils ages did not depart significantly

from a uniform distribution, we used the Strauss and Sadler 25 estimator, otherwise, we excluded the Quaternary record to improve uniformity or used various estimators that do not assume sample uniformity 23. In addition to point estimates, CladeDate uses a Monte Carlo procedure for generating a distribution of ages representing estimation uncertainty. In addition, CladeDate accounts for fossil age uncertainty by resampling fossil ages from their chronostratigraphic intervals as part of a Monte Carlo procedure. The resultant Monte Carlo distribution of clade ages was used to estimate the parameters of a Student-skew distribution that was then used to parameterize calibration priors in MCMCtree (see below). For further details about the CladeDate method see 23. Details of the calibration clades, fossils used, and empirical density estimation are provided in the Supplementary File 2.”

The use of intergenic regions is still not clearly justified and at times the paper seems to put more emphasis on showing that the 63K loci set is superior rather than evaluating the strengths (or model fit) of loci regardless of their genic category. I understand the desire to sample loci across the genome and the strategy of choosing regularly spaced loci. However, the explanation of why intergenic loci are superior is simply an a priori assumption [lines 189-191] and “supported” by their findings, which seems a bit circular to me. Furthermore, given the availability of whole genomes I am a bit surprised that the authors are not trying to maximize the amount of data they are using. What is the rationale for favoring the 63K dataset rather than the 80K (including introns but not exons) datasets? Is it because with the 80K set, the novel clade Elementaves is not recovered [again pointing to the tenuous nature of this clade]? Subsequent studies examine model violations and systematic biases across loci. The 63K loci set is generally the best one, but there are perfectly good loci in the other sets, as well as not good intergenic loci. Why did the authors not attempt to build a tree using all loci excluding the ones that violate model fit? The emphasis on conducting analyses by separating by gene type seems to overemphasize the one unique data type in this paper in contrast to previous papers but also undervalues the vast genomic data they have access to.

Response:

We have edited several sentences for more clarity, added a new analysis (removing loci that failed tests irrespective of data type category, as suggested), and included the above mentioned metatable (Fig. 5d) to clear confusion. Beyond these edits, we argue three points.

Regarding circularity. Perhaps presenting the intergenic main tree up front, before justifying it more strongly in lines 480 (line numbers refer to the previous version of the manuscript) onwards, made the argument for its use seem circular. However, our evidence-based approach was not circular. We do not prefer to exclude exons because they

do or do not recover favored clades. We provided several lines of evidence that did not rely on our intergenic main tree being the correct tree:

1. Statistical tests showing that we can reject more than 30% of exons because they violate the assumptions of the evolutionary models (far more than other data types).
2. Exons are more sensitive to subsetting, comparing pairs of exon trees together, and not to the main tree (Figure 4ab, Extended Data Figures 7g).
3. Using exons leads to lower species tree branch support (Extended Data Figure 7b).
4. We state that we concatenated the individual exons (which are often quite short) to yield a single alignment. This was done for practical reasons, given the limited phylogenetic signal in individual exons, but it comes with the cost of increasing the risk of violating the assumption that analyzed regions are free of intralocus recombination.
5. Exons had less phylogenetic signal (Extended Data Fig. 7d-e) and higher measures of estimation difficulty (Extended Data Fig. 7f).

In addition, as the reviewer points out, we argue that first principles (e.g., the desire to avoid recombination) also support not using exons (which can span 500Kbp). Early in the manuscript, we cite prior literature on data types in avian phylogenetics. This literature provides both theoretical and empirical arguments that analyses of exons are more likely to yield incorrect results. In the interest of brevity, we chose to summarize the theoretical argument (i.e., the statement that “...selection to maintain protein structure and function places constraints on exon evolution”) but the idea that exons might be misleading in analyses of deep avian phylogeny has strong support in the earlier literature. Our results (points 1-5 above) add to that literature. Thus, the *a priori* assumptions match the observed results, which are obtained independently and with no reference to those assumptions. We revised the text to make this point clearer, to prevent the possibility of our argument seeming circular.

Lines 191-194 (line numbers refer to the revised version with tracked changes):

“Among thousands of inferred trees, we focus on this tree because the findings reported below show that intergenic regions reduce systematic error due to model misspecifications – results that match a priori expectations and previous analyses 11,34”

Lines 555-564 (line numbers refer to the revised version with tracked changes):

“These results indicate that the inclusion of exonic loci, even if they constitute just 10% of the loci used for analysis and even if restricted to those that pass tests of model fit, can impact the most unstable parts of the tree. In contrast, exclusion of introns did not make a difference topologically. Nevertheless, we treat the five branches that differ between purely intergenic regions and these alternative trees as uncertain.”

Regarding maximizing the amount of data to use for phylogenetic inference. We would like to clarify that the 80K tree (including introns, but not exons) is identical to the main intergenic tree topology (63K), unlike what the reviewer asserts. Our added metatable should help prevent similar confusion by readers. The topology changes when exons are included (94K), but Elementaves is still present. Our results already included a total evidence tree (159K), which was identical to the 94K tree. In fact, Elementaves is recovered by all analyses that include intergenic loci and use the coalescent-based method; however, it is not recovered using concatenation. The recovery of Elementaves is therefore not a matter of the amount of data but of the analysis framework. The different topologies and analyses supporting it are shown in Extended Data Figure 3a. We hope that the addition of the metatable as a main text figure in Fig 3d makes it easier to assess which analysis supported which clade and to spot the total evidence trees.

Regarding a total evidence tree without loci that violate model fit. Following the reviewer's suggestion, we added an analysis that used all gene trees that passed the saturation and heterogeneity tests resulting in 153k trees of intergenic regions, introns, UCEs, exons. We found that this analysis resulted in the same topology as the 94K tree, which used all intergenic regions including those that overlapped with introns and exons. The topology was also the same as the already included total-evidence trees (159K, 128K), which included all gene trees irrespective of model fit. All these trees with more loci are quite similar to the main intergenic tree, with 5 branches being different (out of 360). They all recover Elementaves. They differ in the local placement of Strisores within Elementaves, position of Strigiformes (with respect to Coraciimorphae and Accipitriformes), and the enigmatic Opisthocomiformes. Because of these disparities, we have treated the exact positions of these groups as uncertain. We have now updated the main text to add these results and clarify this point.

Lines 553-561 (line numbers refer to the revised version with tracked changes):

“Removing loci that failed saturation and stationary tests from the full set (153k loci left) returned the same tree, albeit with low support on branches conflicting with the main tree (Fig. 3d). These results indicate that the inclusion of exonic loci, even if they constitute just 10% of the loci used and even if restricted to those that pass tests of model fit, can impact the most unstable parts of the tree. In contrast, exclusion of introns did not make a difference topologically. Nevertheless, we treat the five branches that differ between purely intergenic regions and these alternative trees as uncertain.”

As the reviewer suggests, we can still simply remove those loci that violate the model fit at statistically significant levels. Our justification for not relying solely on model tests is the following: Recall that model violation statistical tests assume that data follow the model as the null hypothesis. Thus, these statistical tests do not always find all cases of model violation to be statistically significant (false negatives). Even when we do not have high

statistical confidence to reject the null hypothesis, we believe it is best to remain cautious. For example, 30% of exons failed the saturation test at significant levels, leading us to suspect that others too could easily have substantial violations that do not rise to the level of statistical significance. Our tests clearly demonstrated that some data types are worse than others; e.g. for exons, we rejected the null hypothesis in a disproportionately high number of loci. Thus we believe it is more judicious to rely not just on statistical tests of model fit but also the data type.

The most interesting aspect of this paper is the strange behavior of taxon sampling. The general sentiment in phylogenetics is that increasing taxon sampling should improve phylogenetic estimates but in this study there is a strong evidence that increased taxon sampling can also lead to increased discordance. In some of the illustrated examples, like the sister relationship of Accipitriformes and Strigiformes, the main topology and an alternative topology have similar quartet scores and only using the whole dataset recovers the 'main topology'. The authors interpret this as meaning that their tree is robust to taxon sampling (line 429) but again, this feels circular to me because their 'test' is whether the main topology is recovered.

Response: We agree that the taxon sampling results are a very interesting aspect of our paper. The reviewer is partially misreading our results. Here is the paragraph in question (with the underlined phrase being an addition for more clarity):

"We observed that changes in taxon sampling affected ordinal relationships in only three cases (Extended Data Fig. 6f), with the aforementioned Accipitriformes+Strigiformes being the strongest example (Fig. 3a). More frequently, we observed that increasing taxon sampling affected the amount of gene tree discordance but did not change the supported topology (see an example for Telluraves+Elementaves in Fig. 3a). We conclude that the inferred avian species tree is relatively robust to taxon sampling, with some exceptions."

We assert that in most cases, our results are robust to taxon sampling, as clearly shown in Extended Data Figure 6f. The Accipitriformes+Strigiformes is the exception – the interesting case where our results are in fact sensitive to taxon sampling. We point out this exception in detail as it is surprising; we do not interpret it as indicating that the intergenic main tree is correct or our results are robust. In fact, the position of Strigiformes is one of the areas where we are uncertain as to whether the intergenic main tree is correct or if even a single true tree exists, as it could present a case of hybridization. What we take as robustness is that the support for alternatives for other parts of the tree do not change as we change taxon sampling.

There are still several extraneous analyses that do not contribute much to the paper. The analysis of phylogenetic signal in terms of the fit of the main tree to morphological traits (AVONET) is suspicious. All of the traits used have high incidences of convergence across birds. Pagel's lambda measures phylogenetic signal by examining how closely related taxa resemble each other. The sampling in this study has a larger representation of passerines than any previous study that they are comparing to. The result is that the Pagel's lambda is higher for the 63K tree than the Prum tree is really a function of the number of passerines represented in the 63K tree. The authors claim to account for taxon sampling by randomly pruning their tree to be the same number of terminals and the Prum tree but again, this will still overrepresent passerines in the pruned trees. A real test is if they pruned their tree to match the same taxa as in the Prum tree, which I suspect will lead to no difference in Pagel's lambda score to fit these linear measurements. Moreover using these morphological measurements in a univariate, rather than multivariate, context additionally oversimplifies this analysis.

Response:

Regarding the impact of differing taxon sampling on Pagel's lambda estimates. To address the concern that the large number of taxa in Passeriformes in our study compared to the Prum et al. study causes the higher Pagel's lambda fit of our tree topology, ideally, we could directly prune down our sampling to the Prum et al. sampling. However, only 60 species overlap between the two studies (ca. 17% of species in our study). Such sparse sampling could bring bias into the estimates of the Pagel's lambda values because the species that are identical between the two studies are not necessarily evenly distributed across the phylogeny.

Therefore, we pruned our tree to select only taxa that match the 124 families that are also present in the Prum et al. study. This analysis therefore equalized the same represented families between the two studies, avoiding that the larger representation of Passeriformes taxa would bias the estimate. We estimated Pagel's lambda on these two trees (N=124) and still detected significant differences for 7 of 9 morphological traits, except the Tarsus Length and Kipp's distance (average lambda B10K = 0.88, average lambda Prum et al. = 0.78, paired one-sided t-test p value = 0.0014691).

As a second confirmation that passeriform sampling was not driving the results, we removed all but one species of Passeriformes (*Menura novaehollandiae*) from both studies and estimated Pagel's lambda. This removed the impact of greater sampling of Passeriformes in our dataset. The values of Pagel's lambda were still significantly higher for our main tree than for the Prum et al. dataset across all morphological characters (average lambda B10K = 0.98, average lambda Prum et al. = 0.86, paired one-sided t-test p value = 0.0000868), as shown in the figure below.

We have added a statement (addition underlined) into the main text to discuss that sampling within Passeriformes was not the main cause of these differences:

Line 603-605 (line numbers refer to the revised version with tracked changes):

“We found that our main tree fits morphological traits better than the Prum et al. 2 topology, even when controlling for taxon sampling (Fig. 5a), including the larger number of Passeriformes in our study (Supplementary Results in Supplementary File 3).”

In the Methods, we added a description of these experiments:

Line 1162-1164 (line numbers refer to the revised version with tracked changes):

“We also performed a comparison between trees pruned to the 124 families present in both studies. In order to account for the high proportion of Passeriformes in our study, we also excluded all but one passerine from both trees.”

We provide a more detailed description in the Supplementary Methods in Supplementary File 3:

Page 16-17 of Supplementary File 3:

“In order to further account for the differences in taxon sampling between our study as that of Prum et al. study, we performed two tests. First, we pruned both trees to the 124 taxonomic families that overlapped between the two studies. This assured that both trees have the same number of taxa, representing the same lineages. Second, we pruned all but one species of Passeriformes (Menura novaehollandiae) from both tree topologies. This test aimed at testing whether the large number of passeriform taxa in our study impacted the results. Pagel’s lambda was estimated as described above. Results are given in the Supplementary Results below.”

Because we have exhausted space in Extended Data Figure 10 which provides additional data for the section on “Implications for avian diversification”, we have added more detailed treatment of the results of the new analysis to the Supplementary Results in Supplementary File 3.

Page 19 of Supplementary File 3:

“Impact of taxon sampling on phylogenetic signal

When both the main tree and the tree from the Prum et al. study were pruned to have the same families represented (N=124), Pagel’s lambda values were significantly higher for the main topology for 7 out of 9 morphological characters. Kipp’s Distance and Tarsus Length showed no significant difference in inferred Pagel’s lambda values between the two trees (average λ B10K = 0.88, average λ Prum et al. = 0.78, p value = 0.0014691). When trees were pruned to remove all but one species of Passeriformes (B10K N=191, Prum N=155), all 9 morphologically characters fit significantly better to the main tree than to the Prum et al. topology (average λ B10K = 0.98, average λ Prum et al. = 0.86, p value = 0.0000868).”

All raw data has been added to the data repository.

Regarding multivariate analyses instead of univariate treatment. In terms of univariate versus multivariate analyses, we agree that ours is a simple analysis. Its goal was explicitly to compare different variables independently rather than altogether as this shows that there are differences in fit both between different variables and the two topologies. For example, Kipp’s Distance has high Pagel’s lambda across both topologies; or Beak Width has the greatest differences in fit between the B10K and the Prum et al. topology. For these reasons, and due to space limitations, we prefer to reserve a deeper dive with multivariate analyses for a follow up study on evolution of traits along the bird phylogeny.

Line 94 – “phylogenetic loci” – this is unclear; do you mean phylogenetically informative loci?

Response: We rephrased to “Sufficient loci rather than extensive taxon sampling were more effective in resolving difficult nodes.” (L. 95)

The updated Fig 1 has less information than the previous version. I suppose it is a matter of preference which one you use but the smears of color behind the bird icons are distracting and not very aesthetic.

Response: The updated Figure shows the same content in a different layout. The half circular display of the main tree allowed us to incorporate clearer labels of clades and images of birds, as was suggested by another reviewer. These decisions are necessarily subjective, including some design choices, as seen by the positive comments on this figure by the third reviewer. We are happy to remove the smears and replace them with a colored box or something else, if the editor feels that they are not aesthetic.

Lines 244-245 – “In particular, six nodes had quartet support below 35% and another eight nodes had support below 37%.” Why is it important to distinguish these percentages differently? Why not just say 14 nodes below 37%?

Response: We took the reviewer’s suggestion and changed to say 14 nodes below 37%. We had originally distinguished the two because, in theory, branches below 37% could be resolved with 99.9% confidence using 2000 gene trees whereas those below 35% require 10,000 or more gene trees according to the MSC model (see Sayyari and Mirarab, 2017; 10.1093/molbev/msw079). However, as we have no space to go through these details, we decided to remove this distinction.

Lines 260-261 – “Our companion paper...” – the citation for this is incomplete and if I recall correctly, this paper was declined by Nature. This should be reworded.

Response: This paper is now accepted pending minor revisions in PNAS; we will coordinate the publication of the two papers to coincide. We have rephrased the text to read “Mirarab et al. ...”, followed by the citation

Fig. 2c is related to Fig 3a; why not keep those together? I would recommend one figure (2) for discordance and one figure (3) that deals with taxon sampling.

Response: Following the reviewer’s suggestion, we moved Fig. 3a to Fig. 2 as Fig. 2d.

Is GlobalBS the same as multi-locus bootstrapping (MLBS)? This is not a term that is used often and needs better explanation in this paper.

Response: Line 928 in the methods section clarifies that GlobalBS is the same as MLBS, performed in the gene-only mode. We use the term global in order to draw a contrast with the locally estimated posterior probability (localPP) in Fig. 2.

Referee #2 (Remarks to the Author):

The authors have satisfactorily addressed my concerns.

Response: We are glad that reviewer #2 is now satisfied with the paper.

Referee #3 (Remarks to the Author):

I have now gone through the revised version of the manuscript and the point-by-point response from the authors. I stand by my original assessment that this is an impressive piece of research and that the time-calibrated family-level phylogeny of birds will set the standard in bird phylogenomics for some time. In the revised version, and/or in the rebuttal letter, the authors have clarified the major open points; they have improved the figures (I particularly like the new Figure 1 and what is now Figure 2); they improved the "implications" section; and they provide further details (e.g. on evolutionary models) and, importantly, underlying raw data and code to reproduce their findings.

I still find that the manuscript is a bit overloaded and believe that the discussion on increasing taxon sampling versus increasing the number/type of loci is not key to the main story line. However, I also see their arguments in keeping this part as is (If the manuscript had to be shortened, this would be a section that could be moved to the supplement). Also, I am still of the opinion that a cross-validation would have been an elegant strategy with regard to the fossil-calibrations. However, both of these issues are minor.

As a side note, I would like make one additional comment (referring to their response in the rebuttal letter): I am happy to see that the authors agree with me that the number of loci is not the limiting factor anymore. I am less happy about how they responded to this (\$/CPU, CO2 footprint). First, I think we should do science as best as we can. When spending millions of dollars on a research project (just think about the salaries of all people involved in that project), the CPU hours shouldn't matter. Second, \$/CPU hour is constant (and low), irrespective of how long an analysis will take. Third, if the CO2 footprint is an issue, then one should think about using CO2-neutral computing facilities running with renewable energy (It would be very interesting to know how much the computing contributed to the CO2 footprint of this research project compared to all the other parts such as sampling birds in remote locations, shipping the samples, sequencing them, consortia meetings [in person and via zoom], all authors going through that manuscript on their computers, etc.)...

Response: Just one of our analyses, the concatenation, accounted for 7,200 kg of CO2 emissions in our estimates (see below). This is roughly equivalent to 17 people flying from New York to London, and 50% more than the average annual per capita CO2 emissions. Had we not used constraints (the point discussed by the reviewer), we would have needed 2 to 3 times more, without a clear added benefit. And we had many more analyses to conduct. Thus, we believe as we perform these large-scale analyses, we need to remain cognizant of CO2 usage, as argued by Kumar2022 and Grealey2022. Nevertheless, we accept the reviewer's comment that creating more CO2-neutral supercomputers would be beneficial. We also agree that beyond computation, many other sources of emissions are involved in our research. Comparing the costs and benefits of these activities is indeed

difficult, but a worthwhile future endeavor. Thus, we have performed computations that are necessary and have tried to avoid computations we deemed unnecessary.

Estimate of CO₂ emissions for concatenation (20 ML + 50 BS trees, all constrained):

- The analysis took 1,200,000 CPU-hours on the SuperMUC-NG supercomputer (LRZ, Garching, Germany)
- In 2021, LRZ energy consumption was 32,632,950 kWh [LRZ1], and in total 2,308,500,000 CPU-hours were allocated to user jobs [LRZ2]
- On average, this corresponds to roughly 14 Wh per CPU-hour, or 17,000 kWh for the concatenation analysis
- This translates to ~7,200 kg of CO₂ based on carbon intensity of the German electricity mix (0.425 kgCO₂/kWh in 2021 [UBA])

[Kumar2022] Sudhir Kumar, Embracing Green Computing in Molecular Phylogenetics, *Molecular Biology and Evolution*, Volume 39, Issue 3, March 2022, msac043, <https://doi.org/10.1093/molbev/msac043>

[Grealey2022] Jason Grealey, Loïc Lannelongue, Woei-Yuh Saw, Jonathan Marten, Guillaume Méric, Sergio Ruiz-Carmona, Michael Inouye, The Carbon Footprint of Bioinformatics, *Molecular Biology and Evolution*, Volume 39, Issue 3, March 2022, msac034, <https://doi.org/10.1093/molbev/msac034>

[LRZ1] <https://www.lrz.de/wir/berichte/JP/JBer2021.pdf#page=101>

[LRZ2] <https://doku.lrz.de/usage-statistics-for-supermuc-ng-11483092.html>

[UBA]

<https://www.umweltbundesamt.de/themen/co2-emissionen-pro-kilowattstunde-strom-stiegen-in>

Minor comments:

L90: Why does it need reference 4 here, where the authors report what they have done in this study.

Response: The assemblies were already published in reference 4, hence the citation.

L111: Does it really need "widespread" here? If something is incongruent, it is incongruent... Perhaps just say "incongruences".

Response: Removed as suggested.

L130, "through the Chicxulub impact": I would argue that not only the impact itself matters, but also what it caused on a global scale.

Response: We clarified as suggested (change underlined).

L. 130 (line numbers refer to the revised text with tracked changes):

“... requiring survival of multiple neoavian lineages through the global changes causes by the Chicxulub impact”

L158-159: I might be a bit biased here, but working on closely related taxa, it is not really an issue to "create comprehensive datasets ... across many taxa"; perhaps add the notion that this is challenging when dealing with deeper time scales?

Response: We added a statement (change underlined)

L.158-160 (line numbers refer to the revised text with tracked changes): :

“The most compelling solution is also the most challenging: to create comprehensive datasets with whole genomes sampled across many taxa that inform on deeper timescales.”

L171: maybe better "94,402 1-kb loci"? By adding “-“ it is clear that “1” has nothing to do with the number of loci...

Response: Adopted as suggested.

Reviewer Reports on the Second Revision:

Referees' comments:

Referee #2 (Remarks to the Author):

See my comments to the "Confidential comments to the EDITORS ONLY" section.